# SQL-PaLM: Improved large language model adaptation for Text-to-SQL

**Ruoxi Sun**[1]                                                                                     *ruoxis@google.com*

**Sercan Ö. Arik**[1]                                                                            *soarik@google.com*

**Alex Muzio**[2]                                                                                   *amuzio@google.com*

**Lesly Miculicich**[1]                                                                        *lmiculicich@google.com*

**Satya Gundabathula**[2]                                                                *satyakesav@google.com*

**Pengcheng Yin**[3]                                                                            *pcyin@google.com*

**Hanjun Dai**[3]                                                                                    *hadai@google.com*

**Hootan Nakhost**[1]                                                                          *hootan@google.com*

**Rajarishi Sinha**[1]                                                                          *sinharaj@google.com*

**Zifeng Wang**                                                                                       *zifeng@google.com*

**Tomas Pfister**[1]                                                                              *tpfister@google.com*

[1] *Cloud AI Research;* [3] *Google Cloud;* [3] *Google DeepMind*

**Reviewed on OpenReview:** *https://openreview.net/forum?id=rlloVZoKrX*

## Abstract

Text-to-SQL, the process of translating natural language into Structured Query Language (SQL), represents a transformative application of large language models (LLMs), potentially revolutionizing how humans interact with data. This paper introduces the SQL-PaLM framework, a comprehensive solution for understanding and enhancing Text-to-SQL using LLMs, using in the learning regimes of few-shot prompting and instruction fine-tuning. With few-shot prompting, we explore the effectiveness of consistency decoding with execution-based error filtering. With instruction fine-tuning, we delve deep in understanding the critical paradigms that influence the performance of tuned LLMs. In particular, we investigate how performance can be improved through expanded training data coverage and diversity, synthetic data augmentation, and integrating query-specific database content. We propose a test-time selection method to further refine accuracy by integrating SQL outputs from multiple paradigms with execution feedback as guidance. Additionally, we tackle the practical challenge of navigating intricate databases with a significant number of tables and columns, proposing efficient techniques for accurately selecting relevant database elements to enhance Text-to-SQL performance. Our holistic approach yields substantial advancements in Text-to-SQL, as demonstrated on two key public benchmarks, Spider and BIRD. Through comprehensive ablations and error analyses, we shed light on the strengths and weaknesses of our framework, offering valuable insights into Text-to-SQL's future work.

# 1 Introduction

Text-to-SQL aims to automate the process of translating natural language questions into SQL queries that can be executed directly on a database (Androutsopoulos et al., 1995; Hristidis et al., 2003; Li & Jagadish, 2014; Wang et al., 2017). As illustrated in Fig. 1, Text-to-SQL bridges the gap between the way humans naturally communicate, using language, and the way databases are structured, and has the potential to revolutionize how humans interact with data (Zhong et al., 2017; Yu et al., 2018; Li et al., 2023d). Making databases accessible to non-expert users through natural language, Text-to-SQL can empower humans to extract valuable information without needing specialized SQL knowledge (Wang et al., 2019; Scholak et al., 2021; Cai et al., 2021; Qi et al., 2022; Li et al., 2023b; Pourreza & Rafiei, 2023; Gao et al., 2023a; Sun et al., 2023; Chen et al., 2023). This not only enhances the efficiency of data analysis but also broadens the use of databases to

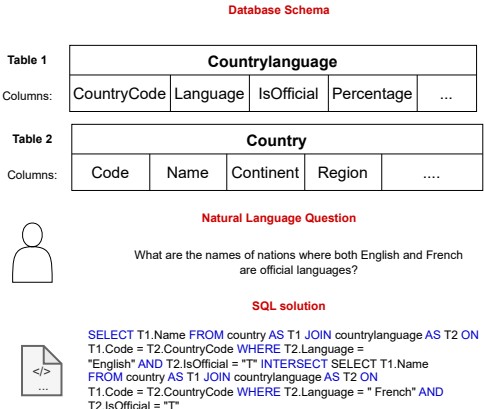

Figure 1: Text-to-SQL systems are developed to transform queries expressed in natural language into Structured Query Language (SQL) based on the information from databases.

a wider range of applications. Intelligent database services, platforms for automated data analytics, and more sophisticated conversational agents capable of understanding and responding to complex data-related questions can all be fueled by advancements in Text-to-SQL (Yu et al., 2019; Gu et al., 2022; Pérez-Mercado et al., 2023; Xie et al., 2023).

At a high level, the Text-to-SQL is a sequence-to-sequence modeling task (Sutskever et al., 2014), with both the database schema and natural language question being converted into a linear input sequence, and the SQL being the target output sequence. Early works attempt to fine-tune domain-specific Transformer architectures or decoding approaches tailored for Text-to-SQL utilizing SQL syntax, semantics or the intricate relationship between questions and databases (Scholak et al., 2021; Qi et al., 2022; Li et al., 2023a; Wang et al., 2019; Bogin et al., 2019; Cai et al., 2021; Hui et al., 2022). Recent years have witnessed the burgeoning application of large language models (LLMs) to Text-to-SQL (Rajkumar et al., 2022; Liu et al., 2023a; Gao et al., 2023a; An et al., 2023; Tai et al., 2023; Pourreza & Rafiei, 2023; Chen et al., 2023; Sun et al., 2023). Along this line, most of the research has focused on leveraging prompting to translate user utterances into SQL queries (Rajkumar et al., 2022; Liu et al., 2023a). More advanced prompting methods has domain-specific adoptions to improve understanding natural language questions and structured database schemas, such as selecting better few-shot exemplars based on question similarity (Gao et al., 2023a; An et al., 2023), decomposing complex questions into sub-tasks (Tai et al., 2023; Pourreza & Rafiei, 2023), verifying the correctness of model-predicted SQL queries through execution feedback (Chen et al., 2023; Sun et al., 2023; Pourreza & Rafiei, 2023), as well as linking NL phrases (e.g. *"nation" in the question in Fig. 1*) to relevant database constructs (e.g. the `Name` column in the `County` table, see (Pourreza & Rafiei, 2023)).

While these few-shot prompting methods have significantly improved Text-to-SQL performance, it still remains unclear whether prompting alone is adequate to handle real-world challenges. As we elaborate in Sec. 2, real-world Text-to-SQL exhibits a variety of challenges. Specifically, a user's natural language questions are often ambiguous (e.g. *"sales in California"*) and can have multiple plausible interpretations (e.g. *sales made by Californian businesses* or *produces sold in the states*). Those questions might also come with semantic constraints (e.g *Who was the president before Joe Biden*) that map to complex SQL queries (e.g. requires reasoning steps that first retrieve the beginning date of Biden's term and then identify the last record whose end date is before that). Moreover, real-world databases may contain large volumes of tables and columns, and the sheer content size would easily exceed the context limit of LLMs. Components in a schema could also have rich data types (e.g. `string`s or `datetime`s) with complex dependencies defined by primary-foreign key mappings, requiring non-trivial SQL queries to process such data. In addition to those inherent challenges, collecting aligned examples of questions and SQL queries for learning also requires

laborious annotation efforts by domain experts, impeding the process of scaling-up data hungry LLMs for Text-to-SQL.

As a first step towards addressing those challenges, in this paper, we propose *SQL-PaLM*, a holistic framework that adapts a LLM, PaLM-2 (Anil et al., 2023) Unicorn variant, for Text-to-SQL tasks. We start with evaluating *SQL-PaLM*'s performance with prompting, and then we focus on *SQL-PaLM*'s tuning as it leads to better performance in challenging scenarios. Apart from existing work that mostly focus on few-shot prompting strategies or tuning relatively smaller LLMs, *SQL-PaLM* focus on *tuning a large LLM*. Larger models have different behaviors with their emergent abilities, a phenomenon of significantly improved understanding and reasoning performance compared to smaller LLMs (Wei et al., 2022a). We systematically explore large models' potential for Text-to-SQL and study the research topics along the key aspects presented in Sec. 4 . Through extensive experiments and analyses, we unravel multiple key factors that influence the LLMs' performance when adapting to Text-to-SQL. First, diversity and coverage of train data can be crucial – we present ablation studies on training data mixture and provide takeaways across tasks and benchmarks. To improve training data coverage with low human cost, *SQL-PaLM* also proposes augmenting with large-scale LLM-generated synthetic data. Second, input representations can greatly influence the overall quality. We present an in-depth study on fine-tuning Text-to-SQL to better leverage different types of information-bearing content, such as database values, column descriptions, and hints. Next, scaling to real-world database sizes would be important for adoption. We present an efficient column selection approach that only encodes information from the subset of relevant database columns as inputs to LLMs. This approach significantly reduces the context size with negligible impact on performance. In particular, we propose a program-aided column selection and retrieval-based column selection approach, We study integration with both *hard* (i.e. with removal of unselected columns) and *soft* (i.e. emphasizing on selected columns) column selection approach into the overall Text-to-SQL pipeline. Finally, we propose execution-based test-time refinement to integrate multiple training paradigms based on execution feedback.

Our contributions can be summarized as follows:

- We focus on *large* LLMs on Text-to-SQL and investigate from multiple important angles including learning perspectives (prompting vs tuning), task perspectives (judiciously selecting input components) and real-world scaling perspectives (e.g. column selection). Through thorough experiment and analysis, we systematically identify components in influencing performance.

- We demonstrate that mixing training data with diverse sets, despite having various formats, can benefit LLMs, indicating the potential of tuned LLMs for superior generalization ability. Complex SQL data particularly have been observed to be more beneficial as tuning data.

- We study effective methods to utilize database content and auxiliary information, such as descriptions. We show improvements with the relevant subset of them included in the prompt while irrelevant information can harm the performance.

- For large-scale databases, we introduce a column selection approach that significantly reduces the length of the inputs going into the LLMs, while yielding negligible impact to performance (*hard* column selection). Among the two proposed approaches, program-aided column selection has higher column selection accuracy, whereas retrieval-based approach is more cost-effective and controllable. *soft* column selection, which doesn't exclude unselected columns, has a higher performance than *hard* column selection, albeit it necessitates LLMs with longer context length.

- We propose a test-time execution-based selection approach to integrate multiple setups to further improve performance.

- Focusing on challenging SQLs that are close to real-world use, we provide in-depth error analysis and case study to reveal the advantages and disadvantages of the proposed approaches.

## 2 Text-to-SQL Challenges

Real-world Text-to-SQL scenarios present a broad spectrum of challenges stemming from the complexity of natural language questions, database structures, inherent SQL intricacies, and data availability. These real-world challenges are often more severe than those encountered in academic benchmarks.

**Natural language question phrasing**: The ambiguity and complexity of natural language questions pose challenges. Input phrases may have multiple interpretations in natural language (e.g. "sales in California" could refer to sales made by people in California or products sold to people in California). They might also come with complex sentence structures such as subordinate clauses and relative clauses. The meaning may also depend on the surrounding context, making it difficult to derive correct interpretations. In addition, databases and applications come from a wide range of domains, and Text-to-SQL systems need to understand domain-specific terminology, which can vary greatly depending on the use case. Specific rules, regulations, formulas or calculations related to a particular domain might need to be applied to generate the correct SQL. For example, the question *"List disease names of patients with proteinuria levels above normal"* requires LLMs to have domain knowledge *"proteinuria level above normal means U-PRO >= 30"* to solve the problem.

**Sizes and diversities of databases**: Real-world large-scale databases might contain numerous tables and columns. The sheer volume of columns can exceed prompt length limits, making including the entire dataset schema impossible. Furthermore, LLMs face difficulties in efficiently accessing and utilizing information within lengthy input contexts as discussed in the phenomenon of *"lost in the middle"* (Liu et al., 2023b). Database schemas often have complex and various structures, including differences in table names, column names, and their relationships. The relationships between tables may not be explicitly defined in the schema, requiring the system to infer them (by understanding *"foreign keys"*). Moreover, database schemas might contain ambiguities – table and column names in a schema may not be informative (e.g. *The column name "property" is vague about the feature it denotes, as opposed to "size" clearly specifies the type of property*) or abbreviated in a way that is not easily understood (*such as "cname"*). Additionally, some tables might contain tens of similar columns (*"sku1", "sku2", ... "sku100"*), potentially leading to confusion for the Text-to-SQL system.

Besides ambiguities in schemas, database contents also have a wide variety of types and formats. The data stored in the database can also vary significantly in types and format, leading to lengthy and specific clauses (*e.g. regular expression, and type casting*) to extract variable information. (1) Type: they include scalars, arrays, nested arrays etc. (2) Format. For instance, the year of *2024* might be saved as a string: *"2024"*, a number: *2024*, or in other forms such as *"year2014"*, or *"2014-01-01"*. Extracting different formats of "year" requires different regular expressions. We show a real-world example in Fig. 2, where extraction of the proper format of the column *"revenue"* requires using *"CASE"* statements, casting string into numbers, schema interpretation, and regular expressions, because the values of revenue is stored in different string formats – single values (i.e. *i.e. "100"*) or ranges (*i.e. "100-200"*).

**Inherent SQL complexity**: Certain SQL queries are complex in nature, marked by the use of multiple SQL keywords, the inclusion of nested sub-queries, a variety of column selections or aggregations, the application of conditional statements, and the involvement of joins across multiple tables.

---

**A challenging real-world example**

Extract the "revenue" values: cast to number if one number ("100"), take average if a range ("100-200")

```
case
  when regexp_contains(revenue, '-')
  then
    (cast(regexp_extract(revenue, r'
    ^(\d+)') as int64)  +  cast(
    regexp_extract(revenue, r'-(\d+)\
    $') as int64)) / 2
  else
    cast(regexp_replace(revenue, r'
    [^0-9]', '') as int64)
  end
```

Figure 2: An example SQL sample corresponding to complex arithmetic operations and handcrafted rules on a complex database.

**Data-centric challenges**: In general, paired (text, SQL) data can be very costly to obtain (Yu et al., 2018), so even the highly popular publicly-available dataset sizes are often much smaller than other text processing applications. Given the challenge of curating such datasets, it is not rare to observe inconsistent, incomplete, or incorrect ground truth data (e.g. human annotators making errors[1]), which might affect the quality of the models trained on them.

## 3    Related Work

### 3.1    Approaches with Deep Neural Networks

**Sequence-to-sequence models** Text-to-SQL can be formulated as a sequence-to-sequence modeling problem, with both the database schema and natural language question being converted into a linear input sequence, and the SQL being the target output sequence. Prior to the recent advances of large language models (LLMs), the approach of fine-tuning Transformer models, such as T5 (Raffel et al., 2020), with SQL-specific customizations had been the prevalent approach dominating the state-of-the-art. PICARD (Scholak et al., 2021) introduces a technique that discards invalid beam search candidates during inference, improving the grammatical correctness of the SQL queries. RASAT (Qi et al., 2022) augments transformer architecture with relation-aware self-attention which is efficient to incorporate a variety of relational structures while also leveraging a pretrained T5 model. RESDSQL(Li et al., 2023b) proposes a ranking-enhanced encoding and skeleton-aware decoding framework to decouple the schema linking (e.g. table or column names) and the skeleton parsing (e.g. keywords).

**Graph encoders for schema understanding** Another line of work is based on employing graph encoders to explicitly model complex relationships within the database schemas and questions. RAT-SQL (Wang et al., 2019) introduces schema encoding and linking, and models the schema and its relationships as a graph. Global-GNN (Bogin et al., 2019) further explores this concept of depicting the intricate structure of a database schema with a graph. SADGA (Cai et al., 2021) employs both contextual and dependency structures for encoding the question-graph, and utilizes database schema relations in the construction of the schema graph. S2SQL (Hui et al., 2022) incorporates syntactic dependency information into the relational graph attention network.

### 3.2    Text-to-SQL with LLMs

**Prompting LLMs** Recent advances in LLMs have yielded groundbreaking capabilities (Chowdhery et al., 2022; Achiam et al., 2023) – their ability to understand, generate, and reason in unprecedented ways with prompting has amplified their penetration into many real-world tasks. Numerous advanced prompting techniques have further extended LLMs' capability, such as Chain-of-Thought (Wei et al., 2022b), Least-to-Most (Zhou et al., 2022), and others (Chen et al., 2022; Yao et al., 2023; Besta et al., 2023). However, these generic-purpose prompting approaches are observed to fall behind on Text-to-SQL tasks compared to approaches tailored to Text-to-SQL[2] (Rajkumar et al., 2022; Liu et al., 2023a; Li et al., 2023d).

To further improve general-purpose prompting methods for Text-to-SQL specifically, advanced prompting approaches tailored to Text-to-SQL task, have been proposed. DIN-SQL (Pourreza & Rafiei, 2023) exemplifies this by breaking down the Text-to-SQL tasks into sub-tasks: schema linking, classifying SQL by difficulty level, SQL generation based on SQL difficulty, and self-correction[3], and etc. CoT-style (Tai et al., 2023) also proposes decomposing Text-to-SQL into sub-problems and presents them all at once to LLMs instead of solving sub-problems iteratively. SQLPrompt (Sun et al., 2023) enhances LLMs with diverse representations of database schemas and questions as inputs to encourage diverse SQL generation to improve performance from execution-based consistency decoding. Self-debugging (Chen et al., 2023) appends error messages to the prompt and performance multiple rounds of few-shot prompting to self-correct the errors. DAIL-SQL (Gao

---

[1]For example, BIRD datasets Li et al. (2023d) contain errors, as we described in error analysis in Sec. 8.7

[2]For example, the standard single-pass prompting approach of LLMs, such as with PaLM-2 or GPT-4 on the development (dev) set of the Spider dataset underperforms fine-tuned smaller capacity models, such as Picard Scholak et al. (2021) and RESDSQL (Li et al., 2023a)

[3]Notably, DIN is the first few-shot prompting that can outperform strong tuning-based alternatives, such as Picard Scholak et al. (2021) and RESDSQL (Li et al., 2023a)

et al., 2023a) provides an investigation on prompt designs, including question representations and example selection on few-shot prompting. In addition, another line of work is selecting similar few-shot demonstrations with the input questions so that LLMs can follow the solution of a similar question. Among those, DAIL-SQL selects based on similarity of embedding of questions plus SQL queries, whereas SKILL-KNN(An et al., 2023) selects based on similarity of the required skills. Another line of research comes from the perspective of agent-based systems, such as MAC-SQL (Wang et al., 2024) and DIN-SQL (Pourreza & Rafiei, 2023), where function modules are framed in the context of an agent's operations. These agent modules typically include a selector (which reduces a large database into a smaller sub-database), a decomposer (which breaks down complex questions into simpler sub-queries), and a refiner (which utilizes SQL execution feedback to improve queries).

**Fine-tuning LLMs** Instruction tuning on coding tasks have achieved remarkable performance on different programming languages (e.g. Python and SQL) (Luo et al., 2023; Muennighoff et al., 2023; Li et al., 2023e), indicating the immense potential of fine-tuning. However, regarding Text-to-SQL, compared to prompting approaches, tuning approaches have been relatively under-explored, partially attributed to the prohibitively high computational cost. DAIL-SQL (Gao et al., 2023a) has investigated fine-tuning open-source LLMs (e.g. LLaMA). Their results suggest that although fine-tuning yields significant enhancements, the performance of fine-tuning open-source LLMs, attributed to their smaller sizes, remains substantially lower than prompting larger models like PaLM-2 or GPT-4. Encouragingly, concurrent work CodeS (Li et al., 2024) ("CodeS-15B") applies Text-to-SQL specific adaptation and achieves impressive results. Unlike existing work, we primarily focus on LLMs at larger scales, to investigate the potential of achieving significant gain with the increase of model size due to the emergent ability of large models (Wei et al., 2022a).

**Schema linkage** Schema linkage, which connects phrase in questions to those in the database schema, is often incorporated as a guidance module into Text-to-SQL. IRNet (Guo et al., 2019) performs schema linkage to generate custom type vectors to augments the embedding of questions and schema, that results in improvements in recognition of the columns and the tables mentioned in a question and superior generation of intermediate representations with abstract syntax tree decoder. RAT-SQL (Wang et al., 2019) integrates schema linkage information directly into the self-attention layers of its encoder, along with question and schema. With end-to-end training on Text-to-SQL, the encoder-decoder transformer learns to effectively utilize these linkage cues, enhancing SQL generation abilities. RESDSQL (Li et al., 2023a) employs a ranking-enhanced encoder to guild the model to prioritize the most pertinent elements for SQL generation. The ranking is obtained by training a cross-encoder is for schema linkage, classifying schema items based on their relevance to the question. Our proposed "column selection" approach is different from previous methods as we rely on LLM's intrinsic reasoning ability to infer relevant columns, whereas previous work relied on either pattern matching or learning trainable parameters through end-to-end Text-to-SQL training. In addition, the purposes are different that ours is an independent module, which serves as a prepossessing step to enable applying Text-to-SQL to large-scale datasets that exceed the prompt length, whereas others are proposed as parts of their Text-to-SQL methods that are hard to decouple from the overall Text-to-SQL pipeline.

**Retrieval-based methods** incorporate retrieval-augmented generation and focus on real-world database content. Zhang et al. (2023) introduces a retrieval-augmentation approach that enhances the structural understanding of SQL by leveraging similar past queries to inform the generation process, addressing the gap between specific structural knowledge and general knowledge. Nan et al. (2023) demonstrates the effectiveness of retrieval-based approaches in selecting diverse and relevant demonstrations through prompt design strategies. Wang et al. (2023) decouples the Text-to-SQL process into schema routing and SQL generation, and employs a compact neural network-based router for effective navigation through large-scale schemas, complemented by LLMs for SQL generation.

**Synthetic data methods** Regarding synthetic data for Text-to-SQL task, GraPPa (Yu et al., 2020) and TaBERT (Yin et al., 2020) propose pretraining frameworks to jointly train structured schema of DB tables and questions. GraPPa proposes a synchronous context-free grammar: extracts question-SQL template in a "context-free" manner from existing dataset, and applies them to other databases. Zhao et al. (2022) proposes a synthetic framework with emphasis on "strong-typing", "key relations", only including "schema-distance-weighted" relevant columns. Wang et al. (2021), Wu et al. (2021) synthesize SQL using context-free grammar

rules (abstract syntax tree grammar), and then generate the corresponding natural question based on the synthesized SQLs". Unlike other work, this work focus on validity of the generated SQL to better benefit from synthetic data.

## 4  Key Aspects of the *SQL-PaLM* Framework

We investigate multiple key aspects of building a Text-to-SQL framework in this paper. Through extensive experimental validation and in-depth analyses, we aim to systematically unravel the factors influencing Text-to-SQL performance.

**Learning perspective – pushing adaptation with prompting vs. tuning:** Central to our investigation is understanding the intrinsic property of LLMs on tackling the Text-to-SQL task, whereby we explore, within the learning paradigms of few-shot prompting and tuning, how LLMs solve Text-to-SQL tasks differently under different learning scenarios, and what factors influence the final performance significantly. Compared with few-shot prompting approaches, tuning approaches have been relatively under-explored, partially due to the prohibitively high computational cost. In this paper, therefore, focus on some pivotal questions for tuning, such as: How does the performance of prompting strategies compare with that of tuning strategies? To what extent does the performance rely on the foundation models' capacity? How well do models generalize across different datasets, especially when faced with limited training data (considering notable publicly-available datasets like Spider and BIRD are not significant for large model size)? What is the impact of parameter-efficient tuning techniques, like LoRA (Hu et al., 2021), on the training performance? How does tuning depend on different foundation models (e.g. PaLM vs. LLaMA)?

**Task perspective – judiciously selecting input components for Text-to-SQL:** The Text-to-SQL task contains a range of potentially-useful information that can be taken advantage of: (i) *database schema*: there are table & column names, descriptions (e.g. clarifications such as abbreviations), data types (e.g. string and integer), data formats (e.g. explanations of formats stored within the database such as with a raw value or an interval), and primary & foreign keys that describe how different tables are connected to each other; (ii) *database content values*, some database entry values are needed to solve the question[4]; (iii) *natural language questions*, whether there are associated hints or formulas, such as the domain knowledge. Each of the above information can be quite critical – with some of them missing, LLMs cannot produce accurate SQL outputs. On the other hand, in some scenarios, they can add up to be lengthy and contain irrelevant details, wiht the potential to distract the LLMs from focus on relevant information and generate the correct SQL outputs. Therefore, within the limited input length, it is crucial to achieve a balance on not missing critical information, and not providing a significant amount of irrelevant information. This necessitates judicious selection of a subset of features for Text-to-SQL model to produce correct SQLs. Some fundamental questions arise: How can we enable LLMs to effectively utilize various forms of available information? Which information sources are most valuable? What are the linear formats to effectively represent the various inputs for LLMs?

**Real-world scaling – column selection:** With the advances in LLMs, numerous challenges associated with Text-to-SQL[5] can be addressed more effectively, bringing us closer to resolving real-world database challenges. However, one remaining key obstacle is the potentially high number of columns. Real-world databases, such as those representing the full inventory of large-scale retailers, might contain a large number of attributes[6]. Directly processing this volume of columns is often infeasible, as the concatenation of column names would exceed the prompt length limits of LLMs. This highlights the need for column selection — the process of identifying a relevant subset of columns from multiple tables. Column selection is related to the broader process of *schema linking*[7], which maps phrases in the natural language question to corresponding columns

---

[4]For instance, Question: What is the revenue for shoes? In the database, the column "product" contains "Running shoes"; Without providing database content, such as "column 'product' contains 'Running shoes'", the output SQL is likely to be "SELECT ... WHERE product='shoes'", because "shoes" is mentioned in the question. However, the correct SQL is "SELECT ... where product='Running shoes'"

[5]For instance, public benchmarks, such as Spider

[6]The number of columns can be hundreds or thousands

[7]Major schema linkage approaches (Guo et al., 2019; Wang et al., 2019; Li et al., 2023a) are discussed in Sec. 3.

and tables in the database schema. The difference is that column selection focuses solely on identifying the set of relevant columns from the schema, without linking.

## 5 Problem Formulation

Text-to-SQL systems transform queries expressed in natural language into SQL programs. Provided a natural language query $Q$, and an associated database $D$, SQL outputs are generated such that when executed against database $D$, would generate the answer to the original natural language query $Q$.

A database $D$ includes two primary components: the schema (including table and column names) and the contents (entry values) of $D$. The schema, represented by $S$, outlines a database's structure and includes a set of table names $T$, and a set of column names $C$. The database content values $V$ are the data values that populate the entries of the tables, adhering to the attributes defined by the database schema.

The database schema $S$ contains $n_T$ tables:

$$S = \{S_T^{(1)}, S_T^{(2)}...S_T^{(n_T)}\} \tag{1}$$

with the $k$-th table schema $S_T^{(k)}$ consisting the table name $T_N^{(k)}$ and a collection of columns $C^{(k)}$, where $j$-th column name is represented by $C_j^{(k)}$. The $k$-th table contains $n_{col}^{(k)}$ number of columns:

$$S_T^{(k)} = \{T_N^{(k)}, C^{(k)}\}$$
$$C^{(k)} = \{C_1^{(k)}, C_2^{(k)}, ..C_j^{(k)}, ..\}_{j=1:n_{col}^{(k)}}.$$

The database content values of the $k$-th table are $V^{(k)}$, a $n_{row}^{(k)} \times n_{col}^{(k)}$ matrix with each row being a data entry and each column being a vector of values for an attribute:

$$V^{(k)} = \{v_1^{(k)}, v_2^{(k)}, ..v_j^{(k)}, ..\}_{j=1:n_{col}^{(k)}}, \tag{2}$$

where $v_j^{(k)}$ are vectors of length $n_{row}^{(k)}$ encompassing all the values of the attributes $C_j^{(k)}$. Usually, the number of entries $n_{row}^{(k)}$ is significantly larger than the number of attributes $n_{col}^{(k)}$. Primary keys $K_P^{(k)} \in C^{(k)}$ are the column(s) that contain values that uniquely identify each row in a table[8]. Foreign keys $K_F^{(k)} \in C^{(k)}$ are the column(s) in one table, referring to the primary key in another table. They are used to link multiple tables.[9] Additionally, databases often include supplementary information for clarification (such as the detailed descriptions of columns) which help interpreting ambiguous or uninformative column names (as explained in Sec. 2). We use $f_{des}(C)$ to indicate the descriptions for column $C$. For $k$-th table, $des^{(k)}$ refer to a collection of column description:

$$des^{(k)} = \{f_{des}(C_j^{(k)})\}_{j=1:n_{col}^{(k)}}. \tag{3}$$

Lastly, there can be hints $H^{(k)}$, user-specified aids for the question. They could contain definitions or formula used to construct SQL queries.[10]

---

[8]For example, the column "StudentIDs" of the "Student" table.

[9]For example, consider the table "Student" with primary key "StudentID", and the table "BookOrders" which has two columns: "OrderID" and "Buyer". The "Buyer" column contains a series of StudentIDs representing the individuals who placed the book orders. In this scenario, "BookOrders.Buyer" is the foreign key which points to a primary key "Student.StudentID". This way, each row in the "BookOrders" table can be associated with a specific student from the student table.

[10]For instance, for the query "List the phone numbers of schools with the top 3 SAT excellence rates", the hint is the definition of the excellence rate, the percentage of take takers with SAT score greater than 1500. "Excellence rate = NumGE1500 / NumTstTakr", where column "NumGE1500" refers to "Number of Test Takers Whose Total SAT Scores Are Greater or Equal to 1500" and "NumTstTakr" refers to "Number of Test Takers"

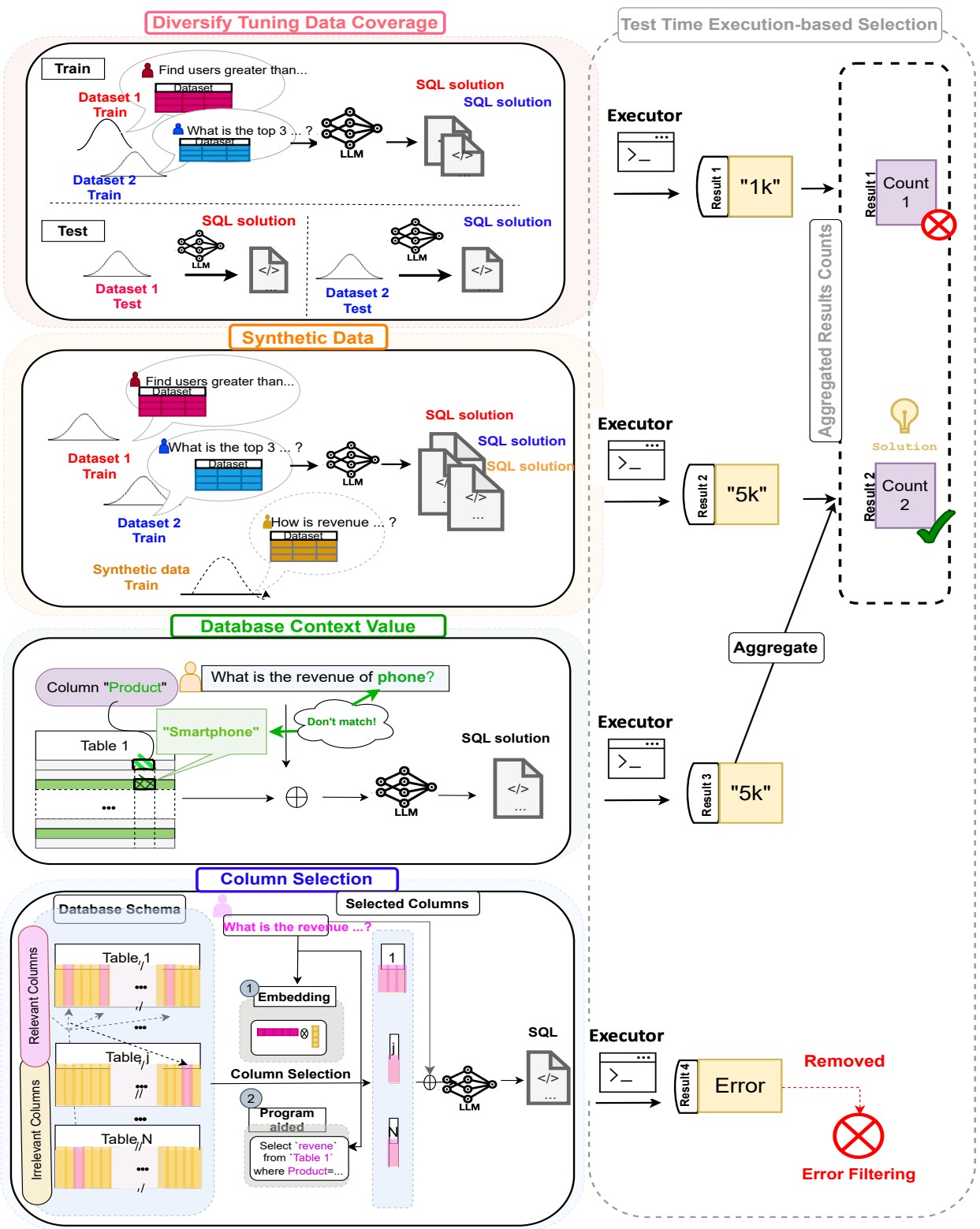

Figure 3: **Overview of the *SQL-PaLM* framework for fine-tuning**. Our framework incorporates different submodules for superior Text-to-SQL performance: (i) diversifying training data coverage (Sec. 6.3.2), (ii) incorporating synthetic data (Sec. 6.3.3), (iii) including database content (Sec. 6.3.4 & 6.3.5), (iv) test-time refinement mechanism (Sec. 6.3.6).

# 6 Methods

This paper presents the *SQL-PaLM* framework (depicted in Fig. 3), a holistic approach to push Text-to-SQL capabilities using LLMs with both few-shot prompting and instruction-tuning. We first describe the input representations (Sec. 6.1) used for both learning paradigms. For few-shot prompting, we propose a prompting approach leveraging execution-based error-filtering-aided consistency decoding (Sec. 6.2). For instruction-tuning, we delve deeply into understanding the critical factors in influencing performance of tuning LLMs, including expanded training data coverage and diversity (Sec. 6.3.1 and Sec. 6.3.2), synthetic data augmentation (Sec. 6.3.3), and integrating query-specific database content (Sec. 6.3.4). Using the test-time selection approach (Sec. 6.3.6), we further enhance accuracy by integrating SQL outputs from various paradigms, leveraging execution feedback for refinement. Furthermore, we address one of the real-world challenges of navigating complex databases with a significant number of tables and columns, presenting effective methods for precise selection of pertinent database components to improve Text-to-SQL performance (Sec. 6.3.5).

## 6.1 Input representation

The primary step of modeling with LLMs is providing judiciously-designed input representations. We start from the database schema $S$ and primary and foreign keys. Following Shaw et al. (2020); Scholak et al. (2021); Sun et al. (2023), we serialize the database schema as:

$$X_1 = \overline{|T_N^{(1)} : C_1^{(1)}(d_1^{(1)}), C_2^{(1)}(d_2^{(1)}), ... C_{n_{col}^{(1)}}^{(1)}(d_{n_{col}^{(1)}}^{(1)})|T_N^{(2)} : C_1^{(2)}(d_1^{(2)}), C_2^{(2)}(d_2^{(2)}), ... C_{n_{col}^{(2)}}^{(2)}(d_{n_{col}^{(2)}}^{(2)})|...|K_P; K_F;}, \quad (4)$$

where $T_N^{(k)}$ denote the $k$-th table name represent the $j$-th column name of the $k$-th table, and $d_j^{(k)}$ indicate its data type (such as number or string). We use the symbol '|' to represent the boundaries between different tables in the schema. Within each table, we use ':' to separate the table name from its columns, and we indicate each column via the delimiter ','. We integrate the primary keys ($K_p$) and foreign ($K_f$) keys to denote relationships between tables. We refer the above schema design as "concise prompt", as it concisely presents the table structure [11]. See Fig. 4 as an input example for the question given in Fig. 1 and a realistic example in Sec. A.10.1 in Appendix.

Besides the database schema, other forms of database content can be beneficial. We use database content values $V(Q)$ to match with the tokens in question to clarify the database value (see Sec. 6.3.4). We also incorporate column descriptions ($des$) and hints ($H$), which provide additional clarification or domain-specific knowledge when applicable. We concatenate them together as:

$$X_2 = \overline{des; V(Q); H}.. \quad (5)$$

Combining all, we form the overall input sequence by concatenating $X_1$, $X_2$, the natural language query $Q$, and a SQL initiation marker "[SQL]":

$$X = f(\overline{X_1; X_2; Q; [\text{SQL}]}), \quad (6)$$

where $f$ is the prompt design to connect the concatenated components (e.g. '[database schema] is ...; [Primary keys]: ..') and human instructions ("Convert text to SQL") explicitly in $X$. See Fig. 4 as an example.

---

[11] An alternative way to describe the schema would be "verbose prompt", where we use human language to describe the schema verbosely. See the example in A.10.2.

---

**Input:** $X$ **(Eq. 6)**

**Convert text to SQL**
« Few-shot examples »
**[Schema]**: | Countrylanguage: CountryCode (Number), Language (String), ... | Country: Code (Number), Name (String), ...
**[Primary Keys]**: Countrylanguage: CountryCode | Country: Code ...
**[Foreign Keys]**:Countrylanguage: CountryCode is equivalent to Country: Code | ...
[Detailed descriptions of tables and columns]: # column description
The column 'IsOfficial' in Table 'Countrylanguage' has column descriptions of "whether language is official language"
...
[Database values that related with questions]: database content
The column 'Language' in Table 'Countrylanguage' has database values: ['English', 'French']
...
[Additional Info]: # hints, if applicable
**[Q]**: What are the names of nations where both English and French are official languages?
**[SQL]**:

---

Figure 4: The overall input representation from Eq. 6. The basic prompts are in black ($X_1$ in Eq. 4), and auxiliary information is gray ($X_2$ in Eq. 5), if the datasets have the corresponding information.

## 6.2 Prompting LLMs for few-shot learning

We first consider the use of LLMs for Text-to-SQL with prompting. Given an $\text{LLM}_\theta$ and a question $Q$, represented as a sequence of tokens, the prediction in zero-shot prompting can be formulated as:

$$Y = \arg\max_Y P_{\text{LLM}_\theta}(Y|X), \tag{7}$$

where $Y$ are the inferred SQL outputs, $X$, as described in Sec. 6.1, are the input prompts including database schema, other auxiliary information (i.e. hints) if applicable, and the question $Q$. $\theta$ are the parameters of the pretrained LLM. With few-shot prompting, the formulation is extended to LLMs generating a sequence of tokens conditioned on the provided demonstrations pairs $demo = [(X_1, Y_1), (X_2, Y_2), ...]$:

$$Y = \arg\max_Y P_{\text{LLM}_\theta}(Y|demo, X). \tag{8}$$

Essentially, in few-shot prompting, the prompts prepend the natural language queries $Q$ with a list of demonstrations (inputs, SQL) pairs, and the LLM follows the input to generate answers in an auto-regressive way.

To further enhance performance beyond the standard few-shot prompting, we adapt an execution-based consistency decoding method, following (Sun et al., 2023; Ni et al., 2023). This method leverages on the unique benefit of coding task, including Text-to-SQL task, where the SQL outputs are executable. This serves as a preliminary validation for the generated output, allowing us to identify invalid results more easily. Concretely, our approach involves the following steps:

**Step 1: Sample multiple SQL outputs from LLMs**: Given an input $X$, we sample $m$ SQL outputs from the $\text{LLM}_\theta(\text{Y|X})$ using a sufficiently high temperature, i.e. $\Omega = \{\hat{Y}_i\}_{i=1}^m \sim P_{\text{LLM}_\theta}(Y|X)$.

**Step 2: Verify with execution** The generated SQL outputs, $\{\hat{Y}_i\}$, are subsequently executed using an executor $\mathcal{E}(\cdot)$, which yields the corresponding results denoted as $e_i = \mathcal{E}(Y_i)$.

**Step 3: Aggregate Execution Result**. Since multiple SQLs are valid for the same question, we aggregate the SQL outputs in $\Omega$ that give the same execution result. The execution output with errors is guaranteed to be wrong, and the execution outputs with the most occurrences are more likely to be correct (Wang et al., 2022). We remove programs with invalid execution results to update the LLM generation probability with the verification probability, and marginalize over SQL outputs with the same execution results. We use this aggregated probability as the ranking score $R$:

$$R(X, \hat{Y}) = \sum_{Y \in \Omega} P_{\text{LLM}_\theta}(\hat{Y}|X) \cdot \mathbb{1}\big[\mathcal{E}(Y) = \mathcal{E}(\hat{Y})\big] \cdot \mathbb{1}\big[\mathcal{E}(\hat{Y}) \in \Phi\big], \tag{9}$$

where $\Phi$ indicate the set of valid execution results without yielding errors (denoted as "execution error filtering"). We choose the outputs $Y^*$ that execute the most probable result

$$Y^* = \arg\max_{\hat{Y} \in \Omega} R(X, \hat{Y}). \tag{10}$$

### 6.3 Model tuning

Despite the significant advances achieved with few-shot prompting of LLMs, it remains a formidable challenge for a pretrained LLM to rely solely on its parametric knowledge and prompting to accurately process highly-complex SQL queries (Sec. 2). Such queries often involve sophisticated semantic logic, complex database schemas, and database contents with numeric edge cases necessitating extensive clauses [12] for each case (See Fig. 2). Additionally, the SQL logic may require the use of clauses that were underrepresented in the pretraining corpus [13]. To address these more intricate scenarios, this section focuses on tuning, wherein we refine the pretrained LLMs to better align with a customized Text-to-SQL distribution. This paper delves into crucial training paradigms that influence the tuning efficacy of LLMs, including expanding the range and diversity of training data, leveraging synthetic data, integrating query-specific database content, and optimizing table and column selection. We then introduce a test-time selection approach that integrates these diverse training paradigms, aiming to enhance accuracy through the utilization of execution feedback.

#### 6.3.1 Instruction tuning

To improve the SQL expertise of pretrained $\text{LLM}_\theta$, we propose adapting pretrained $\text{LLM}_\theta$ to generate SQL from the input sequences by tuning the model with Text-to-SQL datasets (Wei et al., 2021). The training data contain a collection of serialized inputs $X$ & corresponding SQL outputs $Y$ pairs, sampled from the Text-to-SQL distribution $d_{train}$. The training objective is based on maximizing the log probability of co-appearance of the training data $(X, Y)$:

$$\max_\theta \mathbb{E}_{(X,Y)\sim d_{\text{train}}} \log P_{\text{LLM}_\theta}(Y|X), \tag{11}$$

where $X = f(S, K_P, K_F, H, Q)$ [14] are the serialized inputs as discussed in Sec. 6.1. The optimal $\theta^*$ can be obtained by tuning of LLMs with conventional language modeling objectives:

$$\theta^* = \arg\max_\theta \sum_{(X,Y)} \log P_{\text{LLM}_\theta}(Y|X). \tag{12}$$

#### 6.3.2 Diversifying tuning data coverage

Tuning LLMs would require sufficient amount of data given their large model size (Kaplan et al., 2020). This section focuses on enhancing model tuning through the use of diverse datasets. The rationale is that diverse datasets provide a more comprehensive coverage of SQL knowledge to enrich LLMs. However, a notable challenge of using more than one datasets is that they can be very different, which can lead to a decline in performance due to distribution shifts. This means that a model trained on a particular dataset may not perform as well on another datasets – a common issue for machine learning even for the pre-LLM era. Concretely, Text-to-SQL datasets vary significantly from different perspectives: (i) they encompass a wide range of topics, e.g. from healthcare, retail, and finance etc., each requiring specific knowledge; (ii) the clarify of the database schema varies, with some containing straightforward table or column names, while others are vague and demand further exploration or additional descriptions; (iii) the sizes of dataset range widely, setting different challenges on schema linkage challenges; and (iv) the quality of database content values might contain variations with some datasets having columns filled with NULL data that need filtering, while others not needing that. Given such diversity, it remains an open question how combining multiple training datasets improves tuning performance. Towards this end, we extend Eq. 11 as:

$$\max_\theta \mathbb{E}_{(X,Y)\sim d_{\text{mix}}} \log P_{\text{LLM}_\theta}(Y|X), \tag{13}$$

where $d_{\text{mix}}$ is a mixture of $|d|$ datasets: $d_{\text{mix}} = \{d_i\}_{i=1:|d|}$ and $X = f(S, K_P, K_F, H, Q)$. We investigate whether the pretrained LLM, when tuned with a wide range of inputs from various datasets, can learn to understand these diverse inputs presented in different datasets rather than overfitting to specific patterns, and thus generalize better.

---

[12] An example is using CASE statements and regular expressions

[13] An example is conditional expression (e.g., CASE, IFF, PARTITION clauses) and WINDOW functions.

[14] We discuss incorporation of database values $V$ and column descriptions *des* in Sec. 6.3.4 to illustrate the effect of the key factors one at a time.

### 6.3.3 Augmentation with synthetic data

As previously described, introducing diverse datasets can improve tuning. We further extend this by using diverse synthetic SQL data to augment real datasets, especially considering the high cost to obtain real-world data. Since LLMs are pretrained with massive datasets, their prior knowledge can be utilized to create new information and augment training via synthetic SQL data.

For the same natural language questions, there are usually multiple SQLs that are correct with the same execution outputs[15]. Utilizing this, we focus on synthesizing data to incorporate the multiple ground truth SQLs. We start from a Text-to-SQL dataset, for each (natural language question, SQL)-pair in the dataset, we keep the database schema and natural language question unchanged, and generate new SQLs that are correct but different from the ground truth SQLs. To achieve this, we query LLM with carefully-crafted prompts which include database schema, natural language question, and the ground truth SQL, and request LLMs to generate a SQL that is different from the ground truth and to output a similarity score. The similarity score indicates how similar the candidate is from the true SQL. Details of prompt design $F^s$ are explained in Sec. A.5.1 in Appendix. Given the database $D$, question $Q$, and original ground truth query $SQL^*$, we query the $LLM$ to generate a different SQL output, and estimate its similarity from $SQL^*$, formulated as:

$$(SQL^{(S)}, similarity^{(S)}) = LLM^o(F^s(D, Q, SQL^*)) \tag{14}$$

where $SQL^{(S)}$ is the generated SQL output, $similarity^{(S)}$ is the similarity score outputted simultaneously with SQL output and $F^s$ is synthesis prompting design. $LLM^o$ can be any LLMs, but ideally different from the LLMs that are used for tuning so that they can introduce new information.

Synthetic data generation comes with two challenges: accuracy and diversity. Accuracy refers to the generated SQLs are correct SQL for the natural language query. Diversity refers to the generated SQLs bring new information that is different from original data. To ensure the generated SQL outputs are correct, after the generation we evaluate the SQL [16] and keep the correct one. To ensure the diversity, we only keep the SQL with similarity score below a threshold.

After generating synthetic data, we augment real datasets with them, and the training objective becomes

$$\max_\theta \mathbb{E}_{(X,Y) \sim d_{\mathrm{mix+synthetic}}} \log P_{\mathrm{LLM}_\theta}(Y|X), \tag{15}$$

For this approach, we only synthesize target SQL without synthesizing new natural language questions or databases, because we want to ensure that the generated SQL accuracy high[17]. We leave for more open-ended synthetic data generation methods (e.g. altering questions, SQL, or data schema, etc) for future work.

### 6.3.4 Integration of query-specific database content

Incorporating database content can be crucial for Text-to-SQL performance, particularly when natural language questions refer to specific data values different from table or column names [18]. In some scenarios, without access to the database content, it would be infeasible to formulate accurate SQL queries based solely on the database schema and question, even for humans[19]. Furthermore, when multiple column names seem relevant to a question, database values containing the keywords from the question can help identify the appropriate columns[20]. Overall, access to database content can be critical for improving Text-to-SQL.

---

[15]For example, SQL1 = "... order by score DESC LIMIT 1" and SQL2 = "... WHERE score=max(score)" are equivalent

[16]The result of the generated SQL match the result of the ground truth SQL

[17]For example, modifying new natural language questions of database schema lead to the situation where original ground truth SQL cannot be used to evaluate synthetic SQLs, as the natural questions have been changed

[18]For example, a user might ask, "What is the population in Santa Clara?" However, if the database only has an entry for "Santa Clara County," the correct SQL query should be SELECT .. WHERE county = "Santa Clara County", not WHERE county = "Santa Clara".

[19]As they would not know exact format used in the database.

[20]For example, if the question is "Please list schools in Fresno County Office of Education?", and the database has columns like "District", "District Name", and "dname", knowing that only "District Name" has database values "Fresno County Office of Education" indicates that this "District Name" is the column to use.

Database content is often much larger, compared to the input sequence (database schema, the question, and etc). It is not preferable to include all database values to the inputs, because firstly, total values could go beyond the limitation of the input length. Secondly, the valuable information, such as database schema, can be lost in massive irrelevant values. To address this challenge, we propose including only a limited number of entries $V(Q)$ that are directly relevant to the question.

We first describe the process of extracting the relevant database values relevant to the question, inspired by (Lin et al., 2020). Suppose a natural language question $Q$ is broken into words, and each word is considered as a key word: $Q = [w_1, w_2, ..w_{|Q|}]$. For each keyword $w_i$, we conduct a matching against all entries $(v_{rj}^{(k)})$ in each attribute (column) across all tables, and select the ones above the pre-defined threshold $\delta$. To prevent the inclusion of extraneous irrelevant database content, we limit the selection to be $top_K$ values:

$$V(w_i) = \{v_{rj}^{(k)}| \quad \mathbb{1}[F_m(w_i, v_{rj}^{(k)}) > \delta], \tag{16}$$

$$\forall k = 1 : nT; \forall r = 1 : n_{row}^{(k)}; \forall j = 1 : n_{col}^{(k)}\}[: top_K], \tag{17}$$

where $v_{rj}^{(k)}$ is database content value of $r$th-row $j$th-column entry of $k$th-table and $F_m$ is the matching algorithm. Here, we use $F_m$ as the longest contiguous matching subsequence approach (Cormen et al., 2022), as it allows us to accurately extract the exact values stored in the database. This precision is particularly important for some SQL queries[21]. Finally, the total matches for the entire question $Q$ is determined by aggregating all the matches of individual keywords:

$$V(Q) = \{V(w_i)|i = 1 : |Q|\}. \tag{18}$$

Additionally, $F_m$ can also be instantiated using LLM inference (querying LLM with prompt and asking "whether $w_i$ and $v_{rj}^{(k)}$ match") or embedding similarity (e.g. the cosine distance between the embedding of $w_i$ and $v_{rj}^{(k)}$ above a threshold is a match). We choose to use the above proposed fuzzy string matching approach as it is cost-effective and fast. We leave more investigations on $F_m$ to future work.

With the proposed strategy of selectively including relevant database content, we propose training with question-specific database content via jointly modeling of the conditional probability:

$$\max_{\theta} \mathbb{E}_{(X,Y)} \log \left[ P_{\text{LLM}_{\theta}}(Y|X, V(Q))P(V(Q)|D, Q) \right] \tag{19}$$

with the serialized input $X = f(S, K_P, K_F, H, Q)$. See a demo example in Fig. 4 (*basic prompt + "[Database values that related with questions]"*)[22] and a real example in Sec. A.11 in Appendix. $V(Q)$ is database content relevant to $Q$. To enable effective training, we break down Eq. (19) into two stages. First, extracting relevant database content associated with the natural language question $V(Q)$. Second, with the extracted database content, the LLMs are trained to generate output SQL programs $Y$ using both the input sequence $X$ and the relevant database content. This approach tailors the training process to better reflect realistic scenarios when LLMs consider specific database entries relevant to the user's query. Consequently, the training objective is formulated as:

$$\theta' = \arg \max_{\theta} \sum_{(X,Y)} \log P_{\text{LLM}_{\theta}} \left[ (Y|X, V(Q)) \right]. \tag{20}$$

---

[21] For example, being able to distinguish subtle differences "Santa Clara" vs "Santa Clara County"; "apple" vs "apples"

[22] For databases with column names without parentheses like those in Spider dataset Yu et al. (2018), to incorporate database content into $X$, we append the identified database values to their respective column names in the input sequence's schema, separated by a delimiter "()". This approach aligns with (Qi et al., 2022; Xie et al., 2022). For instance, the database schema is represented as $\overline{T_1 : C_1(V_1), C_2, C_3(V_3) \ldots}$, and it indicates that only columns $C_1$ and $C_3$ contain relevant database content, while $C_2$ does not. For datasets with column names that include parentheses, like those in BIRD dataset (Li et al., 2023d), we add the relevant database content separately, clearly specifying the values corresponding to which columns in particular tables, to avoid confusion caused by the delimiter "()".

### 6.3.5 Table and column selection

Handling real-world datasets poses a significant challenge to Text-to-SQL due to the large number of tables and columns. This challenges can arise when including the entire schema within the prompt limit is infeasible. Even when the schema fits, the increased number of columns adds complexity to the reasoning problem – LLMs struggle in scenarios resembling "finding a needle in a haystack" (Liu et al., 2023b). Thus, careful selection of relevant tables and columns is crucial for improving Text-to-SQL performance (Lei et al., 2020a).

Column selection is to select a subset of columns that are relevant for a given natural language question, so column selection $Z_{sel}(Q)$ is a function of question $Q$. To model Text-to-SQL with column selection, we formulate the problem as modeling the joint probability of the conditional probability of column selection $Z_{sel}(Q)$ given the input sequence $X$, and the conditional probability of generated SQL program $Y$ given the input sequence X and column selection $Z_{sel}(Q)$:

$$\max_{\theta,\beta} \mathbb{E}_{(X,Y)} \log P_{\text{LLM}_\theta}\big(Y|X, Z_{sel}(Q)\big) P_\beta\big(Z_{sel}(Q)|X\big), \tag{21}$$

where $\beta$ represents the parameters for the column selection model. Due to the prohibitive computational challenges of joint modeling of LLMs, we instead digest the objective into two steps: first, modeling the selection of columns; and then, integrating these selected columns into the Text-to-SQL modeling process.

We start from inferring column selection. Text-to-SQL datasets typically do not explicitly provide the ground truth for the relevant columns $Z_{sel}^*(Q)$. We propose extraction of this information from the true SQL queries $Y^*$. The selected columns are the columns that are referenced in the true SQL query. Concretely, for the $k$-th table in the database, the selected columns are represented as:

$$Z_{sel}^{*(k)}(Q) = \{C_j^{(k)} \in Y^* \mid C_j^{(k)} \in S_T^{(k)}, 1 \leq j \leq n_{col}^{(k)}\}. \tag{22}$$

where $C_j^{(k)}$ is $j$-th column of $k$-th table and $S_T^{(k)}$ is database schema. For the entire database schema, we aggregate the column selection of individual tables:

$$Z_{sel}^*(Q) = \bigcup_{k=1}^{n_T} Z_{sel}^{(k)}(Q). \tag{23}$$

Similarly, the selected tables are identified based on their presence in the ground-truth SQL.

$$W_{sel}^*(Q) = \{T_N^{(k)} \in Y^* | T_N^{(k)} \in S_T^{(k)}, 1 \leq K \leq n_T\} \tag{24}$$

where $T_N^{(k)}$ is table name of $k$-th table. We consider two approaches for column selection:

**Retrieval-based column selection**: Retrieval-augmented generation has proven to be an effective and efficient method for handling large contexts in generative tasks. We employ a similar approach for the Text-to-SQL task, utilizing a schema retriever based on nearest neighbor search. Given a natural language query, we identify the closest columns in the semantic space. We opt to retrieve columns instead of tables to achieve a more refined and granular selection. Once the columns are identified, we group them according to their respective tables to construct the selected schema. The retrieval corpus is defined as the union of all table columns. The method initiates by calculating embedding representations for both the query and columns using a pretrained embedding model denoted as $E$. Specifically, we represent the query embedding as $Q_E = E(Q)$ and the embedding of $j$-th column as $C_{Ej}^{(k)} = E(C_j^{(k)})$. Subsequently, the column selection score $s_j^{(k)}$ is defined as the cosine similarity between the query and the column vector embedding. The $top_K$ columns closest to the query are then obtained based on this score:

$$s_j^{(k)} = \text{CosineSimilarity}(Q_E, C_{Ej}^{(k)}) = \frac{Q_E \cdot C_{Ej}^{(k)}}{\|Q_E\|\|C_{Ej}^{(k)}\|}. \tag{25}$$

To generate column embeddings, we construct a sentence for each column by combining diverse pieces of information, including the column name, column type, column description, table name, and a set of the most

common distinct values. This text serves as the input to the embedding model. The specific templates and illustrative examples are presented in Appendix A.6.1.

Retrieval-based approach can be parallelized with tensor operations to efficiently scale to a large number of tables and columns with low cost and latency. It also offers controllability via the hyper-parameter $top_K$, which can adjust recall. It can effectively reduce the risk of false negatives.

**Program-aided column selection**: Solving problems with LLM using coding representation has demonstrated impressive results on a variety of tasks (Gao et al., 2023b; Mishra et al., 2023). This is because coding offers greater precision than natural language descriptions and it bypasses the ambiguities inherent in natural language. Program-aided column selection is an approach to infer column selection using preliminary SQLs. Specifically, we use initial LLMs (denoted as $\mathrm{LLM_{pre}}$) to generate a preliminary SQL query $\hat{Y}$.

$$\hat{Y} = \mathrm{LLM_{pre}}(X). \tag{26}$$

Following the procedure used to infer ground truth column selection from true SQL (Eq 23), we generate column selection from the preliminary SQL $\hat{Y}$. The inferred column selection is determined as:

$$Z_{gen}^*(Q) = \bigcup_{k=1}^{n_T} Z_{gen}^{(k)}(Q) = \bigcup_{k=1}^{n_T} \{C_j^{(k)} \in \hat{Y} \mid 1 \leq j \leq n_{col}^{(k)}\}. \tag{27}$$

To ease the matching process, both the preliminary SQL and schema are normalized to lowercase. In practice, we pinpoint "selected tables" by identifying elements in the SQL following "FROM" or "JOIN" keywords that match table names in the schema. "Selected columns" are then identified by locating column names that appear both in the SQL and the schema of the chosen tables. See the output of program-aided column selection in Fig. 5.

---

**Program-aided column selection**

**Preliminary SQL generation**:

```
SELECT 'FRPM Count (K-12)'/'Enrollment (K-12)' FROM
 frpm WHERE 'County Name'='Alameda' ORDER BY (CAST('
FRPM Count (K-12)' AS REAL) / 'Enrollment (K-12)')
DESC LIMIT 1
```

**The selected table name**:

```
frpm
```

**The selected column name**:

```
FRPM Count (K-12), Enrollment (K-12), County Name
```

Figure 5: Program-aided column selection.

---

The initial models used to generate preliminary SQLs can be standard Text-to-SQL framework to achieve better performance, or some *less capable* models due to various constraints. For example, when dealing with large datasets with many columns, in some scenarios, the prompt length limit can only fit column names, leaving no space for auxiliary information, such as data types and database content, or descriptions. Using these limited inputs (only the schema), we can generate preliminary SQL queries for column selection. Then on the selected columns, we can apply complete prompt (schema plus auxiliary information) to obtain more accurate SQL. Another scenario involves prioritizing the reduction of computational costs and latency, where a smaller initial language model may be employed. Although preliminary SQLs generated from the initial model may not be highly accurate, they can be effective for generating column selection because column selection requires less details than Text-to-SQL task.

Program-aided column selection has the following advantages: The number of columns selected by this approach is low as there are limited number of columns referenced in preliminary SQL queries. So this often leads to high precise of column selection. Additionally and importantly, program-aided column selection fosters a mutually reinforcing cycle – enhanced SQL accuracy improves column selection efficacy, which, in turn, increases the accuracy of future SQL queries. This iterative enhancement process can lead to progressively higher levels of accuracy.

**Integration of column selection to Text-to-SQL pipeline:** We explore two approaches to incorporate column selection:

- **Soft column selection** is the approach where, rather than removing the unselected columns from the database schema $S$, we emphasize the selected columns by adding their column descriptions to the prompt:

$$X = f(S, K_P, K_F, H, Des[Z_{sel}(Q)], V(Q), Q). \tag{28}$$

Soft column selection can be considered as a way to effectively incorporate column descriptions of relevant columns. This method's advantage lies in its resilience to errors in column selection; such errors have minimal impact on the Text-to-SQL task since the model continues to have access to the entire database schema in the prompt. See a example of inputs in Sec. A.11.2 in Appendix.

- **Hard column selection** refers to that scenario that in the database schema $S$, only the chosen columns are included, while the non-selected columns are omitted:

$$X = f(S[Z_{sel}(Q)], K_P, K_F, H, V(Q), Q). \tag{29}$$

This approach has the advantage of considerably shortening the length of the data schema by removing irrelevant columns, which in turn increases the chance that LLMs concentrate on more critical information. Additionally, it also acts as an essential preprocessing step that facilitates the application of Text-to-SQL when the prompt length is insufficient to accommodate the full schema. However, inaccuracies in selecting columns can lead to certain errors in the Text-to-SQL task.

Column selection can be utilized in both few-shot prompting and tuning setups. For prompting, we integrating column selection into the prompt and follow the procedures as outlined in Sec. 6.2; For tuning, column selection is applied to the inputs and follows procedures in Sec. 6.3.1.

### 6.3.6 Test-time refinement via execution-based selection

In previous sections (from Section 6.3.1 to Section 6.3.5), we have outlined various training paradigms. Each section focuses on a unique facet of Text-to-SQL, resulting in the generated SQL that exhibit distinct advantages. Through empirical analysis, we observe that these produced SQL have diverse accuracy coverage – the questions correctly answered by one training paradigm often differ substantially from those addressed by others. This diversity suggests that selecting the appropriate SQL can be a viable strategy for integration of multiple training paradigms. To this end, we introduce an approach called *test-time refinement via execution-based selection* to identify the correct SQL at test time by analyzing execution outcomes. A fundamental advantage of Text-to-SQL is SQL programs are executable. If a SQL program leads to an invalid execution, such as error messages, the SQL can immediately be deemed incorrect. However, a valid execution outcome does not guarantee the SQL is correct. To identify correct SQL, we execute the generated SQL outputs for each question across multiple training paradigms and select the SQL that, while producing valid results, has the execution outcomes supported by the majority of the paradigms. This approach is detailed in Algorithm 1, providing a systematic method for integrating multiple training paradigms to improve SQL query generation accuracy. Similar with execution-based consistency decoding for prompting approach in Sec. 6.2, we consider majority of the execution outcome as a judgement for good SQL. The difference between the two is the candidates in Sec. 6.2 come from sampling from the same setup multiple times, whereas here candidates are come from different training paradigms.

An alternative approach involves combining different input configurations in previous sections into a single training experiment. This method entails integrating various factors, such as mixed training data, synthetic data, database content, and column selection, into the inputs for a single experiment. However, unfortunately, the results of such experiment reveal that merging these components does not result in performance improvements over using them individually. This suggests that LLMs may struggle to effectively process and understand all the provided information simultaneously during tuning.

## 7 Experimental Setup

### 7.1 Tasks and datasets

We consider publicly-available large-scale Text-to-SQL benchmarks. **Spider** (Yu et al., 2018) contains 7000 training samples across 166 databases and 1034 evaluation samples ('Dev split') across 20 databases from a variety of domains. **Spider-SYN** (Gan et al., 2021a) is a complex variant of the Spider dev split, created through the manual replacement of synonym substitutions in natural language queries. **Spider-realistic**

---

**Algorithm 1** Test-time refinement via execution-based selection

---

1: **Input:** Database $D$. $N$ number of questions. $\{SQL_i\}^p$ from $P$ training paradigms. SQL executor $\mathcal{E}$ .
2: **Output:** $outputs = []$
3: **for** $i = 1$ **to** $N$ **do**
4:     **for** $j = 1$ **to** $P$ **do**
5:         executions $= []$
6:         indexes $= []$
7:         e $= \mathcal{E}(SQL_i^j$, D$)$
8:         **if** e $==$ valid **then**
9:             executions $\leftarrow$ e
10:            indexes $\leftarrow$ j
11:         **end if**
12:     **end for**
13:     $outputs \leftarrow \{SQL_i^j | \mathcal{E}(SQL_i^j, D) = \arg\max_e (\text{counts}(\text{executions})), j \in \text{indexes}\}$ ▷ Among SQLs without execution error, select the SQL that gives execution output with maximum number of occurrences.
14: **end for**

---

(Deng et al., 2020) samples 508 text-SQL pairs from Spider dev split removing explicit mentions of column names in natural language queries. **Spider-DK** (Gan et al., 2021b) samples 535 question-SQL pairs on 10 databases from Spider dev split and incorporates domain knowledge to them. **BIRD** (Li et al., 2023d) is a comprehensive dataset containing 9428 question-SQL pairs for train split and 1534 pairs for dev split, across 95 databases totalling a size of 33.4 GB. It covers a broad range of over 37 domains, including finance, sports, healthcare, and education. Uniquely, BIRD incorporates four types of external knowledge sources (numeric reasoning, domain-specific information, synonyms, and value illustration) to enhance the accuracy of SQL query generation. Compared with Spider, BIRD SQLs are typically more complex because of longer SQL, more keywords, more JOINs, and so on. BIRD also contains more challenging databases – more database entries and larger number of tables and columns. Statistics of the number of tables and columns of BIRD are shown in Appendix A.7. Note that BIRD datasets remove the errors on September of 2023, however we conducted all our experiments on previous BIRD version before the error correction. Our performance could be higher with the latest version.

## 7.2 Models

**PaLM-2** is a Transformer-based model trained using a mixture of objectives similar to UL2 (Tay et al., 2022), which is an improved version of its predecessor PaLM (Chowdhery et al., 2022) by efficiently applying compute-optimal scaling, improved training dataset mixture, improved model architecture and objective. The PaLM-2 used here is a Unicorn variant fine-tuned on a collection of improved datasets mixture phrased as instructions following (Wei et al., 2021; Chung et al., 2022).

## 7.3 Experiments

For few-shot prompting, we use Spider datasets. For each question, we sample PaLM-2 32 times with temperature of 0.5. The inputs of the model includes database schema, data type, primary keys, foreign keys, database content, and the question. For fine-tuning, we choose more challenging dataset BIRD. The inputs are described in each experiment. We train until convergence, and the number of steps is no more than 10K steps.

**LoRA finetuning**: Following Hu et al. (2021), we incorporate trainable linear low-rank modules into the query and value projections of each self-attention layer. We set the rank of LoRA to 32, learning rate to 1e-4, and the model architecture to Gecko PaLM model.

### 7.4 Baselines

We list several relevant baseline methods in this section. **Fine-tuning baselines**: **PICARD** (Scholak et al., 2021) employs incremental parsing to constrain auto-regressive decoding. **RASAT** (Qi et al., 2022) is a transformer model that integrates relation-aware self-attention and constrained auto-regressive decoders. **RESDSQL** (Li et al., 2023a) decouples schema linking and skeleton parsing using a ranking-enhanced encoding and skeleton-aware decoding framework. **In-context learning baselines**: (Rajkumar et al., 2022) comprehensively evaluates the Text-to-SQL ability of CodeX and GPT3, while (Liu et al., 2023a) conducts a thorough evaluation on ChatGPT. **DIN** (Pourreza & Rafiei, 2023) decomposes the Text-to-SQL tasks into sub-tasks: schema linking, query classification and decomposition, SQL generation, and self-correction; then perform few-shot prompting with GPT-4. DIN only provides test-suite (TS) evaluation results, so we run execution accuracy (EX) evaluation with their provided SQL outputs. **Self-debugging** (Chen et al., 2023) appends error messages to the prompt and performance multiple rounds of few-shot prompting to self-correct the errors. Self-debugging only reports execution accuracy (EX). **DAIL-SQL** (Gao et al., 2023a) provides a systematic investigation on prompt designs, including question representation and example selection on few-shot prompting and fine-tuning paradigm.

### 7.5 Evaluation

**Text-to-SQL evaluation:** We consider the two commonly-used evaluation metrics: *execution accuracy (EX)* and *test-suite accuracy (TS)* (Zhong et al., 2020). EX consists of one test – measuring whether SQL execution outcome matches that of ground-truth. TS consists of multiple EX tests – measuring whether the SQL passes all of the EX tests, generated by augmentation of the database. Since TS requires passing of more tests, we consider TS as a more reliable evaluation metric. Note that *exact match evaluation* is not performed, as multiple correct SQLs exist for single query. For Spider dataset, we follow the official evaluation protocol of Spider[23]. For BIRD dataset, we follow BIRD official evaluation[24]. BIRD does not have augmentation of test datasets, so BIRD does not have TS evaluation. Additionally, we provide *Valid Efficiency Score (VES)* (Li et al., 2023c), which is designed to measure the efficiency of valid SQLs generated by models for BIRD dataset. VES metric considers both the efficiency and accuracy of execution results.

**Column selection evaluation:** To evaluate the accuracy for retrieval of columns and tables, we report recall, precision, and F1. We compute these metrics and report the averaged metrics across all samples. *recall* is the proportion of relevant columns correctly identified, e.g. identified relevant columns/true relevant columns, whereas *precision* is the proportion of the selected columns that is relevant, e.g. identified relevant columns / identified columns. Finally, $F_1$ is defined as $(2 \cdot precision \cdot recall)/(precision + recall)$.

## 8 Results

We present the performance of our proposed framework, *SQL-PaLM*, in both few-shot prompting and tuning settings. For few-shot prompting, we focus on the Spider benchmark, which is recognized for its high-quality assessments, including both *"execution accuracy"* and *"test suite accuracy"*. For tuning, we focus on the BIRD benchmark, known for its complex SQL and sophisticated database schema, to better differentiate various methods. Both benchmarks are assessed on their respective dev split, which are publicly accessible, in contrast to their private test split[25]. For intermediate results or ablation studies, we select representative and high-performing methods as baselines for comparison. BIRD and SPIDER prompt design are the same format.

---

[23]https://yale-lily.github.io/spider

[24]https://bird-bench.github.io/

[25]These test split are only available through evaluation servers hosted by the benchmarks' creators (refer to (Yu et al., 2018) and (Li et al., 2023d) for more details).

### 8.1 Few-shot prompting setting

#### 8.1.1 Ablation studies

Table 1 shows the efficacy of *Few-shot SQL-PaLM* on the Spider dev set. We utilize the concise prompt design with four demonstrations due to the better performance compared against other prompt designs (Appendix Sec. A.1). We conduct ablation studies showing the roles of execution-based consistency decoding and error filtering played in enhancing model performance. The results indicate omitting either component results in a performance degradation of 4.9% and 3.5% respectively, highlighting the substantial contributions to the overall performance.

Table 1: **Few-shot prompting on Spider dev split**. We present both execution accuracy (EX) and test suite accuracy (TS). Ablation studies are provided on removing execution-based consistency decoding or error filtering respectively.

|  | **Execution accuracy (EX)** | **Test-suite accuracy (TS)** |
|---|---|---|
| *Few-shot SQL-PaLM* | 82.7% | 77.3% |
| **Ablation scenarios** | | |
| - Execution-based consistency | 77.3% | 72.4% ($\downarrow$ 4.9%) |
| - Error filtering | 79.0% | 73.8% ($\downarrow$ 3.5%) |

#### 8.1.2 Performance for different SQL difficulty levels

In our analysis, we evaluate the efficacy of *SQL-PaLM* against a spectrum of SQL difficulty levels, which are categorized based on several factors, including the number of SQL keywords used, the presence of nested subqueries, and the application of column selections or aggregations. The results in Table 2 highlight *SQL-PaLM* performance in comparison with standard few-shot prompting approach using GPT-4 and CodeX-Davinci, as well as the advanced prompting approach DIN-SQL (Pourreza & Rafiei, 2023). Our findings reveal that *SQL-PaLM* consistently surpasses the alternative approaches across all evaluated difficulty levels.

Table 2: **Test-suite accuracy on Spider dev split with SQL outputs being categorized by difficulty levels**. The first four rows are taken from (Pourreza & Rafiei, 2023), and specifically the first two rows are based on standard few-shot prompting.

| Methods | Model | Easy | Medium | Hard | Extra Hard | All |
|---|---|---|---|---|---|---|
| **Few-shot** | CodeX-davinci | 84.7% | 67.3% | 47.1% | 26.5% | 61.5% |
| **Few-shot** | GPT-4 | 86.7% | 73.1% | 59.2% | 31.9% | 67.4% |
| **DIN-SQL** | CodeX-davinci | 89.1% | 75.6% | 58.0% | 38.6% | 69.9% |
| **DIN-SQL** | GPT-4 | 91.1% | 79.8% | **64.9%** | 43.4% | 74.2% |
| *Few-shot SQL-PaLM* | PaLM2 | **93.5%** | **84.8%** | 62.6% | **48.2%** | **77.3%** |

#### 8.1.3 Robustness evaluations

The Text-to-SQL models frequently encounter challenges in robustness, such as translating questions into SQL queries when the terminology differs from the database schema or when specialized domain knowledge is required. To address these, variants of the Spider dataset have been created, as detailed in the Table 30 in Appendix. These include the *"Spider-Syn"* and *"Spider-Realistic"* variants, which alter natural language queries by substituting direct schema references with synonyms or by excluding explicit mentions altogether, respectively. Additionally, the *"Spider-DK"* variant incorporates domain-specific knowledge into the schema. To determine if *Few-shot SQL-PaLM* is capable of overcoming such robustness challenges, we evaluate its performance on these Spider variants.

In Table 3, we compare *Few-shot SQL-PaLM* with previous Text-to-SQL methods. Among these, methods such as T5-3B + PICARD (Scholak et al., 2021), RASAT + PICARD (Qi et al., 2022), and RESDSQL-3B

+ NatSQL (Li et al., 2023a) rely on tuning-based strategies[26]. In contrast, ChatGPT (Liu et al., 2023a) and *SQL-PaLM* utilize few-shot prompting methods without further training. While LLMs naturally have the ability to perform reasoning, including understanding synonyms through extensive pretraining, prior evaluations with ChatGPT (Liu et al., 2023a) have demonstrated significantly lower effectiveness compared to the training-based methods. This is attributed to the generation challenge of Text-to-SQL. However, *Few-shot SQL-PaLM* which also adopts a few-shot prompting strategy, has shown to achieve results on par with the best-performing training-based method (RESDSQL-3B + NatSQL), consistently outperforming other approaches. This outcome highlights the potential of *Few-shot SQL-PaLM* in addressing the robustness challenges.

Table 3: **Evaluation of *Few-shot SQL-PaLM* on Spider variants: Spider-Syn, Spider-Realistic and Spider-DK**. Spider-DK does not contain augmented tests so test suite accuracy is not available.

| Methods/Datasets | Spider-Syn | | Spider-Realistic | | Spider-DK | |
|---|---|---|---|---|---|---|
| | EX | TS | EX | TS | EX | TS |
| T5-3B + PICARD (Scholak et al., 2021) | 69.8 | 61.8 | 71.4 | 61.7 | 62.5 | - |
| RASAT + PICARD (Qi et al., 2022) | 70.7 | 62.4 | 71.9 | 62.6 | 63.9 | - |
| RESDSQL-3B + NatSQL (Li et al., 2023a) | **76.9** | 66.8 | **81.9** | 70.1 | 66.0 | - |
| ChatGPT (OpenAI default Prompt) (Liu et al., 2023a) | 58.6 | 48.5 | 63.4 | 49.2 | 62.6 | |
| *Few-shot SQL-Palm* (Ours) | 74.6 | **67.4** | 77.6 | **72.4** | **66.5** | - |

### 8.1.4 Improving few-shot prompting with column-selection

Table 4 demonstrates improved performance of *SQL-PaLM* on BIRD with column-selection enhanced few-shot prompting, which applies the soft column selection approach (Sec. 6.3.5) to the few-shot prompting (Sec. 6.2). We use the BIRD dataset instead of Spider, as it has larger database schema, where column selection can yield larger impact. We opt for the verbose prompt in our experiments due to its superior performance (Table A.2 in Appendix). The results show that compared with few-shot prompting baseline[27]. The proposed approach, column-selection enhanced few-shot prompting, improves performance $\sim 2\%$. To further understand the potential, we also investigate the upper-bond of the proposed method, where we apply the ground truth column selection. In this setup, we observe an improvement of $\sim 5.7\%$, which provides a motivation for further improving column selection performance for better Text-to-SQL performance.

Table 4: **Evaluations of column-selection enhanced prompting on BIRD dev split.**

| Methods | EX |
|---|---|
| *Few-shot SQL-PaLM* | 43.02% |
| + Soft-column selection (inferred) based description | 45.05%($\uparrow 2.03\%$) |
| + Soft-column selection (GT) based description | 48.70%($\uparrow 5.68\%$) |

## 8.2 Tuning settings

In this section, we present results for exploring the effect of various training paradigms that influence tuning performance of LLMs. We use the following experiments to answer questions proposed in Sec. 4.

### 8.2.1 Performance comparisons with few-shot prompting

*"In what scenarios the improvements are observed to be more significant?"*
On more challenging datasets.

---

[26]For instance, fine-tuning a T5 model

[27]The preliminary SQLs used in column-selection enhanced few-shot prompting is the baseline shown in the first line of Table 4. The input sequences for all the experiments in this table are formed of database schema, data type, primary keys, foreign keys, and the natural language question. Database content is not included.

We first explore the improvements with tuning compared to few-shot prompting. Tables 5 and 6 show the comparisons on BIRD and Spider. For both, tuning demonstrates superior results, highlighting the LLMs proficiency to adapt to high-quality Text-to-SQL training data. Notably, tuning yields a larger improvement on BIRD, ($\sim 8.5\%$), compared to Spider ($\sim 1\%$). This suggests that the benefits of tuning become increasingly important in more complex Text-to-SQL tasks. Given the significant improvements observed on BIRD, we conduct our tuning investigations primarily on it.

Table 5: **Evaluations on BIRD dev split with few-shot prompting and tuning.**

| Adaptation approach | EX |
|---|---|
| Few-shot prompting | 45.05% |
| Tuning | 53.59% (↑ 8.51%) |

Table 6: **Evaluations on Spider dev split with few-shot prompting and tuning.**

| Adaptation approach | EX | TS |
|---|---|---|
| Few-shot prompting | 82.7% | 77.3 % |
| Tuning | 82.8% | 78.2 % (↑ 0.9%) |

### 8.2.2 Scaling model size and different foundation models

*"How about tuning with different foundation models: PaLM-2 vs LLaMA?*
*Does foundation models' properties, such as parametric knowledge, matter?"*
Yes, stronger models help with tuning.

Another investigation is whether tuning performance increases with the model size. Table 7 shows results for tuning PaLM-2 Gecko versus PaLM-2 Unicorn model on BIRD dev split after training on BIRD train split[28]. Despite limited training samples, the larger model has a significant improvement.

Table 7: **Execution accuracy on BIRD dev split using PaLM models of different sizes.**

| Model size | Gecko | Unicorn |
|---|---|---|
| EX | 33.96% | 55.8% |

Table 31 in Appendix A.4 shows the results with tuning open-source models LLaMA7B, LLaMA13B, and LLaMA33B on Spider using the best input representation as reported in Gao et al. (2023a)[29]. Compared with PaLM-2 tuning results in Table 6, LLaMA's fine-tuning results are about 10% lower, that is attributed mainly to the capability of base foundation models. Overall, despite the tuning involving updating parameters, foundation models with larger sizes and better reasoning abilities are observed to be beneficial for tuning Text-to-SQL.

### 8.2.3 Comparisons with parameter efficient tuning

*"How is Text-to-SQL performance with parameter efficient tuning compared with full supervised tuning (SFT)?"*
*"Since train data is limited, is LoRA better than SFT?"*
SFT is observed to be better even with limited tuning data.

Next question is whether tuning benefits from parameter efficient tuning such as LoRA, as we do not have significant amount of training data. Table 8 presents results on tuning a PaLM-2 Gecko using full supervised tuning versus LoRA. The results reveal that full model tuning has clear advantages over LoRA even in the

---

[28]The inputs of the two are the same. Input sequence including database content
[29]Gao et al. (2023a) mainly reports tuning results on SPIDER datasets for LLaMA, not on BIRD

limited data regimes that we have considered with Spider and BIRD, suggesting that the customization for improved Text-to-SQL can benefit from more learnable parameters.

Table 8: **Evaluation of BIRD dev Split, PaLM-2 Gecko**

| Model method | Full Supervised Fine-tuning | LoRA |
|---|---|---|
| EX | 33.96% | 15.84% |

### 8.2.4 The impact of training data diversity and generalization

*"What kinds of tuning data might be more helpful?"*
Complex SQLs are observed to be more useful.

Text-to-SQL benchmarks can be quite different from each other. We explore whether training on more datasets, despite of the diversity[30], can help with tuning performance. We tune LLMs on combination of Spider and BIRD datasets, and evaluate their performance on each. Table 9 shows that when evaluating on the BIRD dev split, a model trained on both BIRD and Spider outperforms a model trained solely on BIRD. Similarly, Table 10 illustrates the improved performance on the Spider dev split when trained on both datasets, compared to training only on Spider. The results suggest that tuning LLMs on various datasets benefits tuning performance, and the model after tuning is more robust to distribution shifts, indicating the tuned models are not over-fitting on train dataset. BIRD contains more complex SQL queries compared to Spider. We observe a more significant performance improvement on Spider when BIRD data are incorporated, compared to the improvement seen on BIRD when Spider data are incorporated. This implies that introducing complex SQL queries into training can yield larger benefits compared with less complex SQL samples. BIRD and SPIDER prompt design are the same format.

Table 9: **Evaluations on BIRD Dev Split with different training data used for tuning.**

| Train data | Execution accuracy |
|---|---|
| BIRD Only | 53.59% |
| BIRD + Spider | 55.15 (↑ 1.56%) |

Table 10: **Evaluations on Spider Dev Split with different training data used for tuning.**

| Train data | Execution accuracy | Test-suite accuracy |
|---|---|---|
| Spider Only | 82.8% | 78.2 % |
| BIRD + Spider | 86.8% (↑ 4%) | 82.8 (↑ 3.5%) |

### 8.2.5 Incorporating database content

*"How does the database content help tuning?"*
It clarifies mismatches in questions and database.

We explore whether introducing question-specific database content benefits tuning performance. Table 11 presents the improvements achieved by incorporating database content[31]. The results indicate more than 3% accuracy improvement when testing on the BIRD dev split, highlighting the positive impact of incorporating database content.

We further provide two case studies to illustrate why database content would help. One scenario arises when there is disparity between the words used in natural language query and the words used in the database, as exemplified in Fig. 6. As another example, Fig. 7 illustrates how database content can serve as valuable

---

[30]Other than the difference in SQLs or database. The provided information can be different. BIRD has hints, column descriptions, etc; Spider has none of them

[31]We train on both BIRD and Spider datasets, as they bring good performance described in Sec. 8.2.4

Table 11: **Evaluations on BIRD Dev Split with and without database content.**

| Train data | EX |
|---|---|
| Without database content | 55.15% |
| With database content | 58.80% (↑ 3.65%) |

cues for LLMs to identify relevant columns when multiple columns seem relevant for the question. This is a situation when both LLMs and human experts have a difficult time to deciding which columns to use. The database values presented in Figs. 6 and 7 encompass all the keywords in the questions, not limited to the specific keywords discussed in this context.

---

**Case Study 1**

**Question:**
What is the highest eligible free rate for K-12 students in the schools in **Alameda County**?
**True SQL:**
**SELECT** 'FRPM Count (K-12)' / 'Enrollment (K-12)'
**FROM** frpm **WHERE** 'County Name' = **'Alameda'**
**ORDER BY** (**CAST**('FRPM Count (K-12)' **AS REAL**) / 'Enrollment (K-12)') **DESC LIMIT** 1;

**Without database content**

**Inferred SQL:**
**SELECT** 'FRPM Count (K-12)' / 'Enrollment (K-12)'
**FROM** frpm **WHERE** 'County Name' = **'Alameda County'**
**ORDER BY** (**CAST**('FRPM Count (K-12)' **AS REAL**) / 'Enrollment (K-12)') **DESC LIMIT 1**;
**Error reason:**
Question has "Alameda County", whereas database has values "Alameda" (no "County")

**With database content**

**Extracted database content values:** {table: {column: [matched values]}}
**Table** 'frpm':
**'County Name': ['Alameda']**,
**Table** 'satscores':
'cname': ['Alameda'],
**Table** 'schools':
'AdmFName1': ['Rae'],
'AdmLName1': ['Free'],
'City': ['Alameda'],
'County': ['Alameda'],
'GSoffered': ['K-12'],
'GSserved': ['K-12'],
'MailCity': ['Alameda'],
**Inferred SQL:**
**SELECT** 'FRPM Count (K-12)' / 'Enrollment (K-12)'
**FROM** frpm **WHERE** 'County Name' = **'Alameda'**
**ORDER BY** (**CAST**('FRPM Count (K-12)' **AS REAL**) / 'Enrollment (K-12)') **DESC LIMIT 1**;

---

Figure 6: **Case study 1**: Terminology used in the question is different from that saved in the database. Consider the instance of the keyword "Alameda County" found in the natural language query: "What is the highest eligible free rate for K-12 students in the schools in Alameda County?" While the question uses "Alameda County", in the database, this information is stored as "Alameda" without the word "County". If LLMs have access only to the original question without database content, the resulted SQL query is likely to contain the same keywords from the natural language, leading to an incorrect answer. Following Sec. 6.3.4, we extract database content that is relevant to the question and the output is presented in Fig. 6 (e.g. the column 'County Name' containing 'Alameda').

## 8.3 Improving tuning with synthetic data

We present the impact of synthetic data augmentation on the Text-to-SQL task. Table 12 presents the results of the inclusion of synthetic data into the original training data (Spider and BIRD), which leads to the performance increase of 1.3% on the BIRD dev split. This improvement underscores the effectiveness of synthetic data in enhancing overall performance.

| Case Study 2 |
|---|
| **Question:** Please list the zip code of all the charter schools in **Fresno County Office of Education**. |
| **True SQL:** |
| **SELECT** T2.Zip **FROM** frpm **AS** T1 **INNER JOIN** schools **AS** T2 **ON** T1.CDSCode = T2.CDSCode |
| **WHERE T1.****'District Name'** = **'Fresno County Office of Education'AND** T1.'Charter School (Y/N)' = 1 |
| **Without database content** |
| **Inferred SQL**: |
| **SELECT** T1.Zip **FROM** schools **AS** T1 **INNER JOIN** frpm **AS** T2 **ON** T1.CDsCode = T2.CDsCode |
| **WHERE T2.****'County Name'** = **'Fresno County Office of Education' AND** T2.'Charter School (Y/N)' = 1 |
| **Error Reason**: |
| Multiple columns may contain keywords ( 'Fresno County Office of Education'); |
| LLMs don't know which columns to use ('District Name' vs 'County Name') |
| **With database content** |
| **Extracted database content values:** {table: {column: [matched values]}} |
| **Table** 'frpm': |
| 'County Name': ['Fresno'], |
| **'District Name' : ['Fresno County Office of Education']**, |
| 'District Type': ['County Office of Education (COE)'], |
| **Table** 'satscores': |
| 'cname': ['Fresno'], |
| 'dname': ['Fresno County Office of Education'], |
| **Table** 'schools': |
| 'AdmLName1': ['Coe'], |
| 'City': ['Fresno'], |
| 'County': ['Fresno'], |
| 'DOCType': ['County Office of Education (COE)'], |
| 'District': ['Fresno County Office of Education', 'Colusa County Office of Education'], |
| 'MailCity': ['Fresno'], |
| **Inferred SQL**: |
| **SELECT** T2.Zip **FROM** frpm **AS** T1 **INNER JOIN** schools **AS** T2 **ON** T1.CDSCode = T2.CDSCode |
| **WHERE T1.****'District Name'** = **'Fresno County Office of Education'AND** T1.'Charter School (Y/N)' = 1 |

Figure 7: **Case study 2:** The association of keywords with a specific column is unclear without database content. Consider the keyword "Fresno County Office of Education" in the question "Please list the zip code of all the charter schools in Fresno County Office of Education.". Without utilizing the database content, the LLMs might erroneously select the wrong column, such as "County name." However, with the inclusion of database content (assuming the column "District Name" contains the relevant keywords 'Fresno County Office of Education'), the LLMs learn 'District Name' is the columns to use.

Table 12: Evaluations on BIRD Dev Split showing the impact of extra LLM-generated synthetic data.

| Tuning data | Execution accuracy |
|---|---|
| Spider + BIRD | 55.15% |
| Spider + BIRD + LLM-generated synthetic data | 56.45% (↑ 1.3%) |

To encourage generation of a correct, distinct queries, we prompt the LLM to generate up to three queries, followed by removing SQL that fails official evaluation or with a similarity score (Eq. 6.3.3) greater than 0.9. We choose to augment BIRD datasets, instead of Spider, because Spider contains simpler queries than BIRD, resulting in reduced flexibility in generating diverse queries from the original SQL. We use GPT-4 in synthetic generation, as selecting the LLMs for synthetic data differently from the LLMs used in tuning potentially can bring new information.

We further examine the generated SQLs and their similarity score. Most of the generated SQL outputs do not deviate significantly from ground truth, as indicated by the similarity score distribution (see statistics shown in Fig. 10 and Table 33 in Appendix). Among these generated SQL outputs, 81.4% are correct, validated by official evaluation. After removing similar SQLs (with similarity score > 0.9), 78.8% of the generated queries remains for training, which are considered as both diverse and precise, (Table 32 in Appendix). A few examples of the LLM generated synthetic SQL rewrites are provided in Table 13. It can be observed that the LLM performs well at the given task by diversifying the ground truth queries and augmenting the dataset in a useful way.

Table 13: **Examples of BIRD dataset queries generated.**

| Synthetic Data Examples | |
|---|---|
| **Example 1:** | |
| **Question** | Please name any three restaurants that have an unidentified region. |
| **Ground-truth** | SELECT T2.label FROM location AS T1 INNER JOIN generalinfo AS T2 ON T1.id_restaurant = T2.id_restaurant INNER JOIN geographic AS T3 ON T2.city = T3.city WHERE T3.region = 'unknown' LIMIT 3 |
| **Generated** | |
| **Query-1** | SELECT gi.label FROM generalinfo gi, geographic g WHERE gi.city = g.city AND g.region = 'unknown' LIMIT 3 |
| **Query-2** | SELECT label FROM generalinfo WHERE id_restaurant IN (SELECT id_restaurant FROM location WHERE city IN (SELECT city FROM geographic WHERE region = 'unknown')) LIMIT 3 |
| **Query-3** | SELECT label FROM generalinfo WHERE city IN (SELECT city FROM geographic WHERE region = 'unknown') LIMIT 3 |
| **Comment** | The generated queries 1, 2, 3 have similarities 0.8, 0.7, 0.6 respectively. In this example, the LLM has also identified redundant table usage from the ground truth and removed it from the generated query. |
| **Example 2:** | |
| **Question** | Please give all the list prices of the product LL Fork. |
| **Ground-truth** | SELECT T2.ListPrice FROM Product AS T1 INNER JOIN ProductListPriceHistory AS T2 ON T1.ProductID = T2.ProductID WHERE T1.Name = 'LL Fork' |
| **Generated** | |
| **Query-1** | SELECT ProductListPriceHistory.ListPrice FROM Product JOIN ProductListPriceHistory ON Product.ProductID = ProductListPriceHistory.ProductID WHERE Product.Name = 'LL Fork' |
| **Query-2** | SELECT plph.ListPrice FROM Product p, ProductListPriceHistory plph WHERE p.ProductID = plph.ProductID AND p.Name = 'LL Fork') LIMIT 3 |
| **Query-3** | SELECT ListPrice FROM ProductListPriceHistory WHERE ProductID IN (SELECT ProductID FROM Product WHERE Name = 'LL Fork') |
| **Comment** | The generated queries 1, 2, 3 have similarities 0.95, 0.85, 0.75 respectively. Query-1 is merely an alias change from the original query and hence queries with higher similarity (closer to 1) are not useful for augmenting the dataset. |

## 8.4 Tuning with table and column selection

Table & column selection plays an important role for Text-to-SQL scalability and accuracy, as covered in Sec. 6.3.5 – for database schema with high number of columns, it becomes vital to distill them down to a pertinent subset for Text-to-SQL especially when the schema size exceeds the LLMs' prompt length limit. This process is also crucial even for database schemas that can be represented within prompt limit, as it facilitates LLMs to focus on important information.

Regarding the selection of columns, a high *"recall"* rate (the proportion of relevant columns correctly identified) - is crucial, to ensure that no crucial columns are missed. With *"recall"* meets a satisfactory level, we also aim to enhance the *"precision"* - the proportion of the selected columns that is relevant, since it is beneficial to exclude numerous irrelevant columns.

In this section, we discuss the outcomes of using retrieval-based and program-aided column selection techniques. We begin by evaluating the accuracy of column selection achieved by each method, then assess how these approaches influence the overall effectiveness of Text-to-SQL conversions. Lastly, we explore the advantages and limitations associated with both strategies.

**Retrieval-based column selection**: Table 14 shows the performance of retrieval-based column selection, with top 10 and 25 columns selected based on the retrieval ranking scores. The recall rates for selecting the top 10 and 25 columns are notably high at 81.52% and 92.93% respectively. Nonetheless, the precision is comparatively lower, suggesting that while retrieval-based column selection has a good coverage of true columns, it also incorporates superfluous columns.

Table 15 presents the performance of end-to-end Text-to-SQL on the BIRD dev split, using both soft and hard column selection. The soft column selection method surpasses the baseline performance, which lacks column selection, demonstrating the effectiveness of soft column selection. On the other hand, hard column selection yields results worse than the baseline. This decrease in performance is likely due to hard column selection's incorrect exclusion of relevant columns from the database schema. For instance, with the top 10 selection, approximately 20% of columns $(1 - 81.52\%)$ are missing from the inputs, while soft column selection is more effective retaining the entire schema information while also enriching the selected columns with additional column description information.

Table 14: **Accuracy (%) of retrieval-based column selection when retrieving top 10 and 25 candidates on BIRD dev split.**

|  | Table selection | | Column selection | |
|---|---|---|---|---|
|  | **Top 10** | **Top 25** | **Top 10** | **Top 25** |
| Average counts | 3.39 | 5.4 | 10 | 25 |
| **Recall** | 90.11 | 97.08 | 81.52 | 92.93 |
| Precision | 56.34 | 40.48 | 26.96 | 16.00 |
| F1 | 38.95 | 54.05 | 38.95 | 26.39 |

Table 15: **Execution accuracy of Text-to-SQL on BIRD dev split with retrieval-based column selection incorporated into inputs.** Baseline is from Table 11. Experiments are using the same setups as baseline.

|  | **Baseline** | **Top 10** | **Top 25** |
|---|---|---|---|
| Baseline | 58.8% | - | - |
| + Hard column selection | - | 52.48% | 56.39% |
| + Soft column selection | - | 58.93% | **59.13%** |

**Program-aided column selection**: Table 16 presents the efficacy of the program-aided approach for column and table selection. Overall, we observe impressive performance of this approach for selecting relevant columns, with recall, precision, and F1 are all more than 90%. In comparison to the retrieval-based column selection, the program-aided column selection has a substantially higher precision, indicating fewer irrelevant columns are selected. Additionally, program-aided method depends on preliminary SQL – more accurate preliminary SQL prediction leads to more precise column and table choices, as detailed in Table 34 in the Appendix.

Table 16: **Accuracy (%) with program-aided column selection**. The preliminary SQL used for the program-aided approach is the baseline SQL from Table 11 which has an accuracy of 58.8%.

|  | Table selection | Column selection |
|---|---|---|
| Average count | 1.88 | 5.24 |
| **Recall** | 94.64 | 90.66 |
| Precision | 96.14 | 92.60 |
| F1 | 95.39 | 91.62 |

Table 17 illustrates the comprehensive impact of different ways to integrate chosen columns into model fine-tuning. We present two approaches: *hard* column selection and *soft* column selection, to distinguish including only selected schema versus full schema into prompt. The way to identify the selected column is "program-aided" approach, as it gives optimal results. Hard column selection yields inferior outcomes compared to the baseline, likely because its recall rate is 90%—a seemingly high number that nonetheless results in the omission of 10% of columns. Conversely, soft column selection avoids this shortcoming and outperforms the baseline.

Table 17: **Execution accuracy of Text-to-SQL on BIRD dev split with program-aided column selection incorporated into inputs**. Baseline is from Table 11. Experiments use the same setups as baseline.

|  | **EX** |
|---|---|
| Baseline | 58.80% |
| + Hard column selection | 57.95% |
| + Soft column selection | **59.19%** |

Furthermore, we evaluate our column selection approaches against alternative baseline approaches, such as using LLMs to directly identify relevant tables and columns through prompting (e.g. "What are the relevant tables or columns?") and other existing methods. Our methods demonstrably surpass such alternatives, as detailed in Table 35 within the Appendix.

**Ablation studies and the upper bond:** What is the upper bound on the achievable performance incorporating the soft column selection approach? We conduct an ablation study using ground-truth column selection for soft column selection, which serves as an upper bound of this approach.

Table 18 presents an ablation study on the *soft* column selection approach. We present two ways to identify the selected columns: "retrieval-based" (3rd line) and "program-aided" (4th line). Additionally, the table includes two further ablation studies: one on the impact of incorporating full column descriptions (2nd line) and another on using ground truth column selections (5th line). As Table 18 suggests, with ground truth column selection applied as the oracle, the performance of Text-to-SQL reaches to 62.06%, which is $\sim 3\%$ above the results using inferred column selection, indicating the potential of improving column selection.

Additionally, since soft column selection incorporates the column description of a subset of the columns, what would the performance be if we incorporate the descriptions of all columns (2nd line)? We conduct an ablation study to include full column descriptions (See an input example A.11.1). For the prompts that exceed the input length, we cut them to fit the input length. The results reveal that introducing full column descriptions

Table 18: **Ablation studies on soft column selection.**

| Method | EX |
|---|---|
| Baseline | 58.80% |
| + Full column descriptions | 54.69%($\downarrow$ 4.11%) |
| + Soft column selection (retrieval-based) | 59.13% |
| + Soft column selection (program-aided) | 59.19% |
| + Ground truth column selection (oracle) | 62.06%($\uparrow$ 3.26%) |

actually decreases the performance compared with baseline by 4.11%. The reason might be due to full column descriptions yielding too lengthy inputs that distract LLM from focusing on important information such as database schema and important information might get truncated.

**Comparing retrieval-based and program-aided column selection:** The program-aided approach outperforms retrieval-based approach in the accuracy of column selection, evidenced by its high F1 score and precision (Table 14 and Table 16). It achieves comparable recall with on average of 5 selected columns, unlike the retrieval-based approach that uses 25 to achieve 90% recall, demonstrating its efficiency. However, the overall Text-to-SQL performance of retrieval-based and program-aided column selection methods are comparable. Despite the program-aided approach showing significantly better performance in column selection, its end-to-end performance does not reflect a similar improvement. This seeming contradiction can be explained with the recall of the program-aided method being comparable to the top 25 retrieval approach (90 vs 92) and the recall being more critical for Text-to-SQL performance. Additionally, there is a mismatch between accuracy of column selection on train and dev splits. The program-aided approach, which trains an initial model to produce preliminary SQL outputs, results in higher accuracy in column selection of the train split compared to the dev split. This accuracy disparity could hinder the program-aided method from achieving its highest potential performance.

The retrieval-based method stands out for its computational efficiency and cost savings, as it avoids the need for querying expensive LLMs. This approach also allows for easy adjustment of recall by modifying the number of retrieved candidates (*"topK"*). Furthermore, in cases of extremely large datasets where the schema size makes generating even preliminary SQL infeasible due to prompt length constraints, the retrieval-based approach is the only practical solution. While this scenario might not occur in academic benchmarks, it is a common challenge in real-world applications.

**Comparing hard and soft column selection**:
Compared with soft column selection, hard column selection leads to performance reduction. The reduction, 3% for retrieval-based approach (top 25) and 1% for program-aided approach, might be a reasonable trade-off when we consider the amount of input information that has been reduced and the cost has been saved.

From column level, BIRD dev split contains on average of 76 columns (Table 37 in Appendix), and program-aided approach selects on average 5.24 columns, therefore, there are only $5.24/76 = 6.8\%$ of the total columns used for program-aided column selection at the cost of 1% of accuracy loss. Similarly, for retrieval-based approach, there are only 33% of total columns are used in inputs at the cost of 3% of accuracy. From token level, Table 19 demonstrates that the numbers of token after hard column selection are for only 26% or 42% compared to the full prompt for the two approaches. This suggests an equivalent proportion of cost savings, given that the cost associated with LLMs is directly proportional to the number of tokens. In some scenario, one may want to sacrifice some performance for the cost reduction.

Table 19: **The number of input token saved due to hard column selection.**

|  | Tokens (counts) | Tokens remained (percentage (%)) |
|---|---|---|
| Baseline | 1085.66 | 100% |
| Program-aided | 286.31 | 26.37% |
| Retrieval-based | 454.97 | 41.91% |

**Impact of the size of the database schema:**
Fig. 8 shows how the performance of the end-to-end Text-to-SQL process changes with different numbers of columns in the schema, using a soft column selection approach where the descriptions of selected columns are included in the prompt. The analysis reveals that as the number of columns increases, Text-to-SQL performance declines, indicating that column count is a significant indicator for the difficulty of Text-to-SQL tasks. The program-aided column selection method performs better than others when the column count is below 90, whereas the retrieval-based approach excels when the column count exceeds 90. On average, the program-aided method selects about 5.24 columns per question, whereas the retrieval-based method extracts 25. When the total number of columns is below 90, including descriptions of 25 columns can overwhelm the prompt, leading to poorer performance. However, when the total number of columns is above 90, including 25 columns does not take a significant proportion of the schema, making the retrieval-based approach more effective.

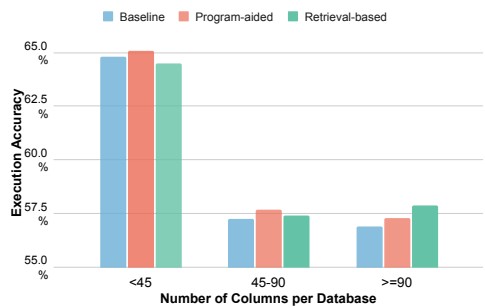

Figure 8: Text-to-SQL performance (y-axis) with respect to column numbers (x-axis). Both retrieval-based and program-aided are based on soft-column selection.

## 8.5 Improvements with test-time refinement via execution-based selection

Table 20 presents the effectiveness of the test-time refinement via execution-based selection approach, as discussed in Section 6.3.6. Test-time refinement via execution-based selection approach integrates multiple predefined training paradigms introduced in previous sections, including mixing of training data, the integration of database content, the use of synthetic data, and the implementation of both hard and soft column selection strategies. This combination method results in a performance improvement of 2.9% over individual training paradigms, such as the results in Table 11.

To assess the robustness of the test time execution selection to distribution shifts, we also apply the model, originally tailored for BIRD, to the Spider dataset. Despite Spider's distinct format differences from BIRD, such as lacking of column descriptions and hints or not employing column selection, Table 21 reveals that the method still enhances performance on a different dataset, indicating its robustness to format variations.

Alternatively, we also explore another approach of combining different training paradigms into a single paradigm. This involves integrating various training components directly into the inputs of one single experiment. This method entails integrating various elements, such as mixed training data, synthetic data, database content, and column selection, into the inputs for a single experiment. Yet, as outlined in Table 38 in the Appendix, this strategy does not yield additional accuracy gain, suggesting combining multiple information at once does not have superimposed positive effects. The reason may be LLMs cannot understand multiple information in the inputs simultaneously effectively, highlighting the inherent difficulties of incorporating various components of inputs.

Table 20: **Execution of text-time refinement via execution-based selection on BIRD dev split.**

| Decoding approach | Execution accuracy |
|---|---|
| Baseline | 58.8% |
| + Test-time execution-based selection | 61.7% (↑ 2.9%) |

Table 21: **Evaluations of text-time refinement via execution-based selection on Spider dev split.**

| Train data | Execution accuracy | Test-suite accuracy |
|---|---|---|
| Baseline | 86.8% | 82.8% |
| Baseline | 87.3% (↑ 0.5%) | 83.5% (↑ 0.7%) |

## 8.6 Combining all constituents in *SQL-PaLM*

We consolidate our findings, as illustrated in Table 22. Our experimentation involves applying standard instruction tuning for an LLM which significantly outperforms in-context learning techniques. Utilizing combined training data BIRD and Spider for tuning led to an improvement of 2%. Furthermore, the integration of synthetically generated data contributes to an additional 1% boost. Incorporating the database content results in a 3% increase, and further incorporating soft column selection into intputs adds another 1%. Additionally, the implementation of test time execution-based selection, which combines the aforementioned training paradigms, provides an additional 3% improvement. Additionally, we present the results of execution-based test-time selection on different difficulty levels in Table 23.

Table 22: **Summary of each component's contribution in Text-to-SQL performance.**

| Run | Type | Train data | Method | Execution accuracy |
|---|---|---|---|---|
| 1 | Tuning | BIRD | | 53.00% |
| 2 | Tuning | BIRD + Spider | | 55.15% |
| 3 | Tuning | BIRD + Spider + Synthetic data | | 56.45% |
| 4 | Tuning | BIRD + Spider | + Database content | 58.80% |
| 5 | Tuning | BIRD + Spider + Synthetic data | + Database content | 58.35% |
| 6 | Tuning | BIRD + Spider | + Database content + Soft column selection (Retrieval-based) | 59.13% |
| 7 | Tuning | BIRD + Spider | + Database content + Soft column selection (Program-aided) | 59.19% |
| 9 | Post-Tuning | - | Execution-based test-time selection | 61.70% |

Table 23: **Execution accuracy of execution-based test-time adaptation algorithm across various SQL complexity levels on the BIRD dev split.**

| | Simple | Moderate | Challenging | Total |
|---|---|---|---|---|
| Count | 933 | 459 | 142 | 1534 |
| Execution-based test-time selection | 68.92% | 52.07% | 47.89% | 61.93% |

### 8.6.1 Overall Text-to-SQL performance comparison with other methods

Putting everything together, *SQL-PaLM* demonstrates strong results on both the Spider and BIRD datasets. We present a comparative analysis of our methodology against leading methods from the BIRD (Table 24) and Spider (Table 25) leaderboards. *SQL-PaLM* has achieved notable results, with execution accuracy of 87.3% on Spider dev split and 61.7% on BIRD dev split. These improvements are made possible by effectively utilizing diverse input components for tuning efficiently and adopting a selective execution approach. Additionally, BIRD provides Valid Efficiency Score (VES) to measure efficiency of valid SQL. Table 24 indicates *SQL-PaLM* significantly better performance for VES (10%+), indicates the *SQL-PaLM* can output efficient SQL.

In Table 24 and Table 25, we compare our approach with a variety of different methods for BIRD and SPIDER, respectively, since the top methods on the different leaderboards vary. The fact that our single method is competitive across different benchmarks and against diverse sets of methods further attests to the efficacy of our strategy.

Note that some leaderboard submissions are by anonymous contributors, lacking detailed documentation or code, and hindering a full comparison on dev splits (Leaderboard number is for test split, not dev split). In such cases, we use '-' to acknowledge their notable leaderboard performance, despite not being able to consider them for a direct evaluation.

Table 24: **Evaluation on BIRD dev set with top-ranked methods.**

|  | Methods/Datasets | EX | VES |
|---|---|---|---|
| **Tuning** | SFT CodeS-15B | 58.47% | 59.87% |
| **Few-shot prompting** | Codex | 25.42% | 43.41% |
|  | ChatGPT | 37.22% | 43.81% |
|  | GPT-4 | 46.35% | 49.77% |
|  | DIN-SQL + GPT-4 | 50.72% | 58.79% |
|  | DAIL-SQL + GPT-4 | 54.56% | 56.08% |
|  | MAC-SQL + GPT-4 | 59.39% | 58.76 % |
| **Not available** | Dubo-SQL | 59.71% | 66.01% |
|  | MCS-SQL + GPT-4 | - | 64.82% |
|  | *Few-shot SQL-PaLM* (Ours) | 45.5% | 45.11% |
|  | *Fine-tuned SQL-PaLM* (Ours) | **61.7%** | **79.35**% |

Table 25: **Evaluation on SPIDER dev set with top-ranked methods.**

|  | Methods/Datasets | EX | TS |
|---|---|---|---|
| **Fine-tuning** | T5-3B + PICARD | 79.3% | 69.4% |
|  | RASAT + PICARD | 80.5% | 70.3% |
|  | RESDSQL-3B + NatSQL | 84.1% | 73.5% |
| **Few-shot prompting** | CodeX davinci (0-shot) | 67.0% | 55.1% |
|  | CodeX davinci (few-shot) | 71.0% | 61.5% |
|  | ChatGPT | 70.1% | 60.1% |
|  | GPT-4 (Zero-shot) | 72.9% | 64.9% |
|  | GPT-4 (Few-shot) | 76.8% | 67.4% |
|  | Self-Debug | 84.1% | - |
|  | DIN-SQL (w/ CodeX Davinci) | 75.6% | 69.9% |
|  | DIN-SQL (w/ GPT-4) | 82.8% | 74.2% |
|  | DAIL-SQL + GPT-4 + Self-Consistency | 83.6% | 72.8% |
|  | MiniSeek | - | - |
|  | MAC-SQL + GPT4 | 86.75% | - |
|  | *Few-shot SQL-PaLM* (Ours) | 82.7% | 77.3% |
|  | *Fine-tuned SQL-PaLM* (Ours) | **87.3%** | **83.5%** |

### 8.7 Error analyses

**For few-shot prompting on Spider**: In our detailed examination of the SQL queries produced by *SQL-PaLM* via few-shot prompting on the SPIDER dataset, we undertook an in-depth manual evaluation to assess the quality of the generated SQL. Our analysis indicates that the queries often exhibit creativity[32], often deviating from the ground-truth by employing varied SQL clause. These queries are mostly free from syntactical errors and consistently display complex reasoning, such as the capability to join multiple tables. To provide more tangible insights, Table 26 shows two complex yet accurately generated queries. Further discussions and examples are provided in Section A.9.1 in the Appendix.

Table 26: **SQL examples generated by few-shot prompting of *SQL-PaLM*.**

Question: What are the number of concerts that occurred in the stadium with the largest capacity ?
SQL:

```
SELECT count(*) FROM concert AS T1 JOIN stadium AS T2 ON T1.stadium_id = T2.stadium_id
WHERE T2.capacity = ( SELECT max(T3.capacity) FROM stadium AS T3 )
```

Question: What are the id sand names of all countries that either have more than 3 car makers or produce fiat model ?

```
SELECT T1.countryid , T1.countryname FROM countries AS T1 JOIN car_makers AS T2 ON
T1.countryid = T2.country GROUP BY T1.countryid HAVING count(*) > 3 UNION SELECT T1.
countryid , T1.countryname FROM countries AS T1 JOIN car_makers AS T2 ON T1.countryid =
T2.country JOIN model_list AS T3 ON T2.id = T3.maker WHERE T3.model = "fiat"
```

**For tuning on BIRD**: To better understand the common error modes of the fine-tuned LLM, we randomly select 100 queries from different databases in the BirdSQL dev split, where the execution results from generated queries don't match those of the ground-truth results. Table 27 shows a high-level breakdown of accuracy on BIRD, categorized by query difficulty. As one would expect, the accuracy on harder examples is lower – the accuracy on easier examples (68.92%) is significantly higher than that of moderate examples (52.07%) which is significantly higher than the challenging examples (47.89%). Additionally, we manually

Table 27: *SQL-PaLM* error analysis statistics on BIRD dev split

| Category | Number of Queries | Percentage |
|---|---|---|
| Total | 1533 | - |
| Correct | 950 | 61.97% |
| Incorrect | 584 | 38.10% |
| Invalid | 14 | 0.91% |

| Per difficulty | Number of queries | EX |
|---|---|---|
| Total Simple | 933 | 60.86% |
| Total Moderate | 459 | 29.94% |
| Total Challenging | 142 | 9.26% |
| Correct Simple | 290 | 68.92% |
| Correct Moderate | 220 | 52.07% |
| Correct Challenging | 74 | 47.89% |

inspect these queries and categorize them according to the types of errors they produce. The error categories are shown in Figure 9 and will be described below. The representation of each category in the pie plot is proportional to its respective percentage.

We subdivide the errors into several categories: **Schema Linking:** Encompasses queries where the model was not able to select the relevant tables for the queries (e.g. failing to join tables). **Misunderstanding Database Content**: The model fails to accurately interpret the data within the tables

---

[32]It is an evidence of not memorization.

(e.g. assumes an incorrect date format for a specific column). **Misunderstanding Knowledge Evidence:** The model wasn't able to interpret the human-annotated evidence or ignores it altogether.
**Reasoning:** The model fails to comprehend the question and the generated query doesn't contain the necessary reasoning steps to generate the correct queries. **Syntax-Related Errors:** The model produces SQL that are not runnable due to some syntactical mistake (e.g., missing backticks to refer to a column which has spaces). Finally, in red, we label an additional error category, denoted **Dataset related errors**, which encompasses different errors due to questions, schema, evidence, inconsistencies between the ground-truth SQL and the question, not because of the outputted SQL. In the 100 queries that we analyze, 31 of them present this error category that, if extrapolated to the full dataset, would upper bound the performance of BIRD dev set to 70%. We describe more of our finding on data-quality in the Appendix A.9.2 and present more error examples of each category in Table 42 in Appendix.

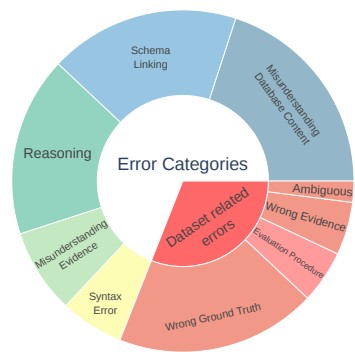

Figure 9: Error categories for *SQL-PaLM* fine-tuned LLM on Bird dev set.

# 9 Conclusions

This paper presents the SQL-PaLM framework, our holistic approach to advancing Text-to-SQL capabilities. We provide insightful discussion for understanding key factors in deciding Text-to-SQL performance. We start with a comprehensive examination of few-shot prompting to enhance Text-to-SQL performance with LLMs. Then we present best practices for instruction fine-tuning, examining how performance can be improved through expanded data coverage and diversity, synthetic data augmentation and integrating query-specific database content. We introduce a test-time refinement approach that leverages query execution feedback to bolster SQL query accuracy. Additionally, we address some of the real-world challenges of navigating complex databases with many tables and columns, presenting effective methods for the precise selection of pertinent database components to improve Text-to-SQL performance. Our integrated approach demonstrates substantial improvements in Text-to-SQL performance, demonstrated on two important public benchmarks.

# 10 Acknowledgement

We thank Sayna Ebrahimi and Slav Petrov for reviewing this paper.

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

# A   Appendix

## A.1   Prompt design and number of demonstrations for Spider

In Table 28, we analyze the performance of *Few-shot SQL-PaLM* method on different number of demonstrations (zero- vs. few-shot) and queries with different prompt designs. Overall, few-shot prompting outperforms zero-shot counterpart. We also explore the effect of different prompt design approaches on performance. For the LLM being queried (e.g. PaLM2), concise prompt

| Prompt design | Adaptation setting | EX | TS |
|---|---|---|---|
| **Concise** | 0-shot | 81.2 | 76.0 |
| **Concise** | 4-shot | **82.7** | **77.3** |
| **Verbose** | 0-shot | 78.5 | 70.9 |
| **Verbose** | 4-shot | 81.3 | 73.7 |

Table 28: Test-suite accuracy for different prompt design approaches in zero- and few-shot set-up on Spider Dev.

is observed to be better. "Verbose" prompts are based on using natural language to describe database schema, which is closer to the way LLMs were trained, whereas "Concise" prompts use the symbols to describe the database schema, which has advantages of clearly presenting table structure. Examples of concise prompt and verbose prompt are provided in Appendix A.10.1 and A.10.2.

## A.2   Prompt design for BIRD via few-shot prompting

In Table 29, we investigate various prompt design on BIRD datasets. Unlike Spider datasets (Table 28) where concise prompt works better, for BIRD datasets verbose prompt works superior.

Table 29: Execution accuracy of Text-to-SQL with different prompt designs across different SQL difficulty levels on BIRD datasets

| | Simple | Moderate | Challenging | Total |
|---|---|---|---|---|
| Count | 933 | 459 | 142 | 1534 |
| **Concise** | 49.95% | 25.05% | 19.01% | 39.63% |
| **Verbose** | 53.27% | 29.85% | 18.31% | 43.02% |

## A.3   Descriptions of Robust Spider Datasets: Spider-SYN, Spider-Realistic, Spider-DK

Table 30: Information on different variants of Spider datasets with the purpose of evaluating robustness.

| | Counts | Modification Category | Source | Modify Natural Question? | Modify Database Schema? | Add New Database Schema? | Examples |
|---|---|---|---|---|---|---|---|
| **Spider-SYN** | 1034 | Manually modifying natural language questions with synonym substitutions | Spider Dev. | Yes | No | No | Spider
# Database Schema: concert_singer
# stadium(Stadium_ID, Location, Name, Capacity, Highest, Lowest, Average)
# singer(Singer_ID, Name, Country, Song_Name, Song_release_year, Age, Is_male)
# concert(concert_ID, concert_Name, Theme, Stadium_ID, Year)
# singer_in_concert(concert_ID, Singer_ID)
#
Q: How many singers do we have?

Spider-SYN
Q: How many vocalists do we have? |
| **Spider-Realistic** | 508 | Modify natural language questions to remove explicitly mentioning column names | Subset of Spider Dev | Yes | No | No | Spider
# Database Schema: concert_singer
Q: How many concerts are there in year 2014 or 2015?

Q: How many concerts are there in 2014 or 2015?
# No year |
| **Spider-DK** | 535 | Modify database schema to incorporate the domain knowledge | Subset of Spider Dev | Yes | Yes | Yes | # Database Schema: concert_singer
Modify database column "Age" into "Birthday";
Replace its values from "52" to "1971-02-09 00:00:00"

Q: List all song names by singers above the average age.
# hard to answer "age"-related question |

### A.4 Tuning performance with different foundation models

Table 31 shows the results of tuning open-source models LLaMA7B, LLaMA13B, and LLaMA33B on Spider using the best input representation as reported in Gao et al. (2023a)[33]

Table 31: **Evaluations on Spider dev split using different foundation models.**

| Foundation model | TS |
|---|---|
| LLaMA-7B | 66.7% |
| LLaMA-2-CHAT-7B | 69.6% |
| LLaMA-13B | 68.6% |
| LLaMA-2-CHAT-13B | 65.1% |
| LLaMA-33B | 69.1% |

### A.5 Synthetic data

### A.5.1 Synthetic data prompt design

---

**Synthetic Data Prompt Design**

You will be provided with a list of tables from a SQL database followed by a natural language query related to the database and the original SQL query answering the question. Your job is to understand the natural language queries and generate up to 3 different SQL queries using diverse commands from the original query while answering the question correctly. You need to make sure to use the same columns from the original query for the generated query. You will also generate a similarity score between the original and the generated query based on how closer they are syntactically.

**Database tables schema are as follows:**

```
CREATE TABLE customers (
  customer_id int,          -- unique customer id
  name varchar(100),        -- name of the customer
  email_address varchar(255), -- email address of the customer
);

CREATE TABLE order (
  order_id int,             -- unique order id.
  customer_id int,          -- unique customer id.
  order_amount decimal(10, 2), -- amount spent by the customer on the order
);
```

**Question:** Find the email of the top spending customer?

**Original SQL query:**

```
SELECT customers.first_name
FROM customers
JOIN order ON customers.customer_id = order.customer_id
GROUP BY customers.customer_id, customers.first_name
ORDER BY SUM(order.order_amount) DESC
LIMIT 1;
```

Output the generated queries and the similarity scores in a json list as follows:

---

[33]The numbers are taken from Gao et al. (2023a);

```
[
  {"sql":        // generated query-1,
   "similarity": // similarity score (0.0-1.0) for query-1
  },
  {...}
]
```

### A.5.2 Synthetic data similarity score distribution

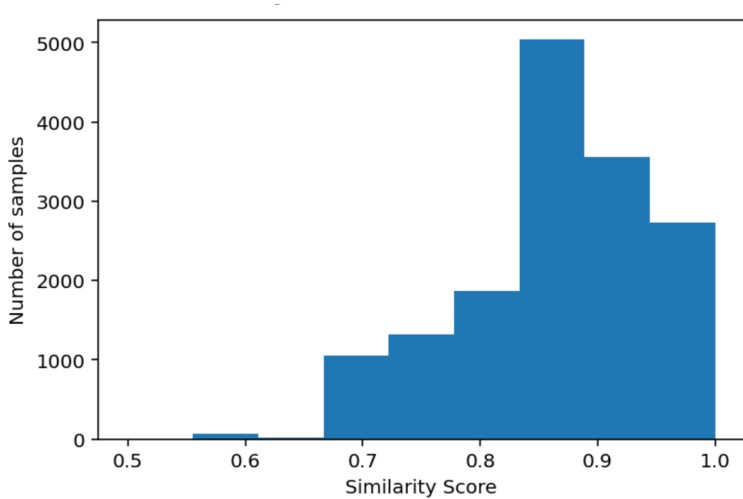

Figure 10: Histogram plot of synthetic data similarity scores

32 displays the statistics of the generated queries with correctness and similarity filters applied. It is important to note that lower similarity score indicates higher diversity.

Table 32: Synthetic data generation statistics for BIRD train split

|  | Correct SQL | Correct SQL + Similarity <= 0.9 |
|---|---|---|
| Samples with 1+ generated queries | 81.4% | 78.8% |

The mean, median, standard deviation of the similarity scores of all generated queries is reported in Table 33 and the histogram is plotted in Figure 10 in Appendix.

Table 33: Statistics of synthetic data similarity scores

| Min | Max | Mean | Median | STD |
|---|---|---|---|---|
| 0.5 | 1.0 | 0.85 | 0.85 | 0.07 |

### A.6 Column Selection

### A.6.1 Example of retrieval-based column selection

*Template:*

Column name [column_name] of type [column_type] from the table [table_name]. Description: [column_description]. Value examples: [common_distinct_values].

*Example:*

> Column name 'size' of type 'STRING' from the table 'package'. Description: 'package size dimensions'. Value examples: 'small', 'medium', 'long'.

### A.6.2 Program-aided column selection ablation studies

The accuracy of program-aided column selection is directly related to the accuracy of the preliminary SQL. The higher the accuracy of these preliminary SQL queries, the better the column selections based on them will be.

Table 34: Higher accuracy in preliminary SQL leads to better column selection

| Accuracy (%) | Table (%) | | | Column (%) | | |
|---|---|---|---|---|---|---|
| Preliminary SQL | Recall | Precision | F1 | Recall | Precision | F1 |
| 43 | 92.96 | 91.15 | 92.04 | 84.75 | 86.62 | 85.67 |
| 50.2 | 94.10 | 94.41 | 94.25 | 87.97 | 90.87 | 89.40 |
| 55 | 93.76 | 95.12 | 94.44 | 89.62 | 91.69 | 90.64 |
| 58.8 | 94.64 | 96.14 | 95.39 | 90.66 | 92.60 | 91.62 |

### A.6.3 Compare with other column selection methods

We presented other column selection methods: **LLM base**: prompting LLMs to request table and column selection (Prompt is in Sec. A.6.4). **LLM CoT**: Following Pourreza & Rafiei (2023), we add few-shot examples and use change-of-thought demonstrations to help the prompt (Prompt is in Sec. A.6.5). The model is PaLM-2 text-bison. **Automatic Annotation** (Lei et al., 2020b) is a proposes pattern matching approach. The last two lines are taken from Table 16 and Table 14 (Top 10), rounded by two decimals. The results in Table 35 indicates that program-aided algorithm outperform the other methods with a clear margin.

Table 35: Comparison of Table and Column selection

| | Table selection | | | Column selection | | |
|---|---|---|---|---|---|---|
| Methods | Recall | Precision | F1 | Recall | Precision | F1 |
| LLM baseline | 0.71 | 0.88 | 0.75 | 0.24 | 0.83 | 0.35 |
| LLM Few-shot | 0.85 | 0.82 | 0.82 | 0.81 | 0.64 | 0.70 |
| Automatic Annotation | 0.87 | 0.74 | 0.80 | 0.68 | 0.90 | 0.78 |
| Retrieval-based (Ours) | 0.90 | 0.56 | 0.39 | 0.82 | 0.27 | 0.39 |
| Program-aided (Ours) | **0.95** | **0.96** | **0.95** | **0.91** | **0.93** | **0.92** |

### A.6.4 LLM base prompt design of column selection

We start with a simple baseline which asks the model to select a schema in two steps. First, select tables and then columns. 1 Table selection

```
Select tables from my database named [database_name], to answer the given query.
   Tables:
    CREATE TABLE [table_name_1] ()
    CREATE TABLE [table_name_2] ()
    ...
   Query: '{query}'
   Only generate a list of comma separated table names without any spaces.
```

2 Column selection

```
Given a table definition and a natural language query,
I am interested in selecting columns related to the query:
```

```
CREATE TABLE [table_name] (
   [column_name] [column_type],
);
Table Values:
[3 row examples with header]
The natural language query: {query}
Select all related column names, including ids, from the table.
Only generate a list of comma separated column names values without any spaces.
```

### A.6.5   LLM Few-shot with CoT for column selection

Following Pourreza & Rafiei (2023), we add few-shot examples and use change-of-thought demonstrations to help the prompt.

```
You are an agent designed to find the schema_links  for generating SQL queries for each
question based on the database schema and Foreign keys.
Hint helps you to find the correct schema_links.
###
Few examples of this task are:
###
Schema of the database with sample rows and column descriptions:
#
CREATE TABLE users (
     user_id INT,
     ...
   );
   Table Values:
   User_id ...
   001 ...
Table users
  User_id: id of the user
  ...
Question: Among the lists created by user 4208563...
Answer: Let's think step by step. In the question , we are asked:
"user" so we need column = [lists_users.user_id]
"number of followers" so we need column = [lists.list_followers]...
Schema_links: [lists.list_followers,lists_users.user_subscriber,
lists.user_id = lists_user.user_id, lists.list_id = lists_user.list_id,
lists_users.user_id, 4208563, 1]

###
Schema of the database with sample rows and column descriptions:
#
CREATE TABLE [table_name] (
     [column_name] [column_type],
     ...
   );
   Table Values:
   [3 row examples with header]
Table [table_name]
  [column_name]: [column_description]
  ...
Question: [question]
```

```
Answer: Let's think step by step.
```

## A.7 Column and table data statistics of BIRD dataset

Table 36 illustrates the distribution of data regarding the number of columns and tables per example in the testing sets. The ground-truth distribution is also provided. Notably, BIRD presents a more challenging scenario compared to Spider, with an average of 73 total columns to be selected per example, of which only 3.7 columns are used in the ground truth. Similarly, the average number of tables is 7, and 1.9 tables are selected in the ground truth. However, it's essential to acknowledge that these sets still deviate from real-world scenarios with thousands of columns. This direction should be further explored in the future.

Table 36: Statistics for table and columns in the data-sets

|  | BIRD Test | | | | | | Spider Test | | | | | |
|---|---|---|---|---|---|---|---|---|---|---|---|---|
|  | Mean | Min. | Max. | P75 | P95 | SDT | Mean | Min. | Max. | P75 | P95 | SDT |
| Tables | 7 | 3 | 13 | 8 | 11.5 | 2.7 | 4 | 2 | 11 | 4 | 8.3 | 2.2 |
| Columns | 73 | 11 | 192 | 91.5 | 155.5 | 48.8 | 20.8 | 7 | 59 | 24 | 50 | 13.5 |
| Columns per table | 10.4 | 2 | 115 | 9 | 42.4 | 16.8 | 5.2 | 2 | 26 | 6 | 11.2 | 3.6 |
| Ground-truth Tables | 1.9 | 1 | 4 | 2 | 3 | 0.7 | 1.5 | 1 | 4 | 2 | 3 | 0.6 |
| Ground-truth columns | 3.7 | 1 | 9 | 5 | 6 | 1.4 | 2.4 | 1 | 6 | 3 | 5 | 1.2 |

Table 37: Average of number of columns and tables for questions for BIRD datasets: we compute the average number of table and columns for each questions.

|  | number of queries | Avg. number of table | Avg. of column counts | Median number of tables | Min. of column counts | Max. of column counts |
|---|---|---|---|---|---|---|
| BIRD-valid | 1534 | 7.4 | 76.3 | 71 | 11 | 201 |
| BIRD-train | 9428 | 12.0 | 77.4 | 48 | 6 | 457 |

## A.8 Exploration on combining submodules

An alternative approach involves combining different input configurations in previous sections into a single training experiment. This method entails integrating various elements, such as mixed training data, synthetic data, database content, and column selection, into the inputs for a single experiment. However, the outcomes of such experiments reveal that merging these components does not result in performance improvements over using them individually. This suggests that LLMs may struggle to effectively process and understand all the provided information simultaneously during tuning.

| Type | Train data | Method | Accuracy |
|---|---|---|---|
| Tuning | BIRD + Spider | + database content | 58.80 |
| Tuning | BIRD + Spider + Synthetic data | + database content | 58.35 |
| Tuning | BIRD + Spider + Synthetic data | + database content + soft column selection | 58.08 |

Table 38: The effect of integrating all components into one training paradigm

## A.9 Case Study of SQL Generation and Error Analysis

### A.9.1 *SQL-PaLM* with few-shot Prompting

We present case studies of *Few-shot SQL-PaLM* in Table 39 and 40 for "correct" and "wrong" SQL generated by *Few-shot SQL-PaLM* based on test-suite accuracy of Spider dataset. Surprisingly, the majority of examples

classified as "errors" by *Few-shot SQL-PaLM* were actually correct when evaluated by human experts, indicating the scores of *SQL-PaLM* might be significantly higher. The evaluation fails due to (1) ambiguous questions, exemplified by 1st example in Table 40, and (2) official test-suite evaluation struggles on evaluating creative solutions with output which deviate from that of the ground-truth. For instance, the 2nd example has multiple valid ground-truths; the 3rd example experiences type-related issues; the 4th example presents different formats (e.g. "full name" and "first name, second name" are equally semantically correct for the question. They both should be considered correct); and the 5th example is false negative due to the omission of the "distinct" keyword.

Regarding the "real" mistakes made by *Few-shot SQL-PaLM*, such as the sixth and seventh examples, we observed a departure from simple errors like syntax errors commonly found in other methods (Liu et al., 2023a). Instead, the mistakes made by *Few-shot SQL-PaLM* closely resemble those that human experts would make when developing the same solution, demonstrating its profound expertise in SQL. Another source of errors is the presence of a "confusing database schema," where *Few-shot SQL-PaLM* encounters difficulties in selecting the appropriate table or column when multiple equivalent options contain similar content (as illustrated in the 5th example of Table 40).

Tables 39 and 40 show the capabilities of *Few-shot SQL-PaLM*, demonstrating that it can efficiently handle complex SQL queries. It successfully deals with tasks such as joining multiple tables using various keywords (as observed in the 1st, 2nd, 4th, and 5th examples in Table 39 and all examples in Table 40), as well as employing nested SQL structures (as seen in the 3rd example of Table 39). Moreover, *Few-shot SQL-PaLM* exhibits the ability to generate creative and diverse SQL outputs that differ from the ground-truth but remain equally correct. This suggests a deep understanding of SQL content rather than mere memorization. Notable examples include the 3rd example in Table 39 and the 2nd, 3rd, 4th, and 5th examples in Table 40. Even in cases of errors, such as the 6th and 7th examples in Table 40, *Few-shot SQL-PaLM* presents alternative solutions distinct from the ground-truth. Furthermore, *Few-shot SQL-PaLM* demonstrates the ability to infer relevant SQL expression based on semantic meaning, i.e. "French singers" and "country=France," as well as "young to old" and "OrderBy age ASC" (as evident in the 1st and 2nd examples). This capability is attributed to the large-scale pretraining of LLMs.

### A.9.2 Fine-tuned *SQL-PaLM*

Next, we present our manual investigation of the generated queries on the BIRD dev set. To quantify the error cases for the queries generated from the fine-tuned *Few-shot SQL-PaLM*, we randomly select 100 samples from BIRD dev set and categorize the queries that we considered mistakes based on the BIRD evaluation procedure.

Table 41 shows a breakdown of the different error types that we have identified and provided some examples in Table 42 for each category. We categorize the false positives as **Wrong Ground-Truth (**19%) which are examples from the dataset that do not correctly answer the original question. **Wrong Evidence (**5%) which denotes examples that have incorrect or misleading human-annotated evidences. **Ambiguous (**2%) which are examples where the question's meaning is ambiguous and open-ended (e.g., not clear what is expected from the question). **Evaluation Procedure (**5% denoting that the generated SQL query was in fact correct but was considered incorrect by the evaluation procedure (e.g., the generated queries contains additional selected columns). For a comprehensive list of examples from the different error types, refer to Table 42.

This investigation demonstrates that from 100 randomly sampled incorrect queries (out of a total of 584 for fine-tuned *Few-shot SQL-PaLM*), 31% of them are not correctly evaluated by the BIRD evaluation suite and if this sample size is representative of the full dev set, it shows that there is a performance upper bound of around 70%.

Table 39: *Few-shot SQL-PaLM* examples that are correct according to test-suite accuracy.

| Data Schema | Question | Ground-truth | SQL-PaLM | Comment |
|---|---|---|---|---|
| Q1: **stadium**: Stadium_ID, Location, Name, Capacity, Highest, Lowest, Average **singer**: Singer_ID, Name, Country, Song_Name, Song_release_year, Age, Is_male **concert**: concert_ID, concert_Name, Theme, Stadium_ID, Year **singer_in_concert**: concert_ID, Singer_ID | What is the average, minimum, and maximum age of all singers from France? | `SELECT avg(age) , min(age) , max(age) FROM singer WHERE country = 'France'` | `SELECT avg(age) , min(age) , max(age) FROM singer WHERE country = "France"` | Inference on the relationship between "French" and "France" |
| | Show name, country, age for all singers ordered by age from the oldest to the youngest. | `SELECT name , country , age FROM singer ORDER BY age DESC` | `SELECT name , country , age FROM singer ORDER BY age DESC` | Inference based on understanding the age ranking via "order by age DESC": |
| Q2: **city**: ID, Name, CountryCode, District, Population **sqlite_sequence**: name, seq **country**: Code, Name, Continent, Region, SurfaceArea, IndepYear, Population, LifeExpectancy, GNP, GNPOld, LocalName, GovernmentForm, HeadOfState, Capital, Code2 **countrylanguage**: CountryCode, Language, IsOfficial, Percentage | What are the names of nations where both English and French are official languages? | `SELECT T1.Name FROM country AS T1 JOIN countrylanguage AS T2 ON T1.Code = T2.CountryCode WHERE T2.Language = "English" AND T2.IsOfficial = "T" INTERSECT SELECT T1.Name FROM country AS T1 JOIN countrylanguage AS T2 ON T1.Code = T2.CountryCode WHERE T2.Language = "French" AND T2.IsOfficial = "T"` | `SELECT T1.name FROM country AS T1 JOIN countrylanguage AS T2 ON T1.code = T2.countrycode WHERE T2.language = "English" AND T2.isofficial = "T" INTERSECT SELECT T1.name FROM country AS T1 JOIN countrylanguage AS T2 ON T1.code = T2.countrycode WHERE T2.language = "French" AND T2.isofficial = "T"` | Generate exact GT |
| Q3: **stadium**: Stadium_ID, Location, Name, Capacity, Highest, Lowest, Average **singer**: Singer_ID, Name, Country, Song_Name, Song_release_year, Age, Is_male **concert**: concert_ID, concert_Name, Theme, Stadium_ID, Year **singer_in_concert**: concert_ID, Singer_ID | What are the number of concerts that occurred in the stadium with the largest capacity ? | `select count(*) from concert where stadium_id = (select stadium_id from stadium order by capacity desc limit 1)` | `SELECT count(*) FROM concert AS T1 JOIN stadium AS T2 ON T1.stadium_id = T2.stadium_id WHERE T2.capacity = ( SELECT max( T3.capacity) FROM stadium AS T3 )` | Creative – different from ground-truth but accurate |
| Q4: **Student**: StuID, LName, Fname, Age, Sex, Major, Advisor, city_code **Has_Pet**: StuID, PetID **Pets**: PetID, PetType, pet_age, weight | Find the first name of students who have both cat and dog pets . | `select t1.fname from student as t1 join has_pet as t2 on t1.stuid = t2.stuid join pets as t3 on t3.petid = t2.petid where t3.pettype = 'cat' intersect select t1.fname from student as t1 join has_pet as t2 on t1.stuid = t2.stuid join pets as t3 on t3.petid = t2.petid where t3.pettype = 'dog'` | `SELECT T1.fname FROM student AS T1 JOIN has_pet AS T2 ON T1.stuid = T2.stuid JOIN pets AS T3 ON T2.petid = T3.petid WHERE T3.pettype = "cat" INTERSECT SELECT T1.fname FROM student AS T1 JOIN has_pet AS T2 ON T1.stuid = T2.stuid JOIN pets AS T3 ON T2.petid = T3.petid WHERE T3.pettype = "dog"` | Complex SQL containing multiple SQL keywords. |
| Q5: **continents**: ContId, Continent **countries**: CountryId, CountryName, Continent **car_makers**: Id, Maker, FullName, Country **model_list**: ModelId, Maker, Model **car_names**: MakeId, Model, Make **cars_data**: Id, MPG, Cylinders, Edispl, Horsepower, Weight, Accelerate, Year | What are the ids and names of all countries that either have more than 3 car makers or produce fiat model ? | `select t1.countryid , t1.countryname from countries as t1 join car_makers as t2 on t1.countryid = t2.country group by t1.countryid having count(*) > 3 union select t1.countryid , t1.countryname from countries as t1 join car_makers as t2 on t1.countryid = t2.country join model_list as t3 on t2.id = t3.maker where t3.model = 'fiat';` | `SELECT T1.countryid , T1.countryname FROM countries AS T1 JOIN car_makers AS T2 ON T1.countryid = T2.country GROUP BY T1.countryid HAVING count(*) > 3 UNION SELECT T1.countryid , T1.countryname FROM countries AS T1 JOIN car_makers AS T2 ON T1.countryid = T2.country JOIN model_list AS T3 ON T2.id = T3.maker WHERE T3.model = "fiat"` | Complex SQL containing multiple SQL keywords |

Table 40: *Few-shot SQL-PaLM* outputs that are incorrect according to test suite evaluation. Indeed, according to human experts, the majority of these "error cases" are actually the correct SQL outputs.

| | |
|---|---|
| **Question 1:** | **What is the maximum capacity and the average of all stadiums ? (id=16, database=concert__singer)** |
| **Ground-truth:** | select max(capacity), average from stadium |
| **SQL-PALM:** | SELECT max(capacity) , avg(capacity) FROM stadium |
| **Comment:** | Ambiguous question. The Ground-truth query would give the **average occupancy column** of each stadium, where as the SQL-PALM one would give the average of the **capacity column** of all stadiums. Without specific context, the SQL-PALM output matches the question better. |
| **Human Expert** | **True (Ground-truth issue)** |
| **Question 2:** | **Which airline has most number of flights? (id=229, database=flight__2)** |
| **Ground-truth:** | SELECT T1.Airline FROM AIRLINES AS T1 JOIN FLIGHTS AS T2 ON T1.uid = T2.Airline GROUP BY T1.Airline ORDER BY count(*) DESC LIMIT 1 |
| **SQL-PALM:** | SELECT T1.airline FROM airlines AS T1 JOIN flights AS T2 ON T1.uid = T2.airline GROUP BY T2.airline ORDER BY count(*) DESC LIMIT 1 |
| **Comment:** | Both T1.Airline (the Airlines.Airline column, which contains names of airlines) and T2.airline (the flights.airline column, which contains the uid of airlines) uniquely identify airlines, and grouping by either one gives the same results. The evaluation failed because the data contains 12 airlines each with 100 flights (i.e. every one is the maximum), and instability of the ordering produces different outcomes when both queries pick one of the maximums. |
| **Human Expert** | **True (Database and Eval issue)** |
| **Question 3:** | **What are the id, name and membership level of visitors who have spent the largest amount of money in total in all museum tickets? (id=420, database=museum__visit)** |
| **Ground-truth:** | SELECT t2.visitor_id , t1.name , t1.Level_of_membership FROM visitor AS t1 JOIN visit AS t2 ON t1.id = t2.visitor_id GROUP BY t2.visitor_id ORDER BY sum(t2.Total_spent) DESC LIMIT 1 |
| **Excution result:** | ('3', 'Arjen Robben', 1), where '3' is a string |
| **SQL-PALM:** | SELECT T1.id , T1.name , T1.level_of_membership FROM visitor AS T1 JOIN visit AS T2 ON T1.id = T2.visitor_id GROUP BY T1.id ORDER BY sum(T2.total_spent) DESC LIMIT 1 |
| **Excution result:** | (3, 'Arjen Robben', 1), where 3 is an integer |
| **Comment:** | The result is actually correct. The highlighted differences t2.visitor_id and t1.id are equivalent because the two tables are joined on t1.id = t2.visitor_id. Evaluation failed because of the field type differences, as t1.id is int and t2.visitor_id is text. |
| **Human Expert** | **True (Database and Eval issue)** |
| **Question 4:** | **List the names of all winners who played in both 2013 and 2016. (id=447, database=wta__1)** |
| **Ground-truth:** | SELECT winner_name FROM matches WHERE YEAR = 2013 INTERSECT SELECT winner_name FROM matches WHERE YEAR = 2016 |
| **Excution result:** | ('Angelique Kerber',), ('Petra Kvitova',) ... |
| **SQL-PALM:** | SELECT T1.first_name , T1.last_name FROM players AS T1 JOIN matches AS T2 ON T1.player_id = T2.winner_id WHERE T2.year = 2013 INTERSECT SELECT T1.first_name , T1.last_name FROM players AS T1 JOIN matches AS T2 ON T1.player_id = T2.winner_id WHERE T2.year = 2016 |
| **Execution result:** | ('Angelique', 'Kerber'), ('Petra', 'Kvitova') ... |
| **Comment:** | The result is actually correct. The highlighted differences, as both execution results make sense from semantic perspective of the query. winner__name and T1.first__name , T1.last__name are equivalent for representation of "name". Evaluation failed because of the differences in the output format type. If multiple ground-truths are provided considering different output formats, this is not an error. |
| **Human Expert** | **True (Eval issue)** |
| **Question 5:** | **What are the different template type codes? (id=322, database=cre__Doc__Template__Mgt):** |
| **Ground-truth:** | SELECT DISTINCT template_type_code FROM Templates |
| **SQL-PALM:** | SELECT DISTINCT template_type_code FROM Ref_Template_Types |
| **Comment:** | The results are actually correct. *Few-shot SQL-PaLM* selects a different table Ref_Template_Types, instead of Templates. The same "template types" appear in multiple entries of Templates. The two SQL outputs generate the same execution result when evaluating with "DISTINCT"". Evaluation failed because Spider official evaluation removes 'DISTINCT' during evaluation. |
| **Human Expert** | **True (Eval issue)** |
| **Question 6:** | **Find the number of professionals who have not treated any dogs. (id=983, database=dog__kennels):** |
| **Ground-truth:** | SELECT count(*) FROM Professionals WHERE professional_id NOT IN ( SELECT professional_id FROM Treatments ) |
| **SQL-PALM:** | SELECT count(*) FROM Professionals EXCEPT SELECT professional_id FROM Treatments |
| **Comment:** | Left and Right sides of "'EXCEPT" need equivalent content. Corrected SQL is SELECT count(*) FROM (SELECT professional_id FROM Professionals EXCEPT SELECT professional_id FROM Treatments) |
| **Human Expert** | **False (Wrong Use of keywords)** |
| **Question 7:** | **Find the number of professionals who have not treated any dogs. (id=754, database=world__1):** |
| **Ground-truth:** | select t1.name from country as t1 join countrylanguage as t2 on t1.code = t2.countrycode where t2.language = "english" and isofficial = "t" union select t1.name from country as t1 join countrylanguage as t2 on t1.code = t2.countrycode where t2.language = "dutch" and isofficial = "t" |
| **SQL-PALM:** | SELECT T1.name FROM country AS T1 JOIN countrylanguage AS T2 ON T1.code = T2.countrycode WHERE T2.language = "English" OR T2.language = "Dutch" AND T2.isofficial = "T" |
| **Comment:** | Operator Precedence: ADD > OR. Need to add parenthesis over "OR". Corrected SQL is SELECT T1.name FROM country AS T1 JOIN countrylanguage AS T2 ON T1.code = T2.countrycode WHERE (T2.language = "English" OR T2.language = "Dutch" ) AND T2.isofficial = "T". Spider evaluation normalizes the ground-truth outputs to all lowercase for easier evaluation, but mismatch exists when referring to database content. Changes:english->English,dutch->Dutch, t->T |
| **Human Expert** | **False (Wrong operator precedence and eval issue)** |

Table 41: BIRD dev set errors from a sample of 100 queries denoted as "incorrect" by the evaluation procedure.

| False-positive category | Number of examples |
|---|---|
| Wrong Ground-Truth | 19 |
| Wrong Evidence | 5 |
| Ambiguous | 2 |
| Evaluation Procedure | 5 |

Table 42: These examples demonstrate the different categories of errors of *SQL-PaLM*-fine-tuned.

**Wrong Ground-Truth**

**Example 1:**

| Question | Write all comments made on the post titled 'How does gentle boosting differ from AdaBoost?' (id=579, database=codebase_community) |
|---|---|
| Ground-truth | `SELECT T1.Text FROM comments AS T1 INNER JOIN posts AS T2 ON T1.PostId = T2.Id WHERE T2.Title = 'How does gentle boosting differ FROM AdaBoost?'.` |
| Comment: | The ground-truth query has an upper case "FROM" instead of "from" which is what is in the question. |

**Example 2:**

| Question | What's the finish time for the driver who ranked second in 2008's Australian Grand Prix? (id=937, database=formula_1) |
|---|---|
| Ground-truth | `SELECT T1.time FROM results AS T1 INNER JOIN races AS T2 on T1.raceId = T2.raceId WHERE T1.rank = 2 AND T2.name = 'Australian GrAND Prix' AND T2.year = 2008` |
| Comment | Similarly to the previous example, the ground-truth query string doesn't match the one from the question. Likely this is due to a data-cleaning procedure in the BirdSQL dev set. |

**Example 3:**

| Question | What race number has the most finishers? (id=979, database=formula_1) |
|---|---|
| Ground-truth | `SELECT raceId FROM results GROUP BY raceId ORDER BY COUNT(time IS NOT NULL) DESC LIMIT 1` |
| Comment | The `COUNT(time IS NOT NULL)` is somewhat unconventional. Typically, `COUNT` is used on a column name directly. However, here it is counting the boolean result of `time IS NOT NULL`. This will count all rows, regardless of whether time is null or not, since the expression `time IS NOT NULL` is always either true or false, both of which are counted. |

**Example 4:**

| Question | Please provide top three football players' IDs who are among the lowest potential players and prefer to use the right foot when attacking. (id=1135, database=european_football_2) |
|---|---|
| Ground-truth | `SELECT id FROM Player_Attributes WHERE preferred_foot = 'right' ORDER BY potential DESC LIMIT 3` |
| Comment | The questions asks the "lowest potential players" so the ground-query should order by descending `potential` - should not have `DESC`. |

**Wrong Evidence**

**Example 1:**

| Question | What is the eligible free rate of the 10th and 11th schools with the highest enrolment for students in grades 1 through 12?' (id=31, database=california_schools) |
|---|---|
| Ground-truth | `SELECT CAST('Free Meal Count (K-12)' AS REAL) / 'Enrollment (K-12)' FROM frpm ORDER BY 'Enrollment (K-12)' DESC LIMIT 9, 2.` |
| Evidence | K-12 refers to students in grades 1 through 12; Eligible free rate for K-12 = 'FRPM Count (K-12)' / 'Enrollment (K-12)' |
| Comment: | The evidence suggests that the information can be found in column `'FRPM Count (K-12)'` but we can see that in the ground-truth another column is actually chosen. |

**Example 2:**

| Question | Among all chemical compounds that contain molecule TR047, identify the percent that form a double-bond.' (id=287, database=toxicology) |
|---|---|
| Ground-truth | `SELECT CAST(COUNT(CASE WHEN T.bond_type = '=' THEN T.bond_id ELSE NULL END) AS REAL) * 100 / COUNT(T.bond_id) FROM bond AS T WHERE T.molecule_id = 'TR047'.` |
| Evidence | TR047 is the molecule id; double bond refers to bond_type = ' = '; percentage = DIVIDE(SUM(bond_type = ' = '), COUNT(all bond_id)) as percent where molecule_id = 'TR047' |
| Comment: | The evidence suggests that `bond_type` has spaces ' = ', whereas the ground-truth query has no spaces. |

**Ambiguous**

**Example 1:**

| Question | How many users last accessed the website after 2014/9/1? (id=533, database=codebase_community) |
|---|---|
| Ground-truth | `SELECT COUNT(Id) FROM users WHERE date(LastAccessDate) > '2014-09-01'.` |
| Comment: | It is not clear from the question whether we should include users that accessed the website exactly on the day of *2014/09/01*. |

**Evaluation Procedure**

**Example 1:**

| Question | What is the height of the tallest player? Indicate his name. (id=1021, database=european_football_2) |
|---|---|
| Ground-truth | `SELECT player_name FROM Player ORDER BY height DESC LIMIT 1` |
| SQL-PALM: | `SELECT height, player_name FROM Player ORDER BY height DESC LIMIT 1;` |
| Comment: | The generated query selects an additional column but is essentially equal to the ground-truth query. |

**Example 2:**

| Question | How many races were there in 2005? Name all the races in descending order. (id=592, database=formula_1 |
|---|---|
| Ground-truth | `SELECT name FROM races WHERE year = 2005 ORDER BY name DESC` |
| SQL-PALM: | `SELECT COUNT(raceId) FROM races WHERE year = 2005 UNION ALL SELECT name FROM races WHERE year = 2005 ORDER BY name DESC;` |
| Comment: | The generated query selects contains the correct selection of races and also includes the count with a `UNION ALL` statement. |

### A.10 Prompt examples

### A.10.1 Concise Prompt Design: 4 shot

This is a task converting text into SQL statement. We will first given the dataset schema and then ask a question in text. You are asked to generate SQL statement.

Here is an example: Convert text to SQL:

[Schema (values)]: | farm | city : city_id , official_name , status , area_km_2 , population , census_ranking | farm : farm_id , year , total_horses , working_horses , total_cattle , oxen , bulls , cows , pigs , sheep_and_goats | farm_competition : competition_id , year , theme , host_city_id , hosts | competition_record : competition_id , farm_id , rank;

[Column names (type)]: city : city_id (number)| city : official_name (text)| city : status (text)| city : area_km_2 (number)| city : population (number)| city : census_ranking ( text)| farm : farm_id (number)| farm : year (number)| farm : total_horses (number)| farm : working_horses (number)| farm : total_cattle (number)| farm : oxen (number)| farm : bulls (number)| farm : cows (number)| farm : pigs (number)| farm : sheep_and_goats (number)| farm_competition : competition_id (number)| farm_competition : year (number)| farm_competition : theme (text)| farm_competition : host_city_id (number)| farm_competition : hosts (text )| competition_record : competition_id (number)| competition_record : farm_id (number)| competition_record : rank (number);

[Primary Keys]: city : city_id | farm : farm_id | farm_competition : competition_id | competition_record : competition_id;

[Foreign Keys]: farm_competition : host_city_id equals city : city_id | competition_record : farm_id equals farm : farm_id | competition_record : competition_id equals farm_competition : competition_id

[Q]: What are the themes of farm competitions sorted by year in ascending order?;

[SQL]: select theme from farm_competition order by year asc;

Here is an example: Convert text to SQL:

[Schema (values)]: | farm | city : city_id , official_name , status , area_km_2 , population , census_ranking | farm : farm_id , year , total_horses , working_horses , total_cattle , oxen , bulls , cows , pigs , sheep_and_goats | farm_competition : competition_id , year , theme , host_city_id , hosts | competition_record : competition_id , farm_id , rank;

[Column names (type)]: city : city_id (number)| city : official_name (text)| city : status (text)| city : area_km_2 (number)| city : population (number)| city : census_ranking ( text)| farm : farm_id (number)| farm : year (number)| farm : total_horses (number)| farm : working_horses (number)| farm : total_cattle (number)| farm : oxen (number)| farm : bulls (number)| farm : cows (number)| farm : pigs (number)| farm : sheep_and_goats (number)| farm_competition : competition_id (number)| farm_competition : year (number)| farm_competition : theme (text)| farm_competition : host_city_id (number)| farm_competition : hosts (text )| competition_record : competition_id (number)| competition_record : farm_id (number)| competition_record : rank (number);

[Primary Keys]: city : city_id | farm : farm_id | farm_competition : competition_id | competition_record : competition_id; [Foreign Keys]: farm_competition : host_city_id equals city : city_id | competition_record : farm_id equals farm : farm_id | competition_record : competition_id equals farm_competition : competition_id

[Q]: What are the maximum and minimum number of cows across all farms.;

[SQL]: select max(cows), min(cows) from farm;

Here is an example: Convert text to SQL:

[Schema (values)]: | department_management | department : department_id , name , creation , ranking , budget_in_billions , num_employees | head : head_id , name , born_state , age | management : department_id , head_id , temporary_acting ( Yes );

[Column names (type)]: department : department_id (number)| department : name (text)| department : creation (text)| department : ranking (number)| department : budget_in_billions ( number)| department : num_employees (number)| head : head_id (number)| head : name (text)| head : born_state (text)| head : age (number)| management : department_id (number)| management : head_id (number)| management : temporary_acting (text);

[Primary Keys]: department : department_id | head : head_id | management : department_id;

[Foreign Keys]: management : head_id equals head : head_id | management : department_id equals department : department_id

[Q]: Show the name and number of employees for the departments managed by heads whose temporary acting value is 'Yes'?;

[SQL]: select t1.name, t1.num_employees from department as t1 join management as t2 on t1.department_id = t2.department_id where t2.temporary_acting = 'Yes';

Here is an example: Convert text to SQL:

[Schema (values)]: | farm | city : city_id , official_name , status , area_km_2 , population , census_ranking | farm : farm_id , year , total_horses , working_horses , total_cattle , oxen , bulls , cows , pigs , sheep_and_goats | farm_competition : competition_id , year , theme , host_city_id , hosts | competition_record : competition_id , farm_id , rank;

[Column names (type)]: city : city_id (number)| city : official_name (text)| city : status (text)| city : area_km_2 (number)| city : population (number)| city : census_ranking ( text)| farm : farm_id (number)| farm : year (number)| farm : total_horses (number)| farm : working_horses (number)| farm : total_cattle (number)| farm : oxen (number)| farm : bulls (number)| farm : cows (number)| farm : pigs (number)| farm : sheep_and_goats (number)| farm_competition : competition_id (number)| farm_competition : year (number)| farm_competition : theme (text)| farm_competition : host_city_id (number)| farm_competition : hosts (text)| competition_record : competition_id (number)| competition_record : farm_id (number)| competition_record : rank (number);

[Primary Keys]: city : city_id | farm : farm_id | farm_competition : competition_id | competition_record : competition_id;

[Foreign Keys]: farm_competition : host_city_id equals city : city_id | competition_record : farm_id equals farm : farm_id | competition_record : competition_id equals farm_competition : competition_id

[Q]: Show the status of the city that has hosted the greatest number of competitions.;

[SQL]: select t1.status from city as t1 join farm_competition as t2 on t1.city_id = t2.host_city_id group by t2.host_city_id order by count(*) desc limit 1;

Here is the test question to be answered: Convert text to SQL:

[Schema (values)]: | concert_singer | stadium : stadium_id , location , name , capacity , highest , lowest , average | singer : singer_id , name , country , song_name , song_release_year , age , is_male | concert : concert_id , concert_name , theme , stadium_id , year | singer_in_concert : concert_id , singer_id;

[Column names (type)]: stadium : stadium_id (number)| stadium : location (text)| stadium : name (text)| stadium : capacity (number)| stadium : highest (number)| stadium : lowest (number)|

```
stadium : average (number)| singer : singer_id (number)| singer : name (text)| singer : country
 (text)| singer : song_name (text)| singer : song_release_year (text)| singer : age (number
)| singer : is_male (others)| concert : concert_id (number)| concert : concert_name (text)|
concert : theme (text)| concert : stadium_id (text)| concert : year (text)| singer_in_concert :
 concert_id (number)| singer_in_concert : singer_id (text);
```

[Primary Keys]: `stadium : stadium_id | singer : singer_id | concert : concert_id |`
`singer_in_concert : concert_id;`

[Foreign Keys]: `concert : stadium_id equals stadium : stadium_id | singer_in_concert : singer_id`
`equals singer : singer_id | singer_in_concert : concert_id equals concert : concert_id`

[Q]: `How many singers do we have?;`

[SQL]:

## A.10.2 Verbose Prompt Design: 4 shot

```
This is a task converting text into SQL statement. We will first given the
    dataset schema and then ask a question in text. You are asked to generate
    SQL statement.
Here is an example: Let us take a question and turn it into a SQL statement about
    database tables. There are 4 tables
    . Their titles are: city, farm, farm_competition, competition_record. Table
     1 is city, and its column names and types are: City_ID (Type is number),
    Official_Name (Type is text), Status (Type is text), Area_km_2 (Type is
    number), Population (Type is number), Census_Ranking (Type is text). Table
    2 is farm, and its column names and types are: Farm_ID (Type is number),
    Year (Type is number), Total_Horses (Type is number)https://braintex.goog/
    project/6590ad9e24c22900a8955399, Working_Horses (Type is number),
    Total_Cattle (Type is number), Oxen (Type is number), Bulls (Type is number
    ), Cows (Type is number), Pigs (Type is number), Sheep_and_Goats (Type is
    number). Table 3 is farm_competition, and its column names and types are:
    Competition_ID (Type is number), Year (Type is number), Theme (Type is text
    ), Host_city_ID (Type is number), Hosts (Type is text). Table 4 is
    competition_record, and its column names and types are: Competition_ID (
    Type is number), Farm_ID (Type is number), Rank (Type is number).
    The primary keys are: city_id from Table city, farm_id from Table farm,
    competition_id from Table farm_competition, competition_id from Table
    competition_record.
    The foreign keys are: host_city_id from Table farm_competition is equivalent
    with city_id from Table city, farm_id from Table competition_record is
    equivalent with farm_id from Table farm, competition_id from Table
    competition_record is equivalent with competition_id from Table
    farm_competition. Use foreign keys to join Tables.  Let us take a text
    question and turn it into a SQL statement about database tables. The
    question is: What are the themes of farm competitions sorted by year in
    ascending order? The corresponding SQL is: SELECT Theme FROM
    farm_competition ORDER BY YEAR ASC;
Here is an example: Let us take a question and turn it into a SQL statement about
    database tables. There are 4 tables
    . Their titles are: city, farm, farm_competition, competition_record. Table
     1 is city, and its column names and types are: City_ID (Type is number),
    Official_Name (Type is text), Status (Type is text), Area_km_2 (Type is
    number), Population (Type is number), Census_Ranking (Type is text). Table
    2 is farm, and its column names and types are: Farm_ID (Type is number),
    Year (Type is number), Total_Horses (Type is number), Working_Horses (Type
    is number), Total_Cattle (Type is number), Oxen (Type is number), Bulls (
```

```
Type is number), Cows (Type is number), Pigs (Type is number),
Sheep_and_Goats (Type is number). Table 3 is farm_competition, and its
column names and types are: Competition_ID (Type is number), Year (Type is
number), Theme (Type is text), Host_city_ID (Type is number), Hosts (Type
is text). Table 4 is competition_record, and its column names and types are
: Competition_ID (Type is number), Farm_ID (Type is number), Rank (Type is
number).
```
The primary keys are: `city_id from Table city, farm_id from Table farm,`
```
competition_id from Table farm_competition, competition_id from Table
competition_record.
```
The foreign keys are: `host_city_id from Table farm_competition is equivalent`
```
with city_id from Table city, farm_id from Table competition_record is
equivalent with farm_id from Table farm, competition_id from Table
competition_record is equivalent with competition_id from Table
farm_competition. Use foreign keys to join Tables.  Let us take a text
question and turn it into a SQL statement about database tables. The
question is: What are the maximum and minimum number of cows across all
farms. The corresponding SQL is: SELECT max(Cows) ,  min(Cows) FROM farm;
```
Here is an example: `Let us take a question and turn it into a SQL statement about`
```
database tables.
```
There are 3 tables
```
. Their titles are: department, head, management. Table 1 is department,
and its column names and types are: Department_ID (Type is number), Name (
Type is text), Creation (Type is text), Ranking (Type is number),
Budget_in_Billions (Type is number), Num_Employees (Type is number). Table
2 is head, and its column names and types are: head_ID (Type is number),
name (Type is text), born_state (Type is text), age (Type is number). Table
 3 is management, and its column names and types are: department_ID (Type
is number), head_ID (Type is number), temporary_acting (Type is text).
```
The primary keys are: `department_id from Table department, head_id from Table`
```
head, department_id from Table management.
```
The foreign keys are: `head_id from Table management is equivalent with head_id`
```
from Table head, department_id from Table management is equivalent with
department_id from Table department. Use foreign keys to join Tables.
Columns with relevant values: Table management Column temporary_acting have
 values: Yes;  Only use columns with relevant values to generate SQL.  Let
us take a text question and turn it into a SQL statement about database
tables. The question is: Show the name and number of employees for the
departments managed by heads whose temporary acting value is 'Yes'? The
corresponding SQL is: SELECT T1.name ,  T1.num_employees FROM department AS
 T1 JOIN management AS T2 ON T1.department_id  =  T2.department_id WHERE T2
.temporary_acting  =  'Yes';
```
Here is an example: `Let us take a question and turn it into a SQL statement about`
```
database tables.
```
There are 4 tables
```
. Their titles are: city, farm, farm_competition, competition_record. Table
 1 is city, and its column names and types are: City_ID (Type is number),
Official_Name (Type is text), Status (Type is text), Area_km_2 (Type is
number), Population (Type is number), Census_Ranking (Type is text). Table
2 is farm, and its column names and types are: Farm_ID (Type is number),
Year (Type is number), Total_Horses (Type is number), Working_Horses (Type
is number), Total_Cattle (Type is number), Oxen (Type is number), Bulls (
Type is number), Cows (Type is number), Pigs (Type is number),
Sheep_and_Goats (Type is number). Table 3 is farm_competition, and its
column names and types are: Competition_ID (Type is number), Year (Type is
number), Theme (Type is text), Host_city_ID (Type is number), Hosts (Type
is text). Table 4 is competition_record, and its column names and types are
: Competition_ID (Type is number), Farm_ID (Type is number), Rank (Type is
number).
```
The primary keys are: `city_id from Table city, farm_id from Table farm,`

```
      competition_id from Table farm_competition, competition_id from Table
      competition_record.
      The foreign keys are: host_city_id from Table farm_competition is equivalent
      with city_id from Table city, farm_id from Table competition_record is
      equivalent with farm_id from Table farm, competition_id from Table
      competition_record is equivalent with competition_id from Table
      farm_competition. Use foreign keys to join Tables.  Let us take a text
      question and turn it into a SQL statement about database tables. The
      question is: Show the status of the city that has hosted the greatest
      number of competitions. The corresponding SQL is: SELECT T1.Status FROM
      city AS T1 JOIN farm_competition AS T2 ON T1.City_ID  =  T2.Host_city_ID
      GROUP BY T2.Host_city_ID ORDER BY COUNT(*) DESC LIMIT 1;
  Here is the test question to be answered: Let us take a question and turn it into a
      SQL statement about database tables.
      There are 4 tables. Their titles are: stadium, singer, concert,
      singer_in_concert. Table 1 is stadium, and its column names and types are:
       Stadium_ID (Type is number), Location (Type is text), Name (Type is text)
      , Capacity (Type is number), Highest (Type is number), Lowest (Type is
      number), Average (Type is number). Table 2 is singer, and its column names
       and types are: Singer_ID (Type is number), Name (Type is text), Country (
      Type is text), Song_Name (Type is text), Song_release_year (Type is text),
       Age (Type is number), Is_male (Type is others). Table 3 is concert, and
      its column names and types are: concert_ID (Type is number), concert_Name
      (Type is text), Theme (Type is text), Stadium_ID (Type is text), Year (
      Type is text). Table 4 is singer_in_concert, and its column names and
      types are: concert_ID (Type is number), Singer_ID (Type is text).
      The primary keys are: stadium_id from Table stadium, singer_id from Table
      singer, concert_id from Table concert, concert_id from Table
      singer_in_concert.
      The foreign keys are: stadium_id from Table concert is equivalent with
      stadium_id from Table stadium, singer_id from Table singer_in_concert is
      equivalent with singer_id from Table singer, concert_id from Table
      singer_in_concert is equivalent with concert_id from Table concert. Use
      foreign keys to join Tables.  Let us take a text question and turn it into
       a SQL statement about database tables. The question is: How many singers
      do we have? The corresponding SQL is:
```

## A.11   Database content

See "[Database values that related with questions]:" in red to show database content values.

```
 Here is the test question to be anwered: Convert text to SQL:
  [Schema (values)]: | california_schools | frpm : CDSCode , Academic Year ,
    County Code , District Code , School Code , County Name , District Name ,
    School Name , District Type , School Type , Educational Option Type , NSLP
    Provision Status , Charter School (Y/N) , Charter School Number , Charter
    Funding Type , IRC , Low Grade , High Grade , Enrollment (K-12) , Free Meal
     Count (K-12) , Percent (%) Eligible Free (K-12) , FRPM Count (K-12) ,
    Percent (%) Eligible FRPM (K-12) , Enrollment (Ages 5-17) , Free Meal Count
     (Ages 5-17) , Percent (%) Eligible Free (Ages 5-17) , FRPM Count (Ages
    5-17) , Percent (%) Eligible FRPM (Ages 5-17) , 2013-14 CALPADS Fall 1
    Certification Status | satscores : cds , rtype , sname , dname , cname ,
    enroll12 , NumTstTakr , AvgScrRead , AvgScrMath , AvgScrWrite , NumGE1500 |
     schools : CDSCode , NCESDist , NCESSchool , StatusType , County , District
     , School , Street , StreetAbr , City , Zip , State , MailStreet ,
    MailStrAbr , MailCity , MailZip , MailState , Phone , Ext , Website ,
    OpenDate , ClosedDate , Charter , CharterNum , FundingType , DOC , DOCType
```

```
, SOC , SOCType , EdOpsCode , EdOpsName , EILCode , EILName , GSoffered ,
GSserved , Virtual , Magnet , Latitude , Longitude , AdmFName1 , AdmLName1
, AdmEmail1 , AdmFName2 , AdmLName2 , AdmEmail2 , AdmFName3 , AdmLName3 ,
AdmEmail3 , LastUpdate;
[Column names (type)] : frpm : cdscode (text) | frpm : academic year (text) |
frpm : county code (text) | frpm : district code (number) | frpm : school
code (text) | frpm : county name (text) | frpm : district name (text) |
frpm : school name (text) | frpm : district type (text) | frpm : school
type (text) | frpm : educational option type (text) | frpm : nslp provision
 status (text) | frpm : charter school (y/n) (number) | frpm : charter
school number (text) | frpm : charter funding type (text) | frpm : irc (
number) | frpm : low grade (text) | frpm : high grade (text) | frpm :
enrollment (k-12) (number) | frpm : free meal count (k-12) (number) | frpm
: percent (%) eligible free (k-12) (number) | frpm : frpm count (k-12) (
number) | frpm : percent (%) eligible frpm (k-12) (number) | frpm :
enrollment (ages 5-17) (number) | frpm : free meal count (ages 5-17) (
number) | frpm : percent (%) eligible free (ages 5-17) (number) | frpm :
frpm count (ages 5-17) (number) | frpm : percent (%) eligible frpm (ages
5-17) (number) | frpm : 2013-14 calpads fall 1 certification status (number
) | satscores : cds (text) | satscores : rtype (text) | satscores : sname (
text) | satscores : dname (text) | satscores : cname (text) | satscores :
enroll12 (number) | satscores : numtsttakr (number) | satscores :
avgscrread (number) | satscores : avgscrmath (number) | satscores :
avgscrwrite (number) | satscores : numge1500 (number) | schools : cdscode (
text) | schools : ncesdist (text) | schools : ncesschool (text) | schools :
 statustype (text) | schools : county (text) | schools : district (text) |
schools : school (text) | schools : street (text) | schools : streetabr (
text) | schools : city (text) | schools : zip (text) | schools : state (
text) | schools : mailstreet (text) | schools : mailstrabr (text) | schools
 : mailcity (text) | schools : mailzip (text) | schools : mailstate (text)
| schools : phone (text) | schools : ext (text) | schools : website (text)
| schools : opendate (time) | schools : closeddate (time) | schools :
charter (number) | schools : charternum (text) | schools : fundingtype (
text) | schools : doc (text) | schools : doctype (text) | schools : soc (
text) | schools : soctype (text) | schools : edopscode (text) | schools :
edopsname (text) | schools : eilcode (text) | schools : eilname (text) |
schools : gsoffered (text) | schools : gsserved (text) | schools : virtual
(text) | schools : magnet (number) | schools : latitude (number) | schools
: longitude (number) | schools : admfname1 (text) | schools : admlname1 (
text) | schools : admemail1 (text) | schools : admfname2 (text) | schools :
 admlname2 (text) | schools : admemail2 (text) | schools : admfname3 (text)
 | schools : admlname3 (text) | schools : admemail3 (text) | schools :
lastupdate (time);
[Primary Keys]: frpm : CDSCode | satscores : cds | schools : CDSCode;
[Foreign Keys] : frpm : CDSCode equals schools : CDSCode | satscores : cds
equals schools : CDSCode;
[Database values that related with questions]:
 The column 'County Name' in Table 'frpm' has database values:  Alameda
The column 'cname' in Table 'satscores' has database values:  Alameda
The column 'County' in Table 'schools' has database values:  Alameda
The column 'City' in Table 'schools' has database values:  Alameda
The column 'MailCity' in Table 'schools' has database values:  Alameda
The column 'GSoffered' in Table 'schools' has database values:  K-12
The column 'GSserved' in Table 'schools' has database values:  K-12
The column 'AdmFName1' in Table 'schools' has database values:  Rae
The column 'AdmLName1' in Table 'schools' has database values:  Free
;
```

```
[Additional Info]: Eligible free rate for K-12 = 'FRPM Count (K-12)' / '
    Enrollment (K-12)'https://braintex.goog/project/6590ad9e24c22900a8955399
 [Q]: What is the highest eligible free rate for K-12 students in the schools
    in Alameda County?;
 [SQL]:
Here is an example: Convert text to SQL:
```

### A.11.1 Full column description

See "[detailed description of tables and columns]" in red to show entire column descriptions, and they are
very lengthy.

```
Here is the test question to be anwered: Convert text to SQL:
 [Schema (values)]: | california_schools | frpm : CDSCode , Academic Year ,
    County Code , District Code , School Code , County Name , District Name ,
    School Name , District Type , School Type , Educational Option Type , NSLP
     Provision Status , Charter School (Y/N) , Charter School Number , Charter
     Funding Type , IRC , Low Grade , High Grade , Enrollment (K-12) , Free
    Meal Count (K-12) , Percent (%) Eligible Free (K-12) , FRPM Count (K-12) ,
     Percent (%) Eligible FRPM (K-12) , Enrollment (Ages 5-17) , Free Meal
    Count (Ages 5-17) , Percent (%) Eligible Free (Ages 5-17) , FRPM Count (
    Ages 5-17) , Percent (%) Eligible FRPM (Ages 5-17) , 2013-14 CALPADS Fall
    1 Certification Status | satscores : cds , rtype , sname , dname , cname ,
     enroll12 , NumTstTakr , AvgScrRead , AvgScrMath , AvgScrWrite , NumGE1500
     | schools : CDSCode , NCESDist , NCESSchool , StatusType , County ,
    District , School , Street , StreetAbr , City , Zip , State , MailStreet ,
     MailStrAbr , MailCity , MailZip , MailState , Phone , Ext , Website ,
    OpenDate , ClosedDate , Charter , CharterNum , FundingType , DOC , DOCType
     , SOC , SOCType , EdOpsCode , EdOpsName , EILCode , EILName , GSoffered ,
     GSserved , Virtual , Magnet , Latitude , Longitude , AdmFName1 ,
    AdmLName1 , AdmEmail1 , AdmFName2 , AdmLName2 , AdmEmail2 , AdmFName3 ,
    AdmLName3 , AdmEmail3 , LastUpdate;
 [Column names (type)]: frpm : CDSCode (text) | frpm : Academic Year (text) |
    frpm : County Code (text) | frpm : District Code (number) | frpm : School
    Code (text) | frpm : County Name (text) | frpm : District Name (text) |
    frpm : School Name (text) | frpm : District Type (text) | frpm : School
    Type (text) | frpm : Educational Option Type (text) | frpm : NSLP
    Provision Status (text) | frpm : Charter School (Y/N) (number) | frpm :
    Charter School Number (text) | frpm : Charter Funding Type (text) | frpm :
     IRC (number) | frpm : Low Grade (text) | frpm : High Grade (text) | frpm
    : Enrollment (K-12) (number) | frpm : Free Meal Count (K-12) (number) |
    frpm : Percent (%) Eligible Free (K-12) (number) | frpm : FRPM Count (K
    -12) (number) | frpm : Percent (%) Eligible FRPM (K-12) (number) | frpm :
    Enrollment (Ages 5-17) (number) | frpm : Free Meal Count (Ages 5-17) (
    number) | frpm : Percent (%) Eligible Free (Ages 5-17) (number) | frpm :
    FRPM Count (Ages 5-17) (number) | frpm : Percent (%) Eligible FRPM (Ages
    5-17) (number) | frpm : 2013-14 CALPADS Fall 1 Certification Status (
    number) | satscores : cds (text) | satscores : rtype (text) | satscores :
    sname (text) | satscores : dname (text) | satscores : cname (text) |
    satscores : enroll12 (number) | satscores : NumTstTakr (number) |
    satscores : AvgScrRead (number) | satscores : AvgScrMath (number) |
    satscores : AvgScrWrite (number) | satscores : NumGE1500 (number) |
    schools : CDSCode (text) | schools : NCESDist (text) | schools :
    NCESSchool (text) | schools : StatusType (text) | schools : County (text)
    | schools : District (text) | schools : School (text) | schools : Street (
    text) | schools : StreetAbr (text) | schools : City (text) | schools : Zip
     (text) | schools : State (text) | schools : MailStreet (text) | schools :
```

```
     MailStrAbr (text) | schools : MailCity (text) | schools : MailZip (text)
     | schools : MailState (text) | schools : Phone (text) | schools : Ext (
     text) | schools : Website (text) | schools : OpenDate (time) | schools :
     ClosedDate (time) | schools : Charter (number) | schools : CharterNum (
     text) | schools : FundingType (text) | schools : DOC (text) | schools :
     DOCType (text) | schools : SOC (text) | schools : SOCType (text) | schools
      : EdOpsCode (text) | schools : EdOpsName (text) | schools : EILCode (text
     ) | schools : EILName (text) | schools : GSoffered (text) | schools :
     GSserved (text) | schools : Virtual (text) | schools : Magnet (number) |
     schools : Latitude (number) | schools : Longitude (number) | schools :
     AdmFName1 (text) | schools : AdmLName1 (text) | schools : AdmEmail1 (text)
      | schools : AdmFName2 (text) | schools : AdmLName2 (text) | schools :
     AdmEmail2 (text) | schools : AdmFName3 (text) | schools : AdmLName3 (text)
      | schools : AdmEmail3 (text) | schools : LastUpdate (time);
[Primary Keys]: frpm : CDSCode | satscores : cds | schools : CDSCode;
 [Foreign Keys]: frpm : CDSCode equals schools : CDSCode | satscores : cds equals
     schools : CDSCode;
 [detailed description of tables and columns]:
Column description of Table "frpm" have the following descriptions:
 Column "County Name" of Table "frpm", means "County Code"
Column "Charter School (Y/N)" of Table frpm has value descriptions "0: N;1: Y"
Column "IRC" of Table frpm has value descriptions "Not useful"
Column "Enrollment (K-12)" of Table frpm has value descriptions "commonsense
    evidence:K-12: 1st grade - 12nd grade"
Column "Free Meal Count (K-12)" of Table frpm has value descriptions "
    commonsense evidence:eligible free rate = Free Meal Count / Enrollment"
Column "FRPM Count (K-12)" of Table "frpm", means "Free or Reduced Price Meal
    Count (K-12)", has value descriptions "commonsense evidence:eligible FRPM
    rate = FRPM / Enrollment"
Column "Free Meal Count (Ages 5-17)" of Table frpm has value descriptions "
    commonsense evidence:eligible free rate = Free Meal Count / Enrollment"
Column description of Table "satscores" have the following descriptions:
Column "cds" of Table "satscores", means "California Department Schools"
Column "rtype" of Table satscores has value descriptions "unuseful"
Column "sname" of Table "satscores", means "school name"
Column "dname" of Table "satscores", means "district segment", district name,
Column "cname" of Table "satscores", means "county name"
Column "enroll12" of Table "satscores", means "enrollment (1st-12nd grade)"
Column "NumTstTakr" of Table "satscores", means "Number of Test Takers in this
    school", Number of Test Takers, , has value descriptions "number of test
    takers in each school"
Column "AvgScrRead" of Table "satscores", means "average scores in Reading"
Column "AvgScrMath" of Table "satscores", means "average scores in Math"
Column "AvgScrWrite" of Table "satscores", means "average scores in writing"
Column "NumGE1500" of Table "satscores", means "Number of Test Takers Whose
    Total SAT Scores Are Greater or Equal to 1500", has value descriptions "
    Number of Test Takers Whose Total SAT Scores Are Greater or Equal to 1500
    commonsense evidence:Excellence Rate = NumGE1500 / NumTstTakr"
Column description of Table "schools" have the following descriptions:
Column "NCESDist" of Table "schools", means "This field represents the 7-digit
    National Center for Educational Statistics (NCES) school district
    identification number. The first 2 digits identify the state and the last 5
     digits identify the school district. Combined, they make a unique 7-digit
    ID for each school district.", National Center for Educational Statistics
    school district identification number,
Column "NCESSchool" of Table "schools", means "This field represents the 5-
    digit NCES school identification number. The NCESSchool combined with the
```

NCESDist form a unique 12-digit ID for each school.", National Center for
    Educational Statistics school identification number,
Column "StatusType" of Table "schools", means "This field identifies the status
     of the district.", has value descriptions "Definitions of the valid status
     types are listed below:        Active: The district is in operation and
    providing instructional services.        Closed: The district is not in
    operation and no longer providing instructional services.        Merged: The
     district has combined with another district or districts.        Pending:
    The district has not opened for operation and instructional services yet,
    but plans to open within the next 912 months."
Column "County" of Table "schools", means "County name"
Column "StreetAbr" of Table "schools", means "The abbreviated street address of
     the school, district, or administrative authoritys physical location.",
    street address, , has value descriptions "The abbreviated street address of
     the school, district, or administrative authoritys physical location. Note
    : Some records (primarily records of closed or retired schools) may not
    have data in this field."
Column "MailStreet" of Table schools has value descriptions "The unabbreviated
    mailing address of the school, district, or administrative authority. Note:
     1) Some entities (primarily closed or retired schools) may not have data
    in this field; 2) Many active entities have not provided a mailing street
    address. For your convenience we have filled the unpopulated MailStreet
    cells with Street data."
Column "MailStrAbr" of Table "schools", means "mailing street address", has
    value descriptions "the abbreviated mailing street address of the school,
    district, or administrative authority.Note: Many active entities have not
    provided a mailing street address. For your convenience we have filled the
    unpopulated MailStrAbr cells with StreetAbr data."
Column "MailCity" of Table "schools", means "mailing city", has value
    descriptions "The city associated with the mailing address of the school,
    district, or administrative authority. Note: Many entities have not
    provided a mailing address city. For your convenience we have filled the
    unpopulated MailCity cells with City data."
Column "MailZip" of Table "schools", means "mailing zip", has value
    descriptions "The zip code associated with the mailing address of the
    school, district, or administrative authority. Note: Many entities have not
     provided a mailing address zip code. For your convenience we have filled
    the unpopulated MailZip cells with Zip data."
Column "MailState" of Table "schools", means "mailing state", has value
    descriptions "The state within the mailing address. For your convenience we
     have filled the unpopulated MailState cells with State data."
Column "Ext" of Table "schools", means "The phone number extension of the
    school, district, or administrative authority.", extension,
Column "Website" of Table "schools", means "The website address of the school,
    district, or administrative authority."
Column "OpenDate" of Table "schools", means "The date the school opened."
Column "ClosedDate" of Table "schools", means "The date the school closed."
Column "Charter" of Table "schools", means "This field identifies a charter
    school.", has value descriptions "The field is coded as follows: 1 = The
    school is a charter 0 = The school is not a charter"
Column "CharterNum" of Table "schools", means "The charter school number,", has
     value descriptions "4-digit number assigned to a charter school."
Column "FundingType" of Table "schools", means "Indicates the charter school
    funding type", has value descriptions "Values are as follows: Not in CS (
    California School) funding model Locally funded Directly funded"
Column "DOC" of Table "schools", means "District Ownership Code", has value
    descriptions "The District Ownership Code (DOC) is the numeric code used to
     identify the category of the Administrative Authority.        00 - County

```
      Office of Education        02  State Board of Education        03  Statewide
   Benefit Charter        31  State Special Schools        34  Non-school
   Location        52  Elementary School District        54  Unified School
   District        56  High School District        98  Regional Occupational
   Center/Program (ROC/P)commonsense evidence:Only the California Education
   Authority has been included in the non-school location category."
Column "DOCType" of Table "schools", means "The District Ownership Code Type is
    the text description of the DOC category.", The District Ownership Code
   Type, , has value descriptions "(See text values in DOC field description
   above)"
Column "SOC" of Table "schools", means "The School Ownership Code is a numeric
   code used to identify the type of school.", School Ownership Code, , has
   value descriptions "08 - Preschool        09  Special Education
   Schools (Public)        11  Youth Authority Facilities (CEA)        13
   Opportunity Schools        14  Juvenile Court Schools        15  Other County
    or District Programs        31  State Special Schools        60  Elementary
   School (Public)        61  Elementary School in 1 School District (Public)
        62  Intermediate/Middle Schools (Public)        63  Alternative
   Schools of Choice        64  Junior High Schools (Public)        65  K-12
   Schools (Public)        66  High Schools (Public)        67  High Schools in
   1 School District (Public)        68  Continuation High Schools        69
   District Community Day Schools        70  Adult Education Centers        98
   Regional Occupational Center/Program (ROC/P)"
Column "SOCType" of Table "schools", means "The School Ownership Code Type is
   the text description of the type of school.", School Ownership Code Type,
Column "EdOpsCode" of Table "schools", means "The Education Option Code is a
   short text description of the type of education offered.", Education Option
    Code, , has value descriptions "ALTSOC  Alternative School of Choice
   COMM  County Community School        COMMDAY  Community Day School        CON
    Continuation School        JUV  Juvenile Court School        OPP
   Opportunity School        YTH  Youth Authority School        SSS  State
   Special School        SPEC  Special Education School        TRAD  Traditional
        ROP  Regional Occupational Program        HOMHOS  Home and Hospital
        SPECON  District Consortia Special Education School"
Column "EdOpsName" of Table "schools", means "Educational Option Name", has
   value descriptions "The Educational Option Name is the long text
   description of the type of education being offered."
Column "EILCode" of Table "schools", means "The Educational Instruction Level
   Code is a short text description of the institution's type relative to the
   grade range served.", Educational Instruction Level Code, , has value
   descriptions "A  Adult        ELEM  Elementary        ELEMHIGH  Elementary-
   High Combination        HS  High School        INTMIDJR  Intermediate/Middle/
   Junior High        PS  Preschool        UG  Ungraded"
Column "EILName" of Table "schools", means "The Educational Instruction Level
   Name is the long text description of the institutions type relative to the
   grade range served.", Educational Instruction Level Name,
Column "GSoffered" of Table "schools", means "The grade span offered is the
   lowest grade and the highest grade offered or supported by the school,
   district, or administrative authority. This field might differ from the
   grade span served as reported in the most recent certified California
   Longitudinal Pupil Achievement (CALPADS) Fall 1 data collection.", grade
   span offered, , has value descriptions "For example XYZ School might
   display the following data:GSoffered = PAdultGSserved = K12"
Column "GSserved" of Table "schools", means "It is the lowest grade and the
   highest grade of student enrollment as reported in the most recent
   certified CALPADS Fall 1 data collection. Only K12 enrollment is reported
   through CALPADS. This field may differ from the grade span offered.", grade
    span served., , has value descriptions "commonsense evidence:1. Only K12
```

enrollment is reported through CALPADS2. Note: Special programs at
    independent study, alternative education, and special education schools
    will often exceed the typical grade span for schools of that type"
Column "Virtual" of Table "schools", means "This field identifies the type of
    virtual instruction offered by the school. Virtual instruction is
    instruction in which students and teachers are separated by time and/or
    location, and interaction occurs via computers and/or telecommunications
    technologies.", has value descriptions "The field is coded as follows: F =
    Exclusively Virtual  The school has no physical building where students
    meet with each other or with teachers, all instruction is virtual. V =
    Primarily Virtual  The school focuses on a systematic program of virtual
    instruction but includes some physical meetings among students or with
    teachers. C = Primarily Classroom  The school offers virtual courses but
    virtual instruction is not the primary means of instruction. N = Not
    Virtual  The school does not offer any virtual instruction. P = Partial
    Virtual  The school offers some, but not all, instruction through virtual
    instruction. Note: This value was retired and replaced with the Primarily
    Virtual and Primarily Classroom values beginning with the 201617 school
    year."
Column "Magnet" of Table "schools", means "This field identifies whether a
    school is a magnet school and/or provides a magnet program.", has value
    descriptions "The field is coded as follows: Y = Magnet - The school is a
    magnet school and/or offers a magnet program. N = Not Magnet - The school
    is not a magnet school and/or does not offer a magnet program.commonsense
    evidence:Note: Preschools and adult education centers do not contain a
    magnet school indicator."
Column "Latitude" of Table "schools", means "The angular distance (expressed in
     degrees) between the location of the school, district, or administrative
    authority and the equator measured north to south."
Column "Longitude" of Table "schools", means "The angular distance (expressed
    in degrees) between the location of the school, district, or administrative
     authority and the prime meridian (Greenwich, England) measured from west
    to east."
Column "AdmFName1" of Table "schools", means "administrator's first name", has
    value descriptions "The superintendents or principals first name.
    commonsense evidence:Only active and pending districts and schools will
    display administrator information, if applicable."
Column "AdmLName1" of Table "schools", means "administrator's last name", has
    value descriptions "The superintendents or principals last name.commonsense
     evidence:Only active and pending districts and schools will display
    administrator information, if applicable."
Column "AdmEmail1" of Table "schools", means "administrator's email address",
    has value descriptions "The superintendents or principals email address.
    commonsense evidence:Only active and pending districts and schools will
    display administrator information, if applicable."
Column "AdmFName2" of Table schools has value descriptions "SAME as 1"
Column "AdmFName3" of Table schools has value descriptions "not useful"
Column "AdmLName3" of Table schools has value descriptions "not useful"
Column "AdmEmail3" of Table schools has value descriptions "not useful"
Column "LastUpdate" of Table schools has value descriptions "when is this
    record updated last time"
;
  [Database values that related with questions]
The column 'County Name' in Table 'frpm' has database values: Alameda
The column 'cname' in Table 'satscores' has database values: Alameda
The column 'County' in Table 'schools' has database values: Alameda
The column 'City' in Table 'schools' has database values: Alameda
The column 'MailCity' in Table 'schools' has database values: Alameda

```
The column ‘GSoffered‘ in Table ‘schools‘ has database values: K-12
The column ‘GSserved‘ in Table ‘schools‘ has database values: K-12
The column ‘AdmFName1‘ in Table ‘schools‘ has database values: Rae
The column ‘AdmLName1‘ in Table ‘schools‘ has database values: Free
;
  [Additional Info]: Eligible free rate for K-12 = ‘FRPM Count (K-12)‘ / ‘
     Enrollment (K-12)‘
  [Q]: What is the highest eligible free rate for K-12 students in the schools
     in Alameda County?;
  [SQL]:
Here is an example: Convert text to SQL:
```

### A.11.2   Inferred column selection

See "[detailed description of tables and columns]" in red for soft column selection.

```
Here is the test question to be anwered: Convert text to SQL:
  [Schema (values)]: | california_schools | frpm : CDSCode , Academic Year ,
     County Code , District Code , School Code , County Name , District Name ,
     School Name , District Type , School Type , Educational Option Type , NSLP
     Provision Status , Charter School (Y/N) , Charter School Number , Charter
     Funding Type , IRC , Low Grade , High Grade , Enrollment (K-12) , Free Meal
      Count (K-12) , Percent (%) Eligible Free (K-12) , FRPM Count (K-12) ,
     Percent (%) Eligible FRPM (K-12) , Enrollment (Ages 5-17) , Free Meal Count
      (Ages 5-17) , Percent (%) Eligible Free (Ages 5-17) , FRPM Count (Ages
     5-17) , Percent (%) Eligible FRPM (Ages 5-17) , 2013-14 CALPADS Fall 1
     Certification Status | satscores : cds , rtype , sname , dname , cname ,
     enroll12 , NumTstTakr , AvgScrRead , AvgScrMath , AvgScrWrite , NumGE1500 |
      schools : CDSCode , NCESDist , NCESSchool , StatusType , County , District
      , School , Street , StreetAbr , City , Zip , State , MailStreet ,
     MailStrAbr , MailCity , MailZip , MailState , Phone , Ext , Website ,
     OpenDate , ClosedDate , Charter , CharterNum , FundingType , DOC , DOCType
     , SOC , SOCType , EdOpsCode , EdOpsName , EILCode , EILName , GSoffered ,
     GSserved , Virtual , Magnet , Latitude , Longitude , AdmFName1 , AdmLName1
     , AdmEmail1 , AdmFName2 , AdmLName2 , AdmEmail2 , AdmFName3 , AdmLName3 ,
     AdmEmail3 , LastUpdate;
  [Column names (type)]: frpm : CDSCode (text) | frpm : Academic Year (text) | frpm
     : County Code (text) | frpm : District Code (number) | frpm : School Code
     (text) | frpm : County Name (text) | frpm : District Name (text) | frpm :
     School Name (text) | frpm : District Type (text) | frpm : School Type (text
     ) | frpm : Educational Option Type (text) | frpm : NSLP Provision Status (
     text) | frpm : Charter School (Y/N) (number) | frpm : Charter School Number
      (text) | frpm : Charter Funding Type (text) | frpm : IRC (number) | frpm :
      Low Grade (text) | frpm : High Grade (text) | frpm : Enrollment (K-12) (
     number) | frpm : Free Meal Count (K-12) (number) | frpm : Percent (%)
     Eligible Free (K-12) (number) | frpm : FRPM Count (K-12) (number) | frpm :
     Percent (%) Eligible FRPM (K-12) (number) | frpm : Enrollment (Ages 5-17) (
     number) | frpm : Free Meal Count (Ages 5-17) (number) | frpm : Percent (%)
     Eligible Free (Ages 5-17) (number) | frpm : FRPM Count (Ages 5-17) (number)
      | frpm : Percent (%) Eligible FRPM (Ages 5-17) (number) | frpm : 2013-14
     CALPADS Fall 1 Certification Status (number) | satscores : cds (text) |
     satscores : rtype (text) | satscores : sname (text) | satscores : dname (
     text) | satscores : cname (text) | satscores : enroll12 (number) |
     satscores : NumTstTakr (number) | satscores : AvgScrRead (number) |
     satscores : AvgScrMath (number) | satscores : AvgScrWrite (number) |
     satscores : NumGE1500 (number) | schools : CDSCode (text) | schools :
```

```
    NCESDist (text) | schools : NCESSchool (text) | schools : StatusType (text)
     | schools : County (text) | schools : District (text) | schools : School (
    text) | schools : Street (text) | schools : StreetAbr (text) | schools :
    City (text) | schools : Zip (text) | schools : State (text) | schools :
    MailStreet (text) | schools : MailStrAbr (text) | schools : MailCity (text)
     | schools : MailZip (text) | schools : MailState (text) | schools : Phone
    (text) | schools : Ext (text) | schools : Website (text) | schools :
    OpenDate (time) | schools : ClosedDate (time) | schools : Charter (number)
    | schools : CharterNum (text) | schools : FundingType (text) | schools :
    DOC (text) | schools : DOCType (text) | schools : SOC (text) | schools :
    SOCType (text) | schools : EdOpsCode (text) | schools : EdOpsName (text) |
    schools : EILCode (text) | schools : EILName (text) | schools : GSoffered (
    text) | schools : GSserved (text) | schools : Virtual (text) | schools :
    Magnet (number) | schools : Latitude (number) | schools : Longitude (number
    ) | schools : AdmFName1 (text) | schools : AdmLName1 (text) | schools :
    AdmEmail1 (text) | schools : AdmFName2 (text) | schools : AdmLName2 (text)
    | schools : AdmEmail2 (text) | schools : AdmFName3 (text) | schools :
    AdmLName3 (text) | schools : AdmEmail3 (text) | schools : LastUpdate (time)
    ;
 [Primary Keys]: frpm : CDSCode | satscores : cds | schools : CDSCode;
 [Foreign Keys]: frpm : CDSCode equals schools : CDSCode | satscores : cds
    equals schools : CDSCode;
  [detailed description of tables and columns]:
Column description of Table "frpm" have the following descriptions:
Column "County Name" of Table "frpm", means "County Code"
Column "Enrollment (K-12)" of Table frpm has value descriptions "commonsense
    evidence:K-12: 1st grade - 12nd grade"
Column "FRPM Count (K-12)" of Table "frpm", means "Free or Reduced Price Meal
    Count (K-12)", has value descriptions "commonsense evidence:eligible FRPM
    rate = FRPM / Enrollment"
;

  [Database values that related with questions]
The column 'County Name' in Table 'frpm' has database values: Alameda
The column 'cname' in Table 'satscores' has database values: Alameda
The column 'County' in Table 'schools' has database values: Alameda
The column 'City' in Table 'schools' has database values: Alameda
The column 'MailCity' in Table 'schools' has database values: Alameda
The column 'GSoffered' in Table 'schools' has database values: K-12
The column 'GSserved' in Table 'schools' has database values: K-12
The column 'AdmFName1' in Table 'schools' has database values: Rae
The column 'AdmLName1' in Table 'schools' has database values: Free
;
 [Additional Info]: Eligible free rate for K-12 = 'FRPM Count (K-12)' / '
    Enrollment (K-12)'
  [Q]: What is the highest eligible free rate for K-12 students in the schools
     in Alameda County?;
  [SQL]:
```

