# OpenReview forum: "SQL-PaLM: Improved large language model adaptation for Text-to-SQL"
_TMLR — Accepted by TMLR_

### Review · Reviewer_wZE4 · 2024-05-01

**Summary Of Contributions:**

This paper presents SQL-PaLM, a comprehensive framework that enables LLMs to handle Text-to-SQL tasks. It explores two learning paradigms, few-shot prompting and instruction fine-tuning of LLMs, to assess their performance and effectiveness. Additionally, the paper analyzes the impact of factors such as input representation, database size, data diversity, coverage, and augmentation strategies, such as using LLMs to generate additional data. To evaluate the proposed SQL-PaLM framework, extensive experiments and analyses are conducted on public Text-to-SQL benchmarks like Spider and BIRD, aiming to gain insights into the framework's strengths, weaknesses, and the key factors influencing its performance.

**Audience:**

Yes

**Claims And Evidence:**

Yes

**Requested Changes:**

See weaknesses

**Strengths And Weaknesses:**

## Strengths
S1. The paper extensively experiments with the components of the SQL-PaLM framework on Text-to-SQL benchmarks (Spider and BIRD). Through ablation studies, the authors analyze the impact of factors like data diversity, input representations, column selection approaches, etc. The results offer insights into the proposed methods' strengths and limitations.

S2.  The paper addresses challenges in applying Text-to-SQL to real-world scenarios, including large database schemas, ambiguous questions, and complex SQL queries. Tackling these practical issues enhances the potential real-world impact of the work. The authors identify and report errors present in the benchmarks mentioned earlier.

## Weaknesses
W1. It appears that this work lacks a timely discussion beyond few-shot prompting and instruction fine-tuning of LLMs. Recent solutions based on LLM-agents and RAG models have achieved state-of-the-art performance on official leaderboards, such as [1].

[1] MAC-SQL: A Multi-Agent Collaborative Framework for Text-to-SQL

W2. All scores are reported based on the dev set. It would be more credible to evaluate the proposed methods on the test set.

W3. While I appreciate the effort put into this comprehensive work, it is worth noting that several of the components have already been explored in prior research and have not been sufficiently compared with in this study. For example, [2-7] are all works on synthesizing data for Text-to-SQL.

[2] GraPPa: Grammar-Augmented Pre-Training for Table Semantic Parsing

[3] TaBERT: Pretraining for Joint Understanding of Textual and Tabular Data

[4] Importance of Synthesizing High-quality Data for Text-to-SQL Parsing

[5] Learning to Synthesize Data for Semantic Parsing

[6] Data Augmentation with Hierarchical SQL-to-Question Generation for Cross-domain Text-to-SQL Parsing

[7] CodeS: Towards Building Open-source Language Models for Text-to-SQL

---

> ### Author Response · Authors · 2024-05-08
> **Thank you for the review! Re: W1 and W2**
>
> We thank the reviewer for the very insightful and constructive review! With your suggestions, our paper can be improved significantly. Please check our response below.  We have updated our manuscript based on your suggestions (highlighted in pink). Let us know if you have any other questions.
>
> ### **W1 missing the discussion about LLM-agents (such as MAC-SQL [1] )**
>
> Thanks a lot for suggesting the direction of "LLM agent"  for LLMs (“MAC-SQ”[1]). “MAC-SQL” consists of a selector (to decompose a large database into a smaller sub-database), decomposer (to decompose complex questions into subqueries), and refiner (to use SQL execution feedback).  We have updated the manuscript to include the discussion of  the "LLM agent line" (e.g. MAC-SQL) of work in the “related work” section. It will definitely help our paper to provide a more comprehensive view of Text-to-SQL, and improve our paper's scope.
>
> We also provide the performance evaluation of MAC-SQL compared with SQL-PaLM  in Table 24 BIRD and Table 25 SPIDER . We also copied results here: Table 1 and 2 below show our results are better than MAC-SQL on BIRD and SPIDER.
>
>
> **Table 1: BIRD**:
> |                     |                             |
> |:-------------------:|:---------------------------:|
> |                     | Execution Accuracy (EX) |
> | SQL-PaLM (Ours) |             **61.7**            |
> |   [1] MAC-SQL   |            59.39            |
> |                     |                             |
>
> **Table 2: SPIDER**:
> |                 |                         |                          |
> |-----------------|:-------------------------:|:--------------------------:|
> |                 | Execution Accuracy (EX) | Test Suite Accuracy (TS) |
> | SQL-PaLM (Ours) |           **87.3**          |           83.5           |
> | [1] MAC-SQL     |          86.75          |               -           |
> |                 |                         |                          |
>
>
>
> ### **Missing RAG methods discussion**
>
> Regarding RAG, we provide discussion about retrieval based approach in “Section 3.2 Retrieval-based methods” in the related work section. Let us know if you have any other good retrieval methods.
>
>
> ### **W2: evaluate on dev split, not test split**
>
>
> While we agree on the benefits of evaluating on leaderboard, unfortunately, test splits of the leaderboards are held private from the public use. It is not permitted to release the weights of the models we used per our organization’s policy. Hence, we are unable to obtain the results on the leaderboards. To better address this concern, we are happy to release the SQL outputs generated by our model. Hopefully the evaluation on various different benchmarks (BIRD and SPIDER) can address some of the concerns about  generalizability and robustness of our approach.

---

> ### Author Response · Authors · 2024-05-08
> **Thank you for the review! Re: W3**
>
> ### **W3: missing the discussion about previous work on "synthesizing data for Text-to-SQL" [2-7]**
>
> Thank you for the insightful feedback! As the reviewer highlighted, synthetic data is indeed an important part of our proposed framework. We have updated this part in the revised manuscript.  It is certainly beneficial to discuss references [2-7] regarding "synthetic data" in the "Related Work" section (Sec. 3.2). This addition will offer a broader perspective and enrich the background information for our readers.
>
> We have updated this part in the revised manuscript and copied below:
>  "Regarding synthetic data for text-to-SQL task, [2] GRAPPA and  [3] TaBERT propose a pretraining framework to jointly train structured schema of DB tables and questions. [2] proposes a synchronous context-free grammar: extracts question-SQL template in a “context-free” manner from existing dataset, and applies them to other databases. [4] proposes synthetic framework with emphasis on “strong-typing”, “key relations”,  only including “schema-distance-weighted” relevant columns. [5][6] synthesize SQL using context-free grammar rules (abstract syntax tree grammar) , and then generate the corresponding natural question based on the synthesized SQLs".
>
> Additionally, we also provide evaluation of SQL-PaLM with these methods. Table 3 and Table 4 below show our method outperform [2-7] clearly on BIRD and SPIDER.
>
> Note that the BIRD datasets were developed later, so earlier works don’t have BIRD evaluation.
>
> **Evaluation metrics**:
>
> Exact Match (EM): generated SQL matches exactly gold SQL
>
> Execution Accuracy (EX): execution output of generated SQL matches that of gold SQL
>
> Test Suite Accuracy (TS): multiple test cases (each test is EX)
>
> **Comments**:
> EM was initially developed but is suboptimal because multiple valid SQL queries can correspond to a single question, making mere SQL matching insufficient. To address these limitations, EX and TS were later introduced. Early research only utilized EM for evaluation without EX [2,3], but more recent studies [4,5] that include both EM and EX evaluation allow for a comparative analysis of performance.
>
>
> **Table 3: BIRD**
> |                                                                                                         | Execution Accuracy (EX) |
> |---------------------------------------------------------------------------------------------------------|:-----------------------:|
> | SQL-PaLM (Ours)                                                                                         |           61.7          |
> | [2] GraPPa                                                                                              |            -            |
> | [3] TaBERT                                                                                              |            -            |
> | [4] Importance  ...                                |            -            |
> | [5] Learning to ...                                                    |            -            |
> | [6] Data Augmentation ... |            -            |
> | [7] CodeS                                 |          58.47          |
>
> **Table 4 SPIDER**
> |                 | Execution Accuracy (EX) | Test Suite Accuracy (TS) | Exact Match (EM) |
> |-----------------|:-----------------------:|:------------------------:|:----------------:|
> | SQL-PaLM (Ours) |           **87.3**          |           **83.5**           |         -        |
> | [2] GraPPa      |            -             |               -           |       73.4       |
> | [3] TaBERT      |            -             |             -             |       64.5       |
> | [4]             |           81.4          |              -            |       76.1       |
> | [5]             |           72.5          |              -            |       71.8       |
> | [6]             |                         |                -          |       68.2       |
> | [7]     CodeS         |           85.4          |           80.3           |      -            |

---

### Review · Reviewer_AbU9 · 2024-06-12

**Summary Of Contributions:**

This paper does a major study into various aspects of building a practical and usable Text-to-SQL system using large LLMs and delves deeply into various critical aspects that a researcher/practitioner has to consider—learning (to ICL or to fine-tune), how does parameter-efficent fine-tuning help (to Lora or not?), how well do models generalize after being fine-tuned on smaller training sets, how to handle ambiguousness of real world queries, complex SQL queries requiring multi-step reasoning, whether to select specific columns relevant for the query, etc. The paper goes into great detail about every step of the system.

A few notable results/findings from the paper - (a) Fine-tuning is more effective than prompting for more complex Text-to-SQL tasks, (b) Full fine-tuning is better than Lora (PEFT) even for few training examples, (c) column/table selection is important, (d) synthetic training data helps, but complex examples are more helpful, (e) given that SQL queries are executable, this is advantageous in program aided column selection as well as execution-based test-time selection.

**Audience:**

Yes

**Claims And Evidence:**

Yes

**Requested Changes:**

* Please add details about Lora fine-tuning
* In equation 14, It is not clear how the similarity score is computed. Is it also computed by the LLM?

**Strengths And Weaknesses:**

Strengths

* This paper does a _major_ study in building production ready text-to-SQL systems with large LLMs and contain important insights that can be used by several practitioners
* Although, the paper is too long, its straightforward to follow and is clearly written

Weakness:

* Not exactly a weakness, but the paper does not bring in a new technical novelty but instead chooses to test several configurations of building a text-to-sql model
* In the fine-tuning section (Sec 6), I did not see any details of Lora fine-tuning. Please add details about the exact Lora experiments.

---

> ### Author Response · Authors · 2024-06-26
> **Thank you for the review!**
>
> We thank the reviewer for the insightful comments! We appreciate the reviewers’ time and efforts in reading our paper and provide helpful suggestions to improve our paper. We have updated our manuscript to reflect your feedbacks, and the modifications are highlighted in pink. Let us know if you have further questions.
>
> ### **Re: Novelty:**
>
> Thanks! As the Reviewer indicates, we primarily focus on systematic understanding of different factors in designing a state-of-the-art Text-to-SQL system. In addition to holistic understanding of the system, we also bring novelty by proposing how to optimally integrate retrieving relevant tables and columns using program-assisted or retrieval-based approach, and query-specific database content values to improve Text-to-SQL performance.
>
> ### **Re: Details of LoRA finetuning**:
> Thanks for the great suggestions! Yes, we have included the details of LoRA fine tuning in our updated manuscript (Sec 7.3 experiments), and copied here.
>
> “LoRA finetuning: Following Hu et al. (2021), we incorporate trainable linear low-rank modules into the query and value projections of each self-attention layer. We set the rank of LoRA to 32, learning rate to 1e-4, and model architecture is Gecko PaLM model.”
>
> ### **Re: eq 14**:
> Yes, similarity scores are outputted by the LLM, by querying the LLM to generate different SQLs and output similarity scores together. The details of prompt design is in “A.5.1 Synthetic data prompt design”.  We have also updated our manuscript to clarify this.

---

### Review · Reviewer_tt9k · 2024-08-06

**Summary Of Contributions:**

The paper proposes a framework for fine-tuning and prompting the PaLM LLM for a text-to-SQL task. The framework is composed of multiple components such as synthetic data augmentation, fine-tuning, prompt design, execution-based consistency decoding, table and column selection, query-specific database content integration, and a method for selecting the appropriate training paradigm at test time. The framework is evaluated on the dev sets of two prominent Text-to-SQL benchmarks, Spider (and its variants) and BIRD, and demonstrated competitive performance. In addition, the paper performs ablation studies to understand the impact each component has on the overall text-to-SQL performance.

**Audience:**

Yes

**Broader Impact Concerns:**

The proposed framework is built around a closed PaLM model and will be difficult to reproduce.

**Claims And Evidence:**

Yes

**Requested Changes:**

1. Various typos throughout the document. Needs proof reading.
2. Table 7 has results for Gecko as 15.84% and Table 8 suggests this might be for LoRA fine-tuning and that the results for Gecko in Table 7 should have been 33.96?
3. It's not clear what baseline is used in Table 17 and 18.

**Strengths And Weaknesses:**

Strengths
1. A performant text-to-SQL model
2. Ablation on various components of the pipeline, showcasing the role of each component.
3. Experiments and discussion around practical implications of several choices of implementations for components.

Weaknesses
1. Reliant on expensive, large-scale LLMs as opposed to efficient, specialised small LLMs (e.g, CodeS), especially when needing to fine-tune for each new set of databases.
2. No evaluation on test data on the leaderboards. The test splits being hosted by benchmarks' creators is not a reason against evaluating on them.
3. No evaluation on execution efficiency score for BIRD.
4. Prompt design doesn't generalise across different datasets (Spider vs BIRD) and might not generalise across LLMs.
5. Synthetic data augmentation relies on existing annotated samples. It's not clear how to extend to cases without manual annotations.
6. Comparison against SOTA baseline uses both Spider + BIRD data for fine-tuning whereas baselines don't.
7. Execution-based consistency decoding and test-time refinement via execution-based selection are costly as they require multiple inferences over multiple training paradigms and multiple executions on the database before selecting the correct SQL.

---

> ### Author Response · Authors · 2024-08-12
> **Thank you for the review!**
>
> Thank you so much for your detailed feedback and for acknowledging the comprehensiveness of our experimental conditions and ablation studies!  We really appreciate your time and effort spent on our paper. With your feedback, our paper can be improved significantly. We have updated our manuscript and the modifications are highlighted in pink.
>
> Please see below for our response to the questions and concerns.
>
> ### **Q1: Focus on large LLMs**
>
> Thanks for raising this important point. While we appreciate the benefits of tuning small LLMs as many other papers reported (e.g. codeS[1]), our paper focuses on pushing the best achievable performance with tuning of large LLMs. We focused on large LLMs, filling a gap in the field where most current papers focus on smaller models due to computational resource constraints.
>
> We note that there are exacerbated generalization and computational challenges when it comes to larger-scale LLM tuning, and that’s why we chose to focus on this regime of models as one of the first papers. We believe at the end, for Text2SQL systems to make real-world impact, their accuracy reaching human performance would be of topmost importance and currently tuning of larger LLMs are much closer to achieving that compared to tuning of smaller LLMs. Larger LLMs can be more expensive than smaller counterparts, but considering the time value of humans, they are still very lucrative to the vast majority of applications.
>
> We also note that we investigate the impact of model size in Table 7. The results indicate that larger models can have significantly improved performance - the smaller LLM Gecko yields 33% while the bigger model Unicorn yields 55%.
>
> ref [1] https://arxiv.org/abs/2402.16347
>
> ### **Q2: Test split results**
>
> While we agree on the benefits of evaluating on leaderboard, unfortunately, test splits of the leaderboards are held private from the public use. It is not permitted to release the weights of the models we used per our organization’s policy. Hence, we are unable to obtain the results on the leaderboards. To better address this concern, we are happy to release the SQL outputs generated by our model. Hopefully the evaluation on different benchmarks (BIRD and SPIDER) can address some of the concerns on generalizability and robustness of our approach.
>
> ### **Q3 Missing execution efficiency scores**
>
> This is indeed a critical point, thanks for bringing it up. Following most papers, we reported execution accuracy (EX), as it is the most critical measurement for comparison with other works.
>
> We agree with the reviewer that the Valid Efficiency Score (VES) measurement is valuable, as it offers a unique evaluation perspective, combining both efficiency and accuracy.So we have evaluated SQL-PaLM results for VES. We have added the VES score in our updated manuscript and below. The results indicate our approach perform significantly better on VES score (15% better), indicating the superior performance of SQL-PaLM from efficacy and accuracy perspective
>
>
> | Methods/Datasets EX              |   |       |
> |----------------------------------|---|-------|
> | Tuning SFT CodeS-15B      |   | 58.47 |
> | Codex                     |   | 43.41 |
> | ChatGPT                   |   | 43.81 |
> | GPT-4                     |   | 49.77 |
> | DIN-SQL + GPT-4           |   | 58.79 |
> | DAIL-SQL + GPT-4          |   | 56.08 |
> | MAC-SQL + GPT-4           |   | 58.76 |
> | Dubo-SQL    |   | 66.01 |
> | MCS-SQL + GPT-4               |   | 64.82 |
> | Fine-tuned SQL-PaLM (Ours) |   | **79.35** |
>
>
> ### **Q4: Generalization of the prompt design**
>
> We note that the same prompt design is used for both BIRD and SPIDER. We added a note to clarify this in Sec 8.
>
> Indeed, this is why we can combine BIRD and SPIDER to train a LLM to  get a better performance when testing on BIRD or SPIDER than training on an individual dataset (Sec 8.2.4 and Table 9 and Table 10). Our prompt design approach is dataset agnostic and robust to distribution shifts.
>
> ### **Q5: Synthetic data needs label data**
>
> Yes, this paper focuses on a more ‘conservative’ synthetic data approach (schema and questions unchanged, only SQL changed), where we can evaluate the quality of synthetic data (by re-using current evaluation as questions are unchanged).
> By doing so, we can eliminate the confounding factors related to the unknown quality of synthetic data affecting the final Text-to-SQL performance.
>
> In this paper,  we only investigate the performance for this particular synthetic data. We leave for more open-ended synthetic data generation methods e.g. (altering questions, SQL, or data schema, etc) for future work.  We have highlighted those in the updated manuscript in Sec 6.3.3.

---

> > ### Author Response · Authors · 2024-08-12
> > **Continued**
> >
> > ### **Q6 Combining BIRD + SPIDER for benchmarks**
> >
> > Yes, we agree on the fairness aspect.  However, this paper focuses on investigating different factors in influencing Text-to-SQL performance: “combining BIRD and SPIDER” is one investigation of diverse training data that helps with fine-tuning performance.
> >
> > While this paper provides performance comparisons (for readers to understand the performance Table 24 and 25),  it doesn’t focus on achieving or claiming state-of-the-art performance.
> >
> > ### **Q7 Execution-based consistency has higher cost**
> >
> > Yes, we agree. Execution-based consistency yields a tradeoff between cost and performance. We agree obtaining multiple samples for finetuning requires more compute budget.  We note that this sampling for in-context learning cost is not proportional to the number of samples though, many LLM APIs yield sublinear pricing and almost no increase in time with decoding many output samples in parallel. Overall, this paper focuses on investigation of factors in improving the performance and such an advanced decoding approach is part of it.
> > ### **Q8: typos**
> >
> > Thank you for the suggestions! We will do careful proofreading and fix typos.
> >
> > ### **Q9: Table 7 results**
> >
> > Yes, we appreciate your careful reading in identifying this typo. We have updated Table 7 in our manuscript. On the smaller model Gecko, the  15.84% is for Lora tuning; whereas 33.96% is full supervised tuning.
> >
> > ### **Q10: Clarifications on Tables 17 and 18**
> >
> > Yes, Table 17 is about how to integrate selected columns into supervised fine-tuning setup – a comparison between soft vs hard column selection. As we describe in Equations 28 and 29 in Sec 6.3.5 “Table and column selection”, hard column selection refers to only include columns that are selected and remove the unselected columns, whereas soft column selection refers to only including column descriptions of the selected column (emphasize them in prompt) while not removing the unselected columns. The way to identify selected columns is a program-aided approach for its good performance.
> >
> > Table 18 is ablation studies on soft column selection. One ablation study is to understand the performance of different ways to identify the selected columns or tables: program-aided  (“+ Soft column selection (program-aided)” in Table 18) or retrieval-based  (“ + Soft column selection (retrieval-based)” in Table 18). Other ablation studies are to understand the upperboard – we use ground truth selected columns (“+ Ground truth column selection (oracle)” in Table 18), or to understand performance if we include entire column descriptions to the input  (“+ Full column descriptions” in Table 18).
> >
> > We have updated the manuscript to add descriptions in the text more clearly.

---

### Author Response · Authors · 2024-08-12
**Updated manuscript to reflect the suggestions**

Thank you so much for all the insightful and valuable feedback. Based on the suggestion, we have updated the paper draft with the following modifications (highlighted in pink):

- Sec 3.2: Added “agent view” of Text-to-SQL in related-work section, discussing about MAC-SQL (reviewer wZE4)
- Sec 3.2:  Added “synthetic data methods” in related-work, where GraPPa, TaBERT and other works are discussed.  (reviewer wZE4),
- Added details of LoRA in Sec. 7.3 Experiments (Reviewer AbU9)
- Clarified Eq. 14  (Reviewer AbU9)
- Added VES score evaluation in Table 24 and Sec. 7.5  Evaluation (Reviewer tt9k)
- Clarified Table 17 and 18 for  column selection on page 26 and 27 (Reviewer tt9k)
- Updated Table 7 in Sec. 8.2.2 (Reviewer tt9k)

We appreciate all these suggestions that have helped to improve the quality of our paper, thank you! Let us know if you have further questions.

---

### Author Response · Authors · 2024-08-18
**Look forward to your response to our rebuttal**

Dear Reviewers,

We thank all reviewers for your time spent carefully reviewing our work, for your helpful and constructive feedback, as well as for generally acknowledging the contribution of the work and comprehensiveness of our experiments!  Your constructive comments have significantly contributed to the improvement of our work.

We have responded to each point raised in each review and hope that our responses and experimental evidence will help to ease any concerns.  In response to your valuable feedback, we have revised our manuscript and highlighted the modifications in pink for your convenience.

The author-reviewer discussion period will end soon. If you have any further questions or suggestions, please let us know and we respond as soon as possible. Looking forward to further discussion.

---

### Decision · Action_Editor_3WT8 · 2024-10-18

**Recommendation:** Accept as is

**Comment:**

All reviewers found the work interesting and well-executed. The reviewers made suggestions to include additional metrics such as VES, a discussion on LoRA finetuning for their framework, and comparisons with additional prior works, which were addressed by the authors during the rebuttal and should be included in the final revision.

**Audience:**

The problem of using LLMs for NLP applications such as text-to-SQL generation is relevant to the TMLR community.

**Claims And Evidence:**

This work explores approaches for using LLMs to generate structured query language (SQL). All reviewers agreed that the paper was well-written and the main claims were well-justified through empirical analysis, including careful ablation studies and a discussion on both benefits and limitations of the framework.